# A peptide encoded by circular form of *LINC-PINT* suppresses oncogenic transcriptional elongation in glioblastoma

Maolei Zhang[1,2], Kun Zhao[1,2], Xiaoping Xu[1,2], Yibing Yang[1,2], Sheng Yan[1,2,3], Ping Wei[4], Hui Liu[5], Jianbo Xu[6], Feizhe Xiao[7], Huangkai Zhou[1,2,8], Xuesong Yang[1,2], Nunu Huang[1,2], Jinglei Liu[1,2], Kejun He[1,2], Keping Xie[9], Gong Zhang[10], Suyun Huang [11,12] & Nu Zhang [1,2]

Circular RNAs (circRNAs) are a large class of transcripts in the mammalian genome. Although the translation of circRNAs was reported, additional coding circRNAs and the functions of their translated products remain elusive. Here, we demonstrate that an endogenous circRNA generated from a long noncoding RNA encodes regulatory peptides. Through ribosome nascent-chain complex-bound RNA sequencing (RNC-seq), we discover several peptides potentially encoded by circRNAs. We identify an 87-amino-acid peptide encoded by the circular form of the long intergenic non-protein-coding RNA *p53*-induced transcript (*LINC-PINT*) that suppresses glioblastoma cell proliferation in vitro and in vivo. This peptide directly interacts with polymerase associated factor complex (PAF1c) and inhibits the transcriptional elongation of multiple oncogenes. The expression of this peptide and its corresponding circRNA are decreased in glioblastoma compared with the levels in normal tissues. Our results establish the existence of peptides encoded by circRNAs and demonstrate their potential functions in glioblastoma tumorigenesis.

[1] Department of Neurosurgery, First Affiliated Hospital of Sun Yat-sen University, Guangzhou 510080 Guangdong Province, PR China. [2] Guangdong Provincial Key Laboratory of Brain Function and Disease, Guangzhou 510080 Guangdong, PR China. [3] Department of Neurosurgery, First Affiliated Hospital of Guangxi Medical University, Nanning 530000 Guangxi Province, PR China. [4] Department of Pathology, Fudan University Shanghai Cancer Center, Fudan University, Shanghai 200433, PR China. [5] Department of Spine Surgery, First Affiliated Hospital of Sun Yat-sen University, Guangzhou 510080 Guangdong, PR China. [6] Department of Gastroenterology Surgery, First Affiliated Hospital of Sun Yat-sen University, Guangzhou 510080 Guangdong, PR China. [7] Department of Scientific Research Section, First Affiliated Hospital of Sun Yat-sen University, Guangzhou 510080 Guangdong Province, PR China. [8] Guangzhou Gene Denovo Biotechnology Co. Ltd, Guangzhou 510006, PR China. [9] Department of Gastroenterology, University of Texas MD Anderson Cancer Center, Houston, TX 77030, USA. [10] Institute of Life and Health Engineering, College of Life Science and Technology, Jinan University, Guangzhou 510632, PR China. [11] Department of Neurosurgery, University of Texas MD Anderson Cancer Center, Houston, TX 77030, USA. [12] Program in Cancer Biology, University of Texas Graduate School of Biomedical Sciences at Houston, Houston, TX 77030, USA. These authors contributed equally: Maolei Zhang, Kun Zhao, Xiaoping Xu. Correspondence and requests for materials should be addressed to N.Z. (email: zhangnu2@mail.sysu.edu.cn)

As determined by deep RNA sequencing and bioinformatics, circRNAs are widely expressed RNA transcripts found in different species[1–6]. CircRNAs are inherently resistant to exonuclease activity (resulting in higher stability than linear RNAs) and often show tissue- or developmental stage-specific expression[2,7,8], implying that they possess important biological functions. To date, circRNAs have been shown to act as microRNA sponges[9–11], to respond to and regulate neuronal synaptic function[12], and to manipulate gene transcription in the nucleus[13]. A typical circRNA is generated through a back-splicing mechanism that covalently connects the 3′-end of a coding or non-coding exon to the 5′-end[14–16]. Until now, circRNAs have generally been considered non-coding RNAs (ncRNAs)[3,9,12]. However, recent evidence has demonstrated that functional peptides are encoded by short open reading frames (sORFs) in ncRNAs[17–22]. Additionally, certain synthetic circRNAs are translatable[23,24], raising the question of whether natural circRNAs containing sORFs in mammalian cells are translated into proteins or peptides. Up-to-date reports showed that endogenous circRNAs, such as *circZNF609*, *circMbl*, *circ-FBXW7*, and *circ-SHPRH*, are translated in vivo, suggesting that additional coding circRNAs and their translated products have yet to be discovered[25–29]. Currently, three broad strategies have been employed to identify sORFs in ncRNAs[19]. These strategies include cross-species comparisons to identify conserved sequences, the examination of codon content or features to differentiate potential coding sORFs, and translational approaches to identify coding sORFs[19]. Even with these strategies, the identification of sORFs in circRNAs is difficult. First, although cross-species conserved sORFs have been identified in circRNAs, there are few computational tools, such as circRNADb, that are available to predict coding circRNAs[30]. Second, as most circRNAs are generated from protein-coding exons, the sORFs in circRNAs may overlap with ORFs in related mRNA sequences[1,4]. In such cases, it is difficult to distinguish the origin of a translated product. Third, the best tool for exploring sORFs, ribosome profiling, has not been broadly applied to circRNAs due to technical difficulties such as library construction[3,31].

In this study, we show that human circRNAs generated from long ncRNAs that contain sORFs encode functional peptides. Through RNC-seq and bioinformatic analysis, we identify several previously uncharacterized peptides that are potentially encoded by circRNAs. Analysis of candidate peptides originating from previously designated ncRNAs confirms an 87-amino acid (aa) peptide, which we term PINT87aa, encoded by the circular form but not linear *LINC-PINT*, according to multiple translation-related lines of evidence. Functionally, PINT87aa, but not its corresponding circRNA, partially controls the cell proliferation and tumorigenesis of cancer cells. Upregulation of the expression of PINT87aa which is decreased in glioblastoma compared with their expression in normal tissues, induces tumor-suppressive effects in vitro and in vivo. Our study demonstrates that the biological and clinical implications of peptides translated from circRNAs are likely underestimated, and we provide some evidence that alternative splicing may produce functional peptides from previously designated non-coding genes.

## Results

### High-throughput sequencing of potential coding circRNAs.
To generate an RNA-seq database of circRNAs (transcriptome sequencing) and RNC-RNAs (translatome sequencing), we used ribosomal RNA-depleted total RNA and RNC-RNAs from normal human astrocytes (NHA) and U251 glioblastoma cells (Fig. 1a). The brief procedure of RNC extraction is listed in the "Methods" section and is based on the article by Wang et al.[32].

Total RNA and RNC-RNAs were sequenced separately on an Illumina HiSeq[TM] 4000. The obtained reads were mapped to reference ribosomal RNA (Bowtie2, http://bowtie-bio. sourceforge.net/bowtie2/) and a reference genome (http://ccb. jhu.edu/software/tophat/)[33,34], and 20mers from both ends of the unmapped reads were extracted and aligned to the reference genome to identify unique anchor positions within the splice site. Anchor reads that aligned in the reverse orientation indicated circRNA splicing was subjected to find_circ (https://omictools. com/find-circ-tool) to identify circRNAs[2]. The anchor alignments were then extended such that the complete read aligned, and the breakpoints were flanked by GU/AG splice sites. A candidate circRNA was called if it was supported by at least two unique back-spliced reads. As RNC-seq may have low identification rate, we designed to collect four times data in RNC-seq than that of RNA-seq. Through sequencing, a total of 15,189 circRNAs (7017 from RNA-seq and 12,863 from RNC-RNA seq) were identified, 4597 of which were matched in circBase[35] (Fig. 1b and Supplementary Table 1). We next analyzed the characteristics of these circRNAs. The majority of the identified circRNAs were generated from exon coding sequences (CDS), but many circRNAs were derived from antisense or intronic regions (Fig. 1c). Most of the identified circRNAs were less than 1500 nucleotide (nt) in length (Fig. 1d), and the chromosome distribution was not unified between NHA and U251 circRNAs as previously reported[36]. Quantitatively, 4879 circRNAs were identified by using RNA-seq and 9451 circRNAs were identified by RNC-seq in NHA. 4066 circRNAs were identified by using RNA-seq and 5992 circRNAs were identified by using RNC-seq in U251 (Fig. 1e, RNC-seq has a larger sequencing depth compared with RNA-seq, see Supplementary Fig. 1a and Supplementary Table 2 and Supplementary Data 1). Among these candidates, 4840 and 666 differentially expressed circRNAs were identified in NHA and U251, respectively, from total RNA or RNC-RNAs with a false discovery rate (FDR) of ≤0.01 and a fold-change ≥2 (Fig. 1f). We cross-matched these differentially expressed candidates from total RNA and RNC-RNAs, and 320 overlapped circRNAs were further identified (Fig. 1g, upper). Gene Ontology (GO) enrichment analysis was performed for the host genes of these circRNAs (274 genes), and this gene set was enriched (FDR < 0.05) for GO molecular functions including protein binding and cellular component organization (Fig. 1g, lower and Supplementary Table 3). As mentioned previously, most circRNAs shared the same CDS with their host protein-coding genes. To exclude false-positive data from further investigation, we focused only on non-coding host genes among these 274 candidates. A total of ten non-coding host genes were found and in which five had coding-potential (Supplementary Table 4). After initial screening, we focused on *LINC-PINT* (ENSG00000231721) for further investigation. *LINC-PINT* is a tumor-suppressive long intergenic non-coding RNA (lincRNA) that is involved in Polycomb repressive complex 2 (PRC2)[37]. No evidence thus far has suggested that *LINC-PINT* is a coding RNA[38,39].

### Identification of a circRNA formed by exon 2 of *LINC-PINT*.
We first visualized the back-spliced reads of exon 2 of *LINC-PINT* in our sequencing data. As shown in Fig. 2a, upper panel, there were 15 back-spliced junction-specific reads in the RNC-RNA group compared with 7 reads in the total RNA group, implying that exon 2 of *LINC-PINT* was identified as a translatable circular RNA. In contrast, junction reads were not identified in either total RNAs or RNC-RNAs from U251. The IGV plot showed that reads number of exon 2 *LINC-PINT* were higher in NHA compared with U251, both in RNA-seq and RNC-seq. Notably, exon 1 and exon 3 reads were much lower than exon 2 reads in

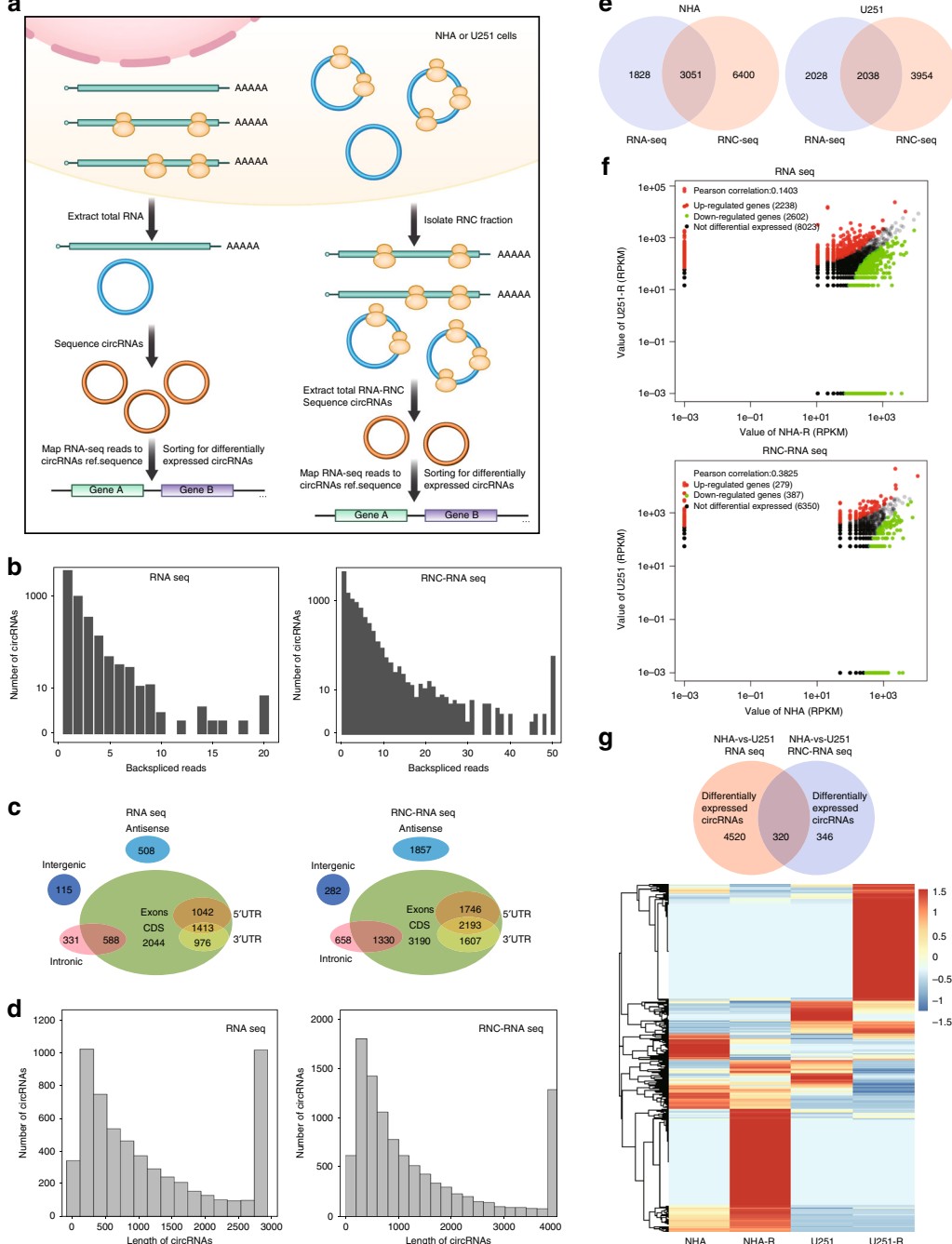

**Fig. 1** Translatome sequencing and proteome profiling of potential coding circRNAs in normal and cancer cells. **a** Illustration of the screening protocol. Briefly, total RNAs or RNC-RNAs were isolated separately from NHA or U251 cells. Equal amounts of total RNA or RNC-RNA were reverse-transcribed and subjected to deep RNA sequencing. Identified differentially expressed circRNAs were annotated in the genome, and the host genes were cross-matched between NHA and U251. **b** RNA-seq read abundance distribution of identified circRNAs. Upper, total RNA seq; Lower, RNC-RNA seq. X-axis: the back-spliced read numbers of circRNAs detected by sequencing. Y-axis: the abundance of circRNAs classified by different read numbers. The majority of called circRNAs in the study were supported by more than 10 reads. **c** Venn plot showing the number of circRNAs derived from different genomic regions. Upper, total RNA seq; lower, RNC-RNA seq. **d** Length distribution of the identified circRNAs. Upper, total RNA seq; lower, RNC-RNA seq. X-axis: the length of circRNAs detected in this study. Y-axis: the abundance of circRNAs classified by different lengths. **e** The Venn plot of the numbers of called circRNAs in NHA and U251 by RNA-seq or RNC-seq. RNC-seq identified more circRNAs due to the higher sequencing depth (see Supplementary Figure 1). **f** Scatter plot of all differentially expressed circRNAs between NHA and U251 cells. Upper, total RNA-seq; lower, RNC-seq (x and y axes represent circRNA expression value, RPKM). **g** Upper, differentially expressed circRNAs between NHA and U251 cells in total RNA or RNC-RNA were cross-matched. A total of 320 differentially expressed circRNAs were identified, generated from 274 host genes. Lower, the host genes were subjected to GO enrichment analysis (The gene expression value in heatmap was normalized by Z score in each row.)

RNC-seq, implied that linear *LINC-PINT* is not translated (Fig. 2a, lower panel). The long exon 2 of *LINC-PINT*, which contains the 3′ AG receptor and 5′ GT donor sequences required for back-splicing, was identified as a circRNA in human CircBase (*has circ-0082389*, Fig. 2b, upper panel, and Supplementary Fig. 1b). Therefore, we first determined whether exon 2 of *LINC-PINT* formed an endogenous circRNA in human cells. Head-to-tail splicing was assayed by performing quantitative polymerase

chain reaction (q-PCR) after reverse transcription with con/ divergent primers specific for the linear or circular form of *LINC-PINT* (Fig. 2b, lower panel). The PCR products from divergent primers were analyzed via Sanger sequencing to reveal the junction of circular *LINC-PINT* exon 2 (Fig. 2c). To exclude the possibility that this back-splicing was attributable to genomic rearrangement or was a PCR artifact, we validated this circRNA through northern blotting with an exon probe or a circular probe,

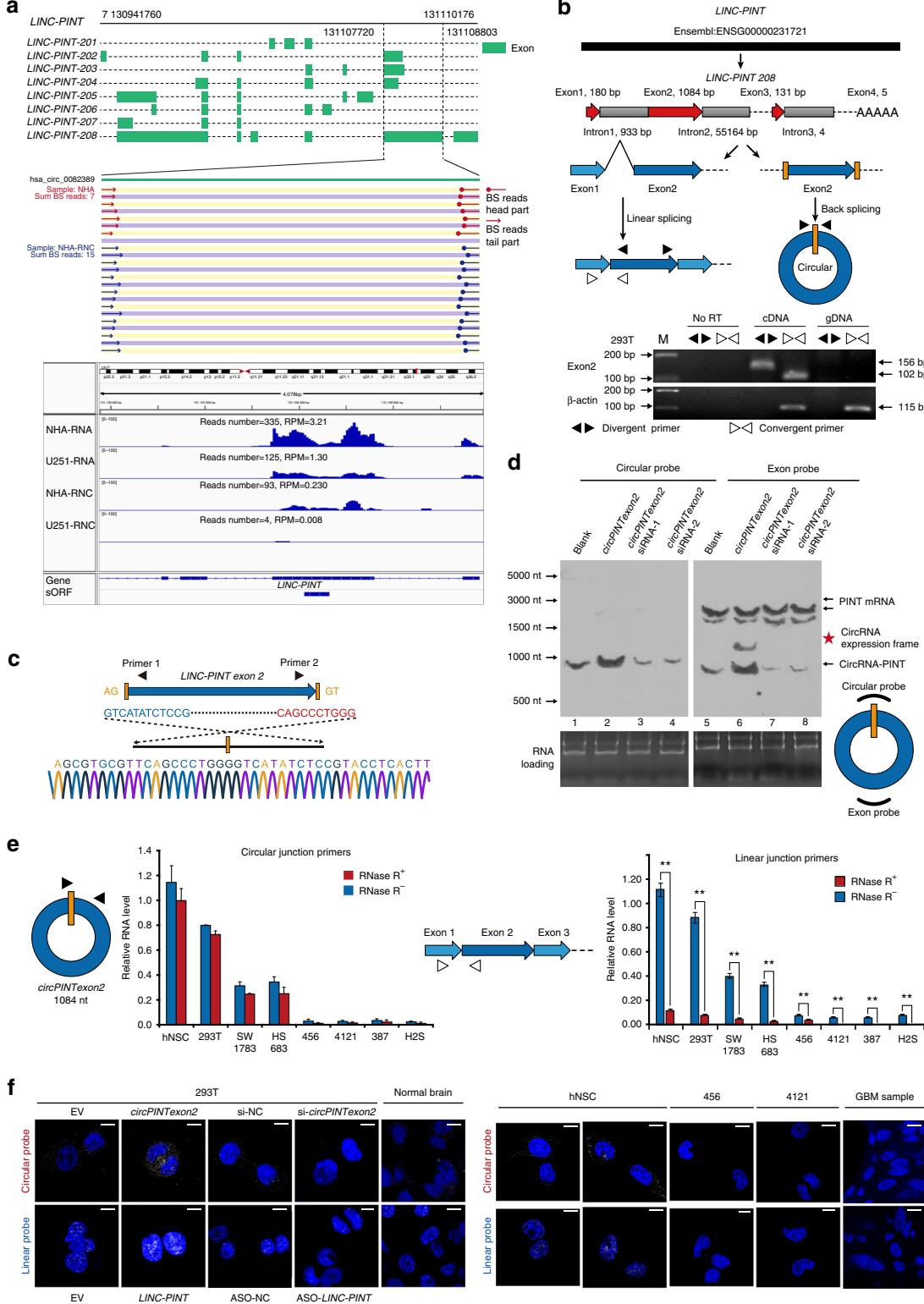

which recognize both the linear and circular forms or only the circular forms of *LINC-PINT* exon 2 (which we designated *circPINTexon2*), respectively. *CircPINTexon2* was detected endogenously in 293T cells, while upregulated or decreased accordingly after synthetic *circPINTexon2* plasmid overexpression or junction siRNA transfection (Fig. 2d, lanes 1–4, schematic diagram of the overexpression plasmid shown in Fig. 3f). Furthermore, exon probes detected both *LINC-PINT* and *circPINTexon2*, as shown in Fig. 2d, lanes 5–8. Q-PCR and fluorescence in situ hybridization (FISH) analyses with primers/ probes specifically designed to detect the circular junction or linear junction further confirmed the existence of the head-to-tail spliced circular form of *LINC-PINT* exon 2 in human cell lines and tissues. Both linear *LINC-PINT* and *circPINTexon2* were expressed in human neural stem cells (hNSC) and 293T cells, but their expression decreased in different glioma/brain tumor-initiating cells (BTIC) cell lines (Fig. 2e). *LINC-PINT* and *circPINTexon2* presented different cellular localizations: linear *LINC-PINT* largely localized to the nucleus, whereas *circPINTexon2* was mostly cytoplasmic (Fig. 2f and Supplementary Fig. 2).

**CircPINTexon2 encodes a peptide**. To test the hypothesis that *circPINTexon2* is translatable (*circPINTexon2* junction reads were identified in RNC-seq, and exon 2 reads on *LINC-PINT* were much higher than exon 1 and exon 3), we first analyzed all putative sORFs in *LINC-PINT* exon 2. We identified two sORFs, potentially encoding peptides 69 aa and 87 aa in length, in this exon that were highly conserved among multiple species (Supplementary Fig. 3). To explore whether these putative ORFs were active, we cloned full-length exon 2 of *LINC-PINT* into the PLCDH-ciR vector (with artificial side flanking sequences and SA/SD sequences), containing green fluorescent protein (GFP), without start or stop codons, fused immediately upstream of the stop codons of the two sORFs. Immunofluorescence (IF) results showed that only the second sORF, which encoded the 87-aa peptide, was expressed in 293T cells with the overexpression of the above vector (Supplementary Fig. 4). Although the 87-aa sORF was an active in-frame sORF in vivo compared with the 69-aa sORF, further investigations were needed to show that the 87-aa peptide was translated by endogenous *circPINTexon2* but not linear *LINC-PINT* in vivo.

First, to demonstrate that *circPINTexon2* contains a natural internal ribosome entry site (IRES), which is required for translation initiation in 5′-cap-independent coding RNAs[40–42], we carried out a bioinformatics analysis (Supplementary Fig. 5) and an IRES activity test[43], as shown in Fig. 3a. mCherry and GFP were cloned

into a dual-cistronic reporter construct with or without the putative IRES (478 bp upstream of the 87-aa sORF) or with the indicated mutations between them (Fig. 3a, left). Under normal eIF4E conditions, both mCherry and GFP were detected with the putative IRES; in contrast, only mCherry was detected if the putative IRES was deleted. GFP was not detected when the sequence −478 to −231 upstream of the 87-aa sORF was deleted, indicating the IRES is located in this area. When eIF4E was inhibited (under treatment with 4EGI-1), the putative IRES induced the expression of GFP but not mCherry; thus, the natural IRES in circPINTexon2 induced ribosome entry and initiated translation. To quantitatively test the putative activity of *circPINTexon2* IRES, we performed further experiments. Briefly, we used a tandem Rluc–Luc reporter plasmid in which the Rluc ORF was driven by a CMV promoter, and the Luc ORF was fused immediately after the RLuc stop codon, without any promoter between them. We cloned the IRES-478bp, IRES-209bp and IRES-231bp sequences between RLuc and Luc, and the luciferase activity of Luc relative to that of RLuc was measured for each plasmid. IRES-209bp presented the lowest luciferase activity (Luc/Rluc), close to that of the empty vector, indicating that IRES-209bp does not induce ribosome entry. IRES-478bp and IRES-231bp exhibited significantly higher Luc/Rluc activities than the empty vector, further supporting the notion that an active IRES is located upstream of the 87-aa sORF (Fig. 3b).

Next, we performed qPCR on the translating RNAs (RNC-RNAs) from 293T cells to further show that PIN87aa is encoded by *circPINTexon2* but not linear *LINC-PINT*. RNC-RNAs were reverse-transcribed using oligo-dT (for mRNAs) or random primers (for all RNAs). Total RNAs were reverse-transcribed by random primers as a positive control. As shown in Fig. 3c, both *circPINTexon2* and linear *LINC-PINT*-specific PCR products were amplified from cDNA reverse-transcribed from total RNAs with random primers, indicating that *circPINTexon2* and linear *LINC-PINT* are both present in 293T cells. However, only *circPINTexon2*-specific PCR products were amplified from cDNA reverse-transcribed from RNC-RNAs with random primers, indicating that *circPINTexon2* is bound to the ribosomal nascent chain complex and is undergoing translation. In contrast, linear *LINC-PINT*-specific PCR products were not detected in cDNA reverse-transcribed from RNC-RNAs with random primers or oligo-dT primers, indicating that linear *LINC-PINT* does not undergo translation. These data were consistent with our RNC-seq data described above (Fig. 2a, RNC-seq showed few reads on exon 1 and 3 of *LINC-PINT*, indicating that linear *LINC-PINT* was not translated).

Based on the above information, we generated an antibody against the 87-aa peptide (antibody construction is described in the Supplementary Data 2). Immunoblotting showed that this

**Fig. 2** Identification of exon 2 of *LINC-PINT* as a circRNA. **a** Upper, visualization of the forward reads within the exon 2 region in the *LINC-PINT* junction site of NHA cell in RNA-seq and RNC-seq. These junction reads are specific for circular form of *LINC-PINT* exon 2. Lower, IGV plot of all reads located on exon 2 of *LINC-PINT* in RNA-seq and RNC-seq. The IGV plot also included the reads on exon 1 and 3 of *LINC-PINT*. **b** Illustration of the annotated genomic region of *LINC-PINT* (Ensembl number: ENSG00000231721), the putative different mRNA splicing forms (linear splicing and head-to-tail splicing) and the validation strategy for *LINC-PINT* circular exon 2 (*circPINTexon2*). Divergent primers detected the circular form of *circPINTexon2* in cDNA but not in gDNA. Convergent primers spanning exon 1 and exon 2 of *LINC-PINT* (variants *LINC-PINT*-208, shown in **a**) specifically detected the linear splicing form. *β-actin* was used as a linear RNA control. **c** Sanger sequencing was performed following PCR using the indicated divergent flanking primers to confirm the head-to-tail splicing of *circPINTexon2* in 293T cells. **d** Northern blots of 293T total RNA with the exon probe and the junction-specific circular probe for *circPINTexon2*. Lanes 1–4 detected *circPINTexon2* with circular probes. Lanes 5–8 detected *circPINTexon2* and *LINC-PINT* with exon probes. *CircPINTexon2*-overexpression plasmid was shown in Fig. 3f. **e** Q-PCR followed by with junction-specific primers was used to detect the expression of *circPINTexon2* in vitro. Primers specific for linear *LINC-PINT* were also used to detect *LINC-PINT* expression. RNase R treatment was used to validate *circPINTexon2*. Data are presented as mean ± s.e.m. from three independent experiments. \*\*P < 0.01; ns, P > 0.05, determined by two-tailed Student's *t*-tests. **f** FISH with junction-specific probes specific to *circPINTexon2*, whereas linear specific probes specific to linear *LINC-PINT* indicated their cellular localization in vitro. Normal brain tissues and GBM samples were stained with indicated probes. Overexpressed or knocked-down *circPINTexon2* or *LINC-PINT* using corresponding plasmids or siRNA/ASOs in 293T cells to indicate the specificity of these probes. Scale bars, 10 μM. EV empty vector, si-NC random scrambled siRNA, *circPINTexon2* circPINTexon2 overexpression vector, ASO *LINC-PINT* anti-sense oligos

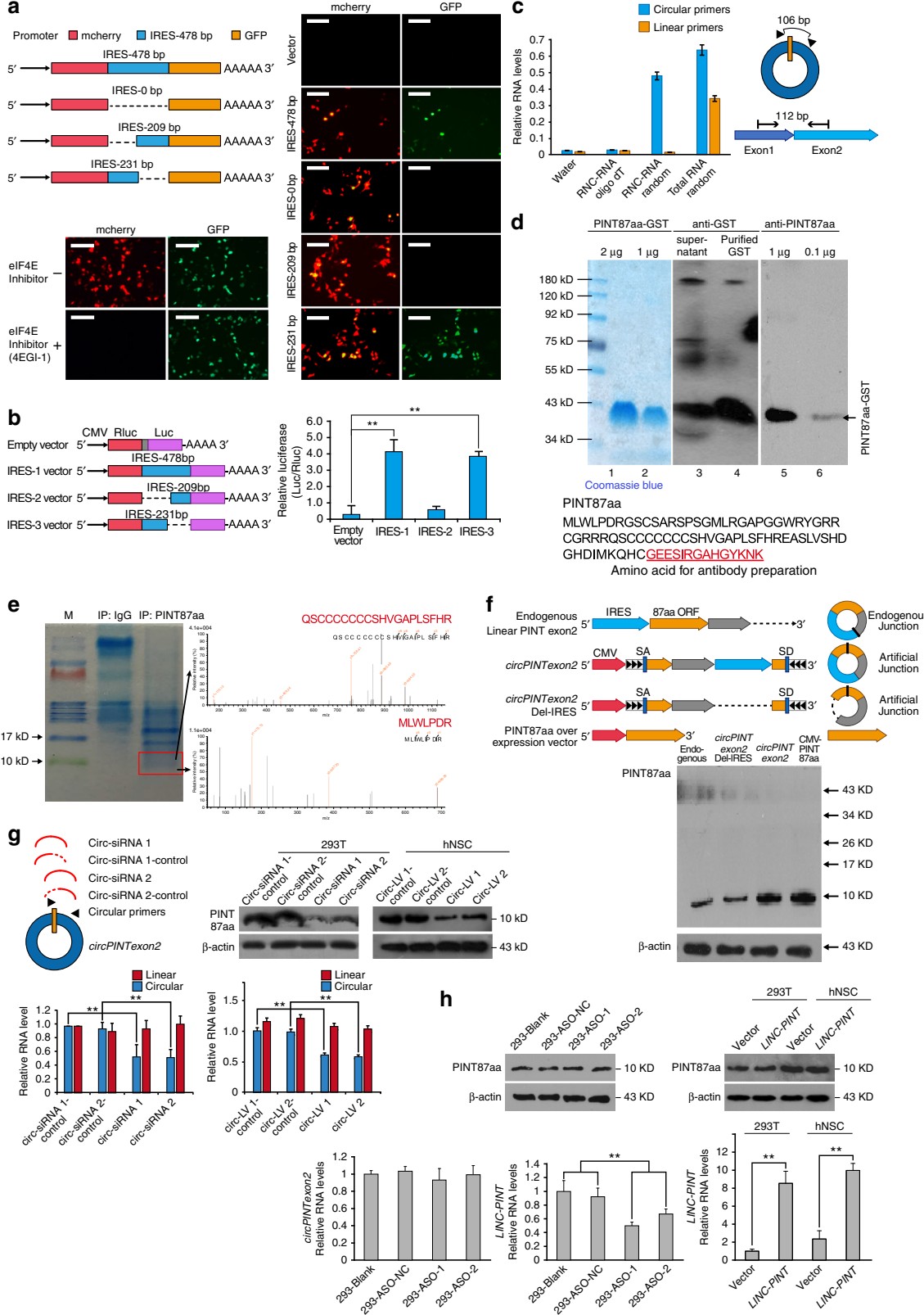

antibody successfully recognized the predicted glutathione-S-transferase (GST)-fused protein (Fig. 3d). Endogenous immuno-precipitation using this antibody followed by LC-MS/MS in 293T cells further confirmed that the 10-kDa peptide sequences matched the predicted 87-aa sORF (Fig. 3e). Because the 87-aa sORF did not overlap the *circPINTexon2* junction, we designed

several synthetic plasmids to further test the coding ability of this circRNA (Fig. 3f, upper panel). In the *circPINTexon2* vector, the circRNA junction was moved inside the 87-aa sORF, and circularization of this plasmid induced by side flanking sequences resulted in the formation of the same circRNA as the natural *circPINTexon2*[15]. In contrast, the 87-aa sORF was not present in

**Fig. 3** circPINTexon2 encodes an 87-aa peptide. **a** Full-length or truncated *circPINTexon2* IRES (478, 231, and 209 bp) were cloned between mCherry and GFP as indicated to construct several reporter plasmids. These plasmids were transfected into 293T cells as indicated, with or without 4EGI-1 treatment. IF was performed to determine mCherry and GFP signals. Scale bars, 50 μM. **b** Rluc and Luc were tandemly cloned into the luciferase reporter plasmid, with or without the indicated truncated IRES between them. Luc/Rluc activities were measured in each transfected plasmid. **c** RNC-RNA or total RNA from 293T cells was extracted and reverse-transcribed using oligo-dT or random primers as indicated. Specific primers for *circPINTexon2* or *LINC-PINT* were analyzed by using q-PCR. **d** Upper, antibody recognition test for the predicted 87-aa peptide. Lanes 1 and 2, Coomassie blue staining of the GST-PINT87aa fusion protein; lanes 3 and 4, Western blot performed with GST antibody; lanes 5 and 6, Western blot performed with PINT87aa antibody. Lower, the predicted 87-aa peptide sequence and antibody generation region was shown as indicated. **e** Endogenous immunoprecipitation using anti-PINT87aa antibodies in 293T cells. LC-MS/MS analysis following SDS-PAGE was performed to identified peptide sequences of PINT87aa. **f** Upper panel, endogenous *circPINTexon2*: endogenous formation of *circPINTexon2*; *CircPINTexon2* vector: the artificial *circPINTexon2* overexpression plasmid. Note the junction was moved inside the 87-aa ORF. CicrPINTexon2 Del-IRES vector: negative control, in which the IRES sequence was deleted from the artificial *circPINTexon2* plasmid. 87-aa overexpression vector: positive control, in which the 87-aa ORF was cloned downstream of a linear CMV promoter. Lower panel, PINT87aa expression was tested in 293T cells after transfection of the plasmids indicated above. **g** *circPINTexon2* and PIN87aa expression were determined using junction-specific siRNA or shRNA specific for *circPINTexon2*-transfected 293T or hNSC. **h** Left, *LINC-PINT*, *circPINTexon2*, and PIN87aa expression were determined in two ASOs specific for *LINC-PINT*-transfected 293T. Right, *LINC-PINT* and PINT87aa were determined in *LINC-PINT* stably transduced 293T and hNSC. **b**, **c**, **g**, **h** Data are presented as mean ± s.e.m. from three independent experiments. **P < 0.01; ns, P > 0.05, determined by two-tailed Student's *t*-tests

the *circPINTexon2* vector's linear reading frame. As shown in Figs. 2c and 3f, lower panel, northern blotting and western blotting indicated that transfection of this synthetic plasmid resulted in successful overexpression of *circPINTexon2* and the 87-aa peptide. However, the transfection of control plasmids in which the IRES was deleted did not result in overexpression of the 87-aa peptide (Supplementary Fig. 6). Importantly, 87-aa peptide overexpression induced by this synthetic circRNA plasmid was as strong as that of the CMV-driven linear 87-aa sORF overexpression plasmid, demonstrating the high translation efficiency of circRNA[44] (Fig. 3f, lower panel). We next used two siRNAs that specifically target the circular junction of *circPINTexon2*. Junction-specific siRNAs successfully reduced *circPINTexon2* and 87-aa peptide levels without affecting linear *LINC-PINT* (Fig. 3g). In contrast, two antisense oligonucleotides (ASOs) specifically designed to target linear *LINC-PINT* did not decrease the expression of *circPINTexon2* or the 87-aa peptide (Fig. 3h, left). Furthermore, stable overexpression of linear *LINC-PINT* in 293T/hNSC did not elevate PINT87aa expression (Fig. 3h, right). Collectively, these results demonstrated that the 87-aa peptide is produced by *circPINTexon2* but not the linear form of *LINC-PINT*. We named this peptide PINT87aa.

**Tumor-suppressive effects of PINT87aa in vitro**. To investigate its possible biological functions, we first detected the cellular localization of PINT87aa. Using RFP fusion protein labeling, we found that this peptide was concentrated in nucleus, suggesting PINT87aa plays potential cellular regulatory roles (Fig. 4a). We next detected *circPINTexon2* and PINT87aa baseline expression in several human tissues. *CircPINTexon2* and PINT87aa were expressed in the brain, liver, kidney, and stomach but showed lower expression in the breast, intestine, thyroid, and pancreas tissues (Fig. 4b). Because PINT87aa were abundant in the human brain, we further detected its expression in several established glioma and BTIC cell lines. hNSC exhibited high PINT87aa expression, similar to 293T cells, whereas the anaplastic astrocytoma cell lines SW1783 and Hs683 exhibited the modest expression. The BTICs demonstrated the lowest levels of PINT87aa (Fig. 4c, left). The *circPINTexon2* expression in these cell lines detected using junction-specific primers was consistent with PINT87aa levels (Fig. 4c, right). Decreased *circPINTexon2* and PINT87aa expression also reflected WHO grades in clinical glioma samples (Fig. 4c, lower). To determine the biological roles of PINT87aa in tumor cells, we established 456 and 4121 cells that stably overexpressed the linear PINT87aa-GFP vector or the circular *circPINTexon2* vector (Fig. 4d). Additionally, we

generated CRISPR/Cas9-induced PINT87aa K.O. SW1783 and Hs683 cells (Fig. 4e and Supplementary Fig. 7). These cells were selected because 456 and 4121 exhibited very low PINT87aa expression, while SW1783 and Hs683 cells exhibited moderate PINT87aa expression. Compared with corresponding control cells, both 456 and 4121 cells overexpressing linear PINT87aa and *circPINTexon2* exhibited G1 arrest and reduced cell proliferation without obvious cellular toxicity (Fig. 5a, b, left), whereas PINT87aa K.O. SW1783 and Hs683 cells showed increased cell cycle and cell proliferation rates (Fig. 5a, b, right). 456-PINT87aa and 4121-PINT87aa cells overexpressing both linear and circular vectors showed significant growth reduction (Fig. 5c, left), while PINT87aa K.O. SW1783 and Hs683 cells showed an enhanced cell proliferation and malignant phenotype in plate colony and soft agar (Fig. 5c, right; 5d). As expectation, 456-PINT87aa and 4121-PINT87aa cells overexpressing both linear and circular vectors showed less effective neuro-sphere formation (Fig. 5e). PINT87aa overexpression also enhanced the IR sensitivity in both 456 and 4121 cells (Fig. 5f). We also observed a slight inhibition of invasion ability in PINT87aa transduced 456 and 4121 cells, although this phenomenon may be induced by an accelerated cell proliferation (Supplementary Fig. 8a). In contrast, PINT87aa K.O. SW1783 and Hs683 cells showed radiation resistance compared with their parental cells (Supplementary Fig. 8b). Based on the above results, we showed that *circPINTexon2* exerts its tumor-suppressive functions through PINT87aa instead of *circPINTexon2*.

**PINT87aa regulates the RNA elongation of multiple oncogenes**. To further explore the potential molecular mechanisms underlying the tumor-suppressive functions of PINT87aa, we first checked some proto-oncogenes that are critical in glioma tumorigenesis in PINT87aa overexpressed BTICs. However, EGFR, MET, and PDGFR were not changed after PINT87aa or *circPINTexon2* overexpression (Supplementary Fig. 8c). Next, we performed co-immunoprecipitation in PINT87aa-3XFlag-overexpressing 293T cells to identify its potential targeting molecules. As shown in Fig. 6a, left, 293T cells transfected with PINT87aa-3XFlag and corresponding control cells were subjected to immunoprecipitation using an anti-Flag antibody. The precipitates were subjected to LC-MS/MS to identify potential PINT87aa-interacting proteins. Among various candidates, PINT87aa was found potentially bound to the PAF1 protein complex (Fig. 6a, right). Immunoprecipitation further confirmed a mutual PINT87aa/PAF1 interaction in 293T cells (Fig. 6b). To investigate the potential direct interaction between PINT87aa

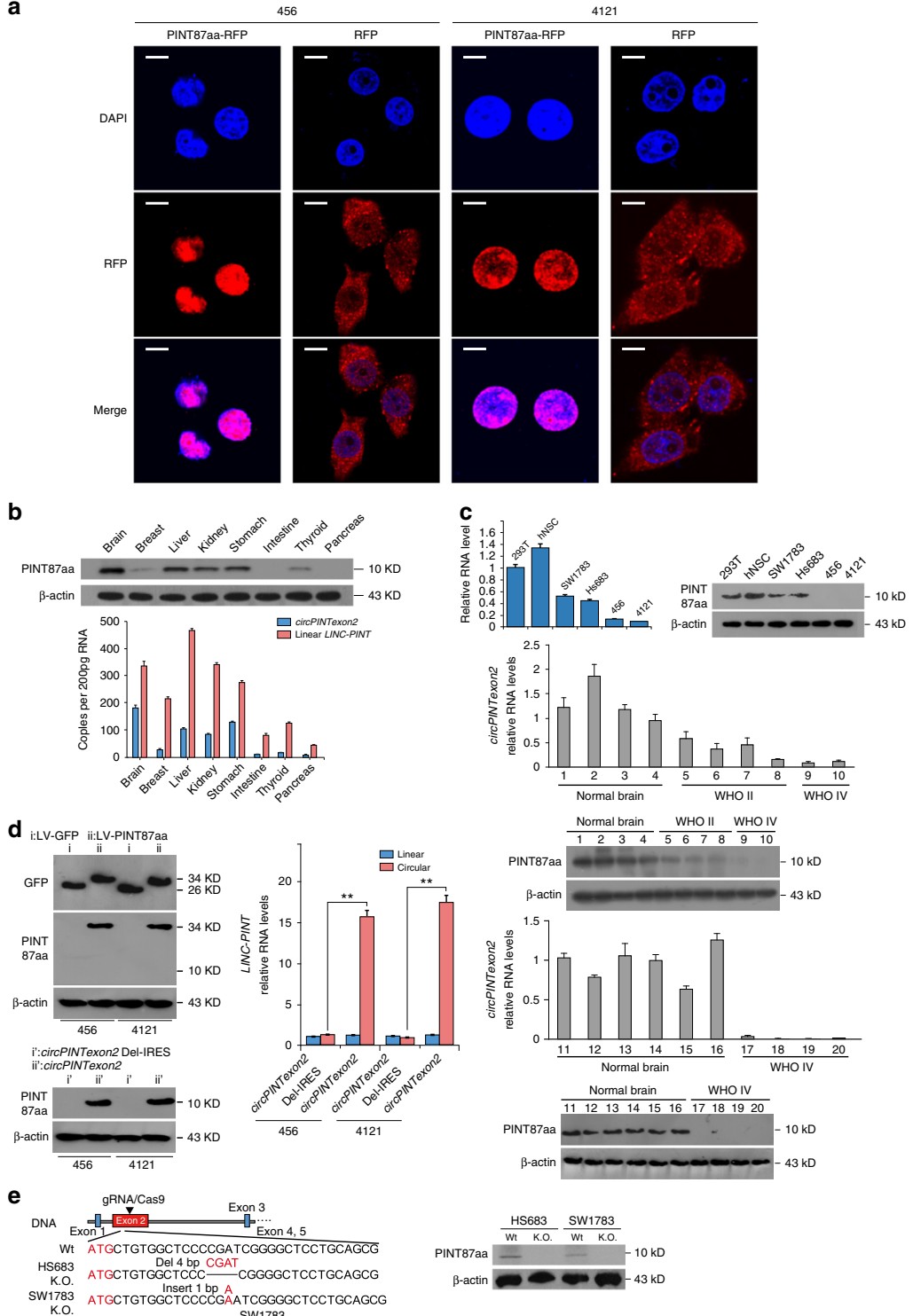

**Fig. 4** Localization and expression of PINT87aa in cells and human tissues. **a** IF microscopy images of the cellular localization of the PINT87aa-RFP-fusion protein in 456 and 4121 BTIC cells showing stable overexpression. Scale bars, 20 μM. **b** The expression of *circPINTexon2* and PINT87aa was detected in human brain, breast, liver, kidney, stomach, intestine, thyroid, and pancreas tissues. **c** Upper panel, the expression of *circPINTexon2* and PINT87aa was detected in hNSC, 293T, Hs683, SW1783, 4121, 456, 387, and H2S cells. Lower panel, *circPINTexon2* and PINT87aa levels were determined in normal brain and gliomas with different WHO grades. **d** Establishment of linear vector-transduced stable PINT87aa-GFP 456 and 4121 cells and artificial *circPINTexon2*-transduced stable PINT87aa 456 and 4121 cells. Data are presented as mean ± s.e.m. from three independent experiments. **P < 0.01; ns, P > 0.05, determined by two-tailed Student's t-tests. **e** Establishment of PINT87aa K.O. SW1783 and Hs683 cells using CRISPR/Cas9 technology. Schematic illustration showing that the genomic regions of the *LINC-PINT* exon2 contained the 87-aa ORF. gRNAs designed to target the PINT87aa ORF are shown. Genomic K.O. effects were confirmed via Sanger sequencing, as shown in Supplementary Fig. 7. Western blotting revealed the effects of PINT87aa K.O. in SW1783 and Hs683 cells

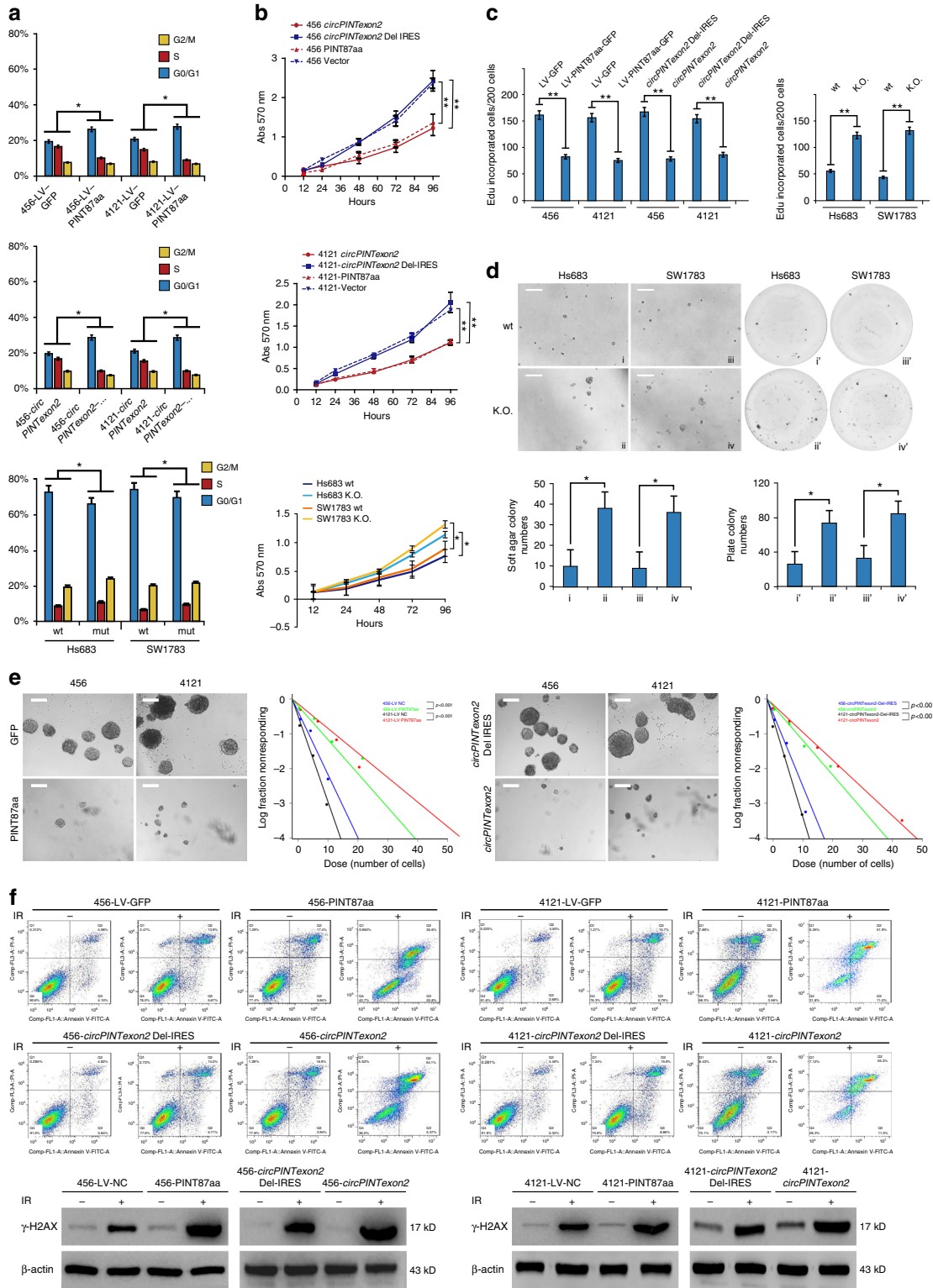

**Fig. 5** Biological functions of PINT87aa. **a** Left, cell cycle was determined in PINT-87aa or *circPINTexon2* stably overexpressed 456 and 4121 BTICs. Right, cell cycle analysis of PINT87aa K.O SW1783 and Hs683 cells. **b** MTT proliferation assay was examined in PINT87aa-GFP or *circPINTexon2* stably overexpressed 456 and 4121 cells or PINT87aa K.O SW1783 and Hs683 cells. **c** Edu proliferation assay was examined in PINT87aa-GFP or *circPINTexon2* stably overexpressed 456 and 4121 cells, PINT87aa K.O SW1783 and Hs683 cells and their control cells. **d** Soft-agar and plate colony assay was performed in PINT87aa K.O SW1783 and Hs683 cells and their control cells. Scale bar, 100 μM. **e** In vitro extreme limiting dilution assays (ELDAs) were performed to evaluate BTICs self-renewal capacity. Scale bar, 100 μM. Spheres were counted at 14 days. Cell density per well ranged from 1, 10, 25, 50, 100, 250, 500 to 1000. Each condition was tested in 10 independent wells. Neurosphere-forming capability was determined using the ELDA web-based tool (456, $P < 0.001$; 4121, $P < 0.001$, by ELDA analysis). **f** PINT-87aa or *circPINTexon2* stably overexpressed 456 and 4121 BTICs and their control cells were subjected to 6 Gy radiation. DNA damage was determined by flow cytometry and γ-H2AX expression. **a–d**, **f** Data are presented as mean ± s.e.m. from three independent experiments. *$P < 0.05$; ns, $P > 0.05$, determined by two-tailed Student's *t*-tests

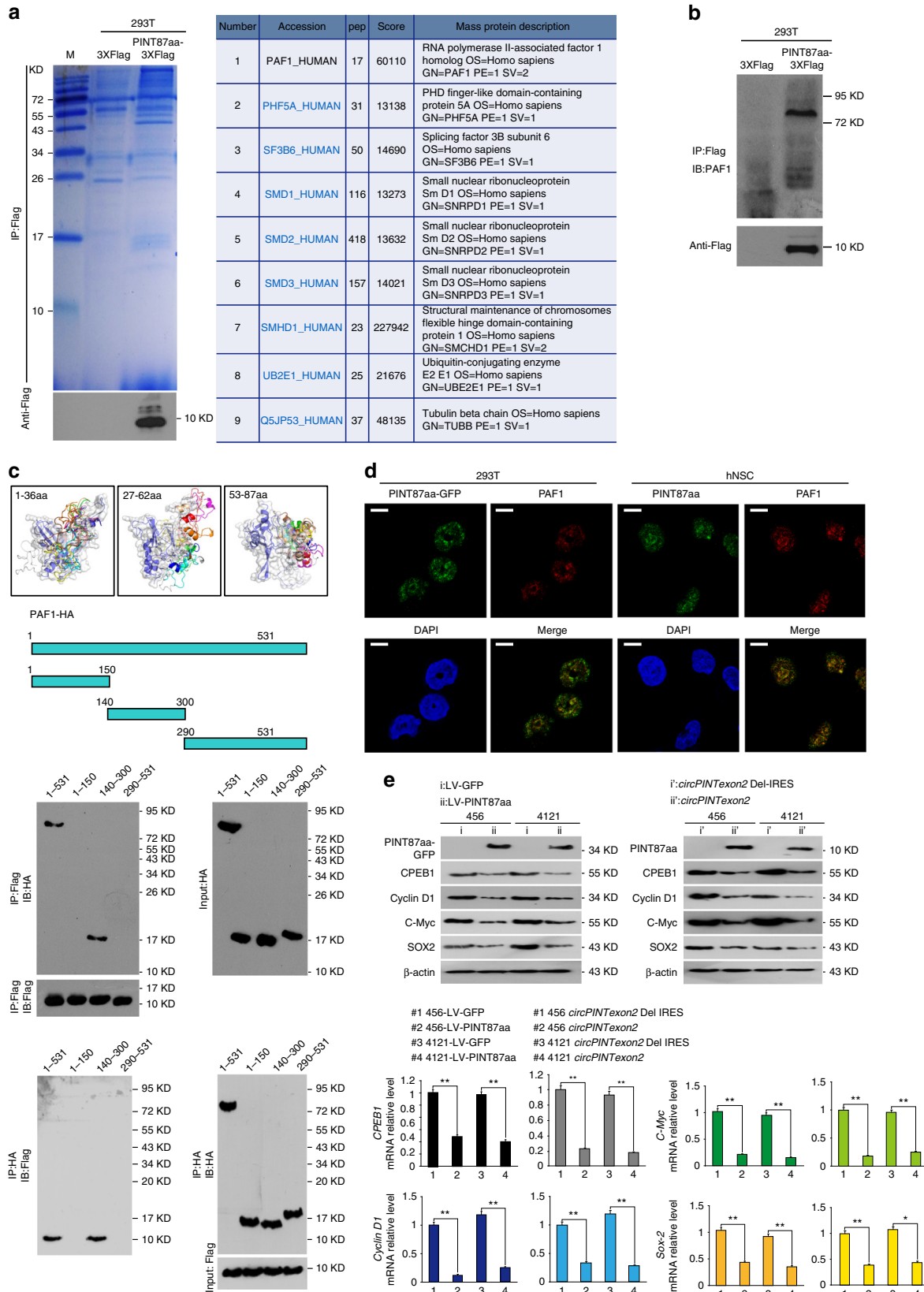

**a**

| Number | Accession | pep | Score | Mass protein description |
|---|---|---|---|---|
| 1 | PAF1_HUMAN | 17 | 60110 | RNA polymerase II-associated factor 1 homolog OS=Homo sapiens GN=PAF1 PE=1 SV=2 |
| 2 | PHF5A_HUMAN | 31 | 13138 | PHD finger-like domain-containing protein 5A OS=Homo sapiens GN=PHF5A PE=1 SV=1 |
| 3 | SF3B6_HUMAN | 50 | 14690 | Splicing factor 3B subunit 6 OS=Homo sapiens GN=SF3B6 PE=1 SV=1 |
| 4 | SMD1_HUMAN | 116 | 13273 | Small nuclear ribonucleoprotein Sm D1 OS=Homo sapiens GN=SNRPD1 PE=1 SV=1 |
| 5 | SMD2_HUMAN | 418 | 13632 | Small nuclear ribonucleoprotein Sm D2 OS=Homo sapiens GN=SNRPD2 PE=1 SV=1 |
| 6 | SMD3_HUMAN | 157 | 14021 | Small nuclear ribonucleoprotein Sm D3 OS=Homo sapiens GN=SNRPD3 PE=1 SV=1 |
| 7 | SMHD1_HUMAN | 23 | 227942 | Structural maintenance of chromosomes flexible hinge domain-containing protein 1 OS=Homo sapiens GN=SMCHD1 PE=1 SV=2 |
| 8 | UB2E1_HUMAN | 25 | 21676 | Ubiquitin-conjugating enzyme E2 E1 OS=Homo sapiens GN=UBE2E1 PE=1 SV=1 |
| 9 | Q5JP53_HUMAN | 37 | 48135 | Tubulin beta chain OS=Homo sapiens GN=TUBB PE=1 SV=1 |

and PAF1, PINT87aa conformations were modeled with PEP-FOLD and docked to PAF1 based on the ATTRACT2 force field through the PEP-SiteFinder pipeline[45]. Because PINT87aa was too long, it was split into three sections: 1–36 aa, 27–62 aa, and

53–87 aa. The top ten peptides in complex with PAF1 were visualized with different colors by PyMOL (The PyMOL Molecular Graphics System, Version 1.8 Schrödinger, LLC.), while protein–peptide interface residues (within 5 Å for each peptide)

**Fig. 6** PINT87aa directly interacts with the PAF1 complex and inhibits mRNA transcriptional elongation. **a** Left, immunoprecipitation was performed using an anti-Flag antibody in PINT87aa-3XFlag or empty vector transfected 293T cells. Western blotting using an anti-Flag antibody confirmed PINT87aa overexpression. The precipitates were subjected to LC-MS/MS to identify potential PINT87aa-interacting proteins. Right, PAF1 complex-related proteins were identified in PINT87aa precipitates. **b** Immunoprecipitation was performed in PINT87aa-3XFlag-transfected cells or control cells. Western blotting was performed using an anti-PAF1 antibody. **c** Upper, PINT87aa conformations were modeled with PEP-FOLD and docked to PAF1 based on the ATTRACT2 force field using the PEP-SiteFinder pipeline. PINT87aa was split into three segments: 1–36 aa, 27–62 aa, and 53–87 aa. The top ten peptides in complex with PAF1 were visualized with different colors using PyMOL (The PyMOL Molecular Graphics System, Version 1.8 Schrödinger, LLC.), while the protein–peptide interface residues (within 5 Å for each peptide) were labeled. Lower, the direct mutual interactions of PINT87aa with different domains of HA-tagged PAF1 were tested using purified proteins (Flag-tagged PINT87aa and HA-tagged PAF1). **d** IF was performed to determine PINT87aa-PAF1 colocalization in PINT87aa-GFP transfected 293T cells or hNSC. Scale bar, 20 μM. **e** The expression of PAF1 downstream genes was determined by performing Western blotting and q-PCR in PINT87aa- or *circPINTexon2*-overexpressed 456 and 4121 BTIC and their respective controls. Data are resented as mean ± s.e.m. from three independent experiments. **P < 0.01, ns, P > 0.05, determined by two-tailed Student's *t*-tests

were labeled. As shown in Fig. 6c, upper left, PINT87aa most likely directly interacts with the middle region of PAF1. A direct binding assay using purified proteins further indicated that PINT87aa interacts with the 150–300-aa domain of PAF1 (Fig. 6d, lower panel), and the amino acid of R20, G21, P23, C32, R36, and S53 in PINT87aa are critical for PAF1 interaction (Supplementary Fig. 9). Furthermore, GFP-PINT87aa and PAF1 co-localized in the nucleus (Fig. 6e), suggesting PINT87aa may be involved in PAF1 target gene regulation. The PAF1 complex is involved in RNA II polymerase (Pol II) recruitment and regulating the transcriptional elongation of downstream genes[46,47], and evidence has shown that PAF1 regulates potential oncogenes during human tumorigenesis, including that of gliomas[48–50]. We performed further experiments in PINT87aa-overexpressing 456 and 4121 cells to determine whether PINT87aa is involved in PAF1 target gene elongation. The mRNA of *PAF1* downstream genes, including *CPEB1*, *SOX-2*, *c-Myc*, etc., was inhibited transcriptionally in PIN87aa-overexpressing 456 and 4121 cells compared with that in control cells (Fig. 6e). In PINT87aa K.O. Hs683 and SW1783 cells, the expression of these genes increased at both the mRNA and protein levels (Supplementary Fig. 10a). Further 2nd ChIP assay showed that PINT87aa and PAF1 were co-occupied in *CPEB1* promoter (Fig. 7a), suggesting PINT87aa involved in PAF1/Pol II complex regulation. However, PINT87aa overexpression did not change PAF1 expression in 456 or 4121 BTICs. Instead, PINT87aa enhanced PAF1 and its target genes' promoter interaction (Fig. 7b). PINT87aa overexpression or knocking down enhanced or decreased PAF1/*CPEB1* promoter affinity, respectively (Fig. 7c), suggesting that PINT87aa could decide PAF1/*CPEB1* promoter interaction. We assumed that PINT87aa may work as an anchor and keep PAF1 complex on target genes' promoter, which sequentially pauses Pol II-induced mRNA elongation. Loss of PINT87aa or overexpression of PAF1, which was seen in many cancers, results in PAF1 losing its proper localization[51]. Freed PAF1 is sequentially involved in many other biological processes such as cell cycle regulation, histone modification, MAPK signaling transduction as well as cancer-stem-cell self-renewal[51]. *LINC-PINT* knockdown in SW1783 and Hs683 cells using specific ASOs increased cell viability but did not alter PAF1 downstream gene mRNA or protein levels (Supplementary Fig. 10b and c), indicating *LINC-PINT* and PINT87aa are involved in different signaling pathways. Interestingly, stable knockdown of PINT87aa in normal cells, such as NHA cells, also decreased cell vitality (Supplementary Fig. 11). PINT87aa appears to be required for normal cell survival, but its loss in cancer cells induces cell cycle acceleration and cell proliferation. *LINC-PINT* was reported to be regulated by *p53*[37]. We did not find PINT87aa altering *p53* expression or affecting *p53* downstream targets, in both *p53* wild type or mutant cells (Supplementary Fig. 12a, b and c). In contrast, overexpression of *p53* in A172 and 293T cells could

upregulate PINT87aa (Supplementary Fig. 12d). We inferred that PINT87aa exerts biological functions independent of *LINC-PINT* although they are possibly both regulated by *p53*. Clearly, further investigations are still needed to clarify upstream signaling of PINT87aa.

**The tumor-suppressive role of PINT87aa.** To understand the potential clinical implications of PINT87aa, we detected the expression of *circPINTexon2* and PINT87aa in human glioma samples and paired adjacent normal tissues. *CircPINTexon2* and PINT87aa were expressed in tumor-adjacent normal brain tissues, but their expression decreased in all brain tumor tissues, similar to *LINC-PINT* (Fig. 8a and Supplementary Fig. 13). Specifically, WHO grade IV glioblastomas exhibited the lowest expression levels of PINT87aa. In WHO grade I astrocytomas, PINT87aa was detectable, but levels were decreased compared with those in normal tissues. These data suggest that PINT87aa negatively impacts the clinical prognosis of glioma. We obtained similar results regarding *circPINTexon2* and PINT87aa expression in other human malignancies, including breast cancer, hepatic cell carcinoma, and gastric cancer (Fig. 8b and Supplementary Fig. 13). Clearly, *circPINTexon2* and PINT87aa downregulation is normal in human malignancies. In animal models, 456 and 4121 cells stably overexpressing PINT87aa either on linear or circular plasmids exhibited decreased in situ tumorigenic potential compared with control cells, as assessed by tumor growth and animal survival (Fig. 8c). Furthermore, Hs683 and SW1783 PINT87aa K.O. cells and control cells were subcutaneously injected into nude mice, and tumor growth was subsequently monitored with calipers and through in vivo fluorescence imaging (Fig. 8d). Compared with their parental cells, PINT87aa K.O. cells resulted in significantly increased xenograft tumor volumes, further supporting the anti-cancer effects of the PINT87aa peptide in vivo.

**Discussion**
CircRNAs are a widespread RNA species in the human transcriptome[4,52]. In addition to acting as regulators of gene expression or development by adsorbing microRNAs, circRNAs were recently demonstrated to be critical in human malignancies[53–55]. However, only a few circRNAs contain perfect multiple microRNA trapping sites, raising the question of whether circRNAs exhibit functions beyond acting as microRNA sponges[2,3]. Recent evidence has confirmed the existence of functional peptides translated from sORFs in ncRNAs, including pri-microRNAs and lncRNAs[21,22], suggesting that the coding potential of these ncRNAs has been largely underestimated. To explore the coding potential of circRNAs, we perform RNC-RNA deep sequencing and confirmed that an endogenous circRNA (*circPINTexon2*) encodes a tumor-suppressive peptide in human cells. Although the 87-aa sORF did not span the *circPINTexon2*

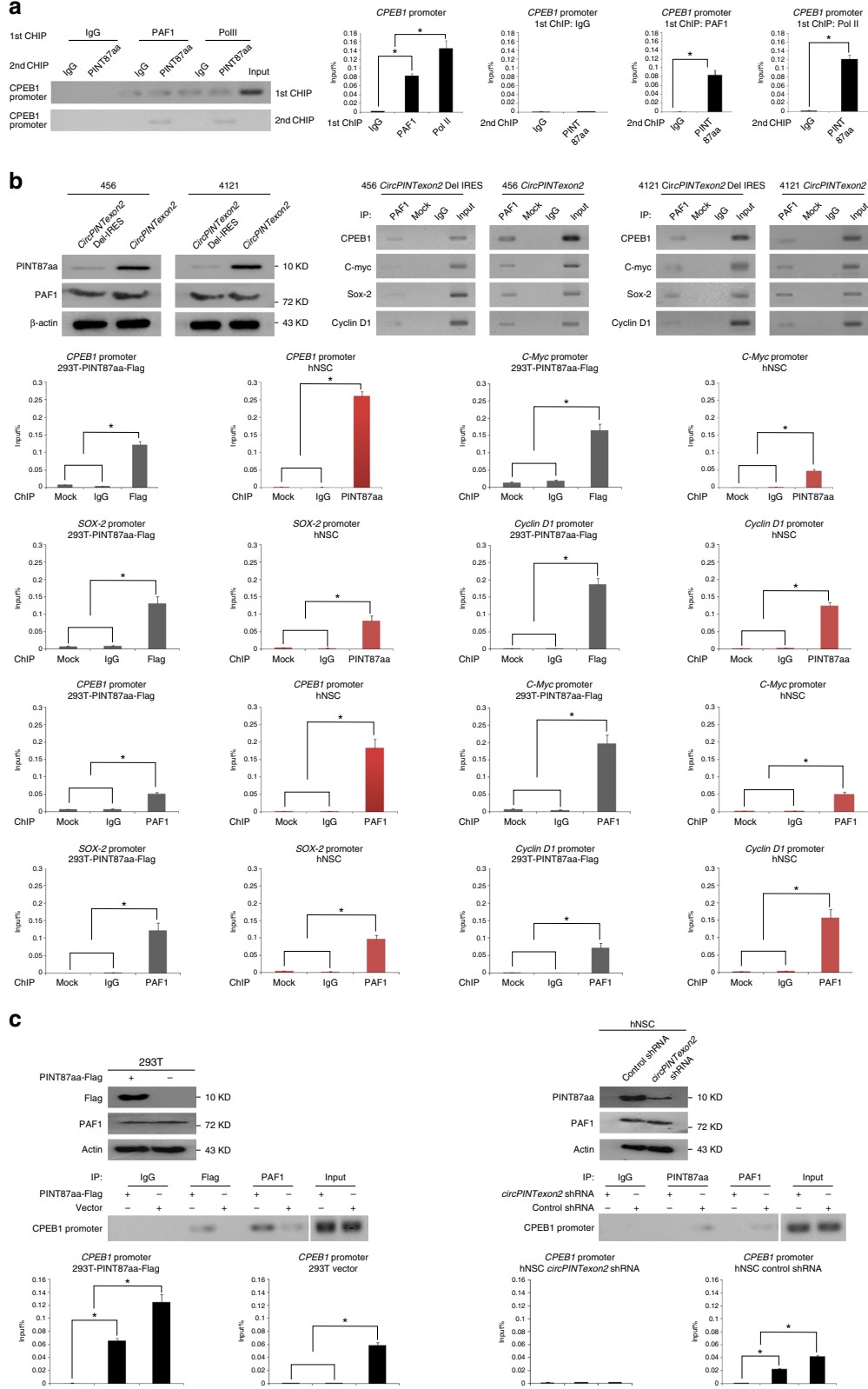

**Fig. 7** PINT87aa decides the localization of PAF1 complex on target genes promoter. **a** 2nd ChIP assay was used to determine the co-occupation of PAF1 complex/PINT87aa in *CPEB1*'s promoter region by using RT-PCR or q-PCR. At least three independent experiments were performed. **b** ChIP assay was performed in *circPINTexon2* stably overexpressed 456 and 4121 cells by using indicated antibodies. PCR products of indicated genes' promoter were analyzed by using RT-PCR or q-PCR. **c** PINT87aa was overexpressed or knocked-down in 293T or hNSC cells by using indicated plasmid or shRNA, respectively. ChIP assay was performed by using indicated antibodies and PCR products of *CPEB1* promoter region were analyzed by using RT-PCR or q-PCR. **b**, **c** At least three independent experiments were performed

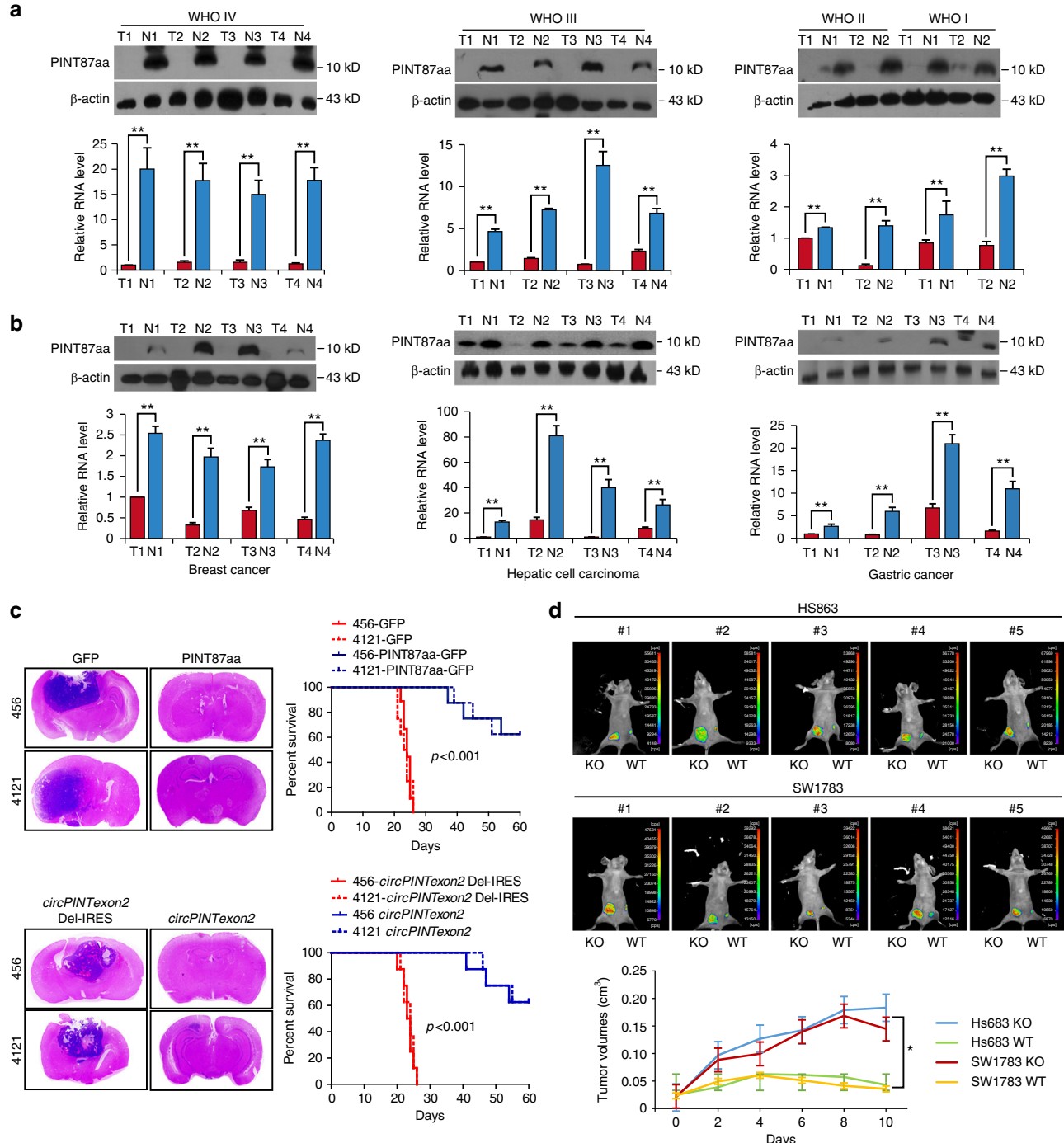

**Fig. 8** Clinical implications of *circPINTexon2* and PINT87aa. **a** Expression of *circPINTexon2* and PINT87aa in human glioma samples with different WHO grades and matched adjacent normal tissues. **b** Expression of *circPINTexon2* and PINT87aa in breast cancer, hepatic cellular carcinoma, gastric cancer, and adjacent normal tissues. **c** Anti-cancer effects of PINT87aa in vivo; left, mice with brain xenograft tumors were sacrificed 60 days after implantation; right, Kaplan–Meier survival curves of mice implanted with 456 and 4121 cells stably overexpressing PINT87aa and their respective parental cells ($N = 8$ mice/treatment group, $P < 0.001$, two-tailed log-rank test). **d** Nude mice were subcutaneously injected with $5 \times 10^6$ Hs683 and SW1783 PINT87aa K.O. cells and their parental cells using Matrigel and were then monitored daily. Day 0 indicates the first day of injection. Tumor progression was monitored using a small animal imaging system or through caliper measurements. Error bars indicate standard error ($N = 5$ mice/treatment group, $P < 0.01$, two-tailed log-rank test)

junction, multiple lines of evidence indicated that this peptide was not translated from its related linear RNA. Similarly, Pamudurti et al. and Yang et al. used ribosomal profiling to identify translatable circRNAs, showing that translatome high-throughput technologies are the most reliable methods to discover coding circRNAs[25,27].

Although we identified approximately 320 potentially translatable circRNAs, we focused only on candidates whose host genes were previously designated as ncRNAs. This criterion may result in certain real coding-circRNAs being overlooked, but it reduces the false-negative rate to a maximum extent. Among the candidate coding-circRNAs, only few exhibit spanning junction sORFs.

A spanning junction sORF is the distinctive feature of circRNA-encoded peptides. Most translatable circRNAs share similar sORFs with their related linear RNAs, although it is possible for their functions to differ. In addition, pre-mRNAs that generate long ncRNAs may also be back-spliced into translatable circRNAs, as is the case for *LINC-PINT* and *circPINTexon2*. Our discovery suggests that coding RNAs and ncRNAs are not very different. A recent publication described a coding circRNA database (circRNADb) in which 16,328 circRNAs were annotated as having putative ORFs, 7170 of which contained IRES elements. Furthermore, 46 circRNAs from 37 genes were reported to have corresponding proteins[30]. We match our screening results with circRNADb, but *circPINTexon2* is not included because circRNADb excludes peptides less than 100 aa in length. Although this report did not provide experimental evidence supporting the coding ability of circRNAs, we believe that circRNADb provides an initial reference for screening translatable circRNAs based on our comparative results. Furthermore, our results show that circRNAs encoding peptides shorter than 100 aa should not be disregarded.

*CircPINTexon2* was previously identified in several cancer cell lines with only 1–2 junction reads[7]. In a recently reported brain circRNA dataset, *circPINTexon2* was not annotated[8]. The GC-rich junction region of *circPINTexon2* or the sequencing depth may result in it being overlooked during high-throughput identification; nevertheless, absolute quantification PCR revealed that *circPINTexon2* copy numbers were not necessarily very low in normal brains in our study and a most recent study supported that the translation efficiency of circRNAs may be higher than linear mRNA (most likely due to the higher stability of circRNAs)[44].

Thus far, no coding circRNAs generated from long ncRNAs have been reported. Based on our results, *circPINTexon2* exerts its biological functions independently in glioma, although it is generated from *LINC-PINT* gene. Glioma is a highly heterogenetic tumor[56–58], which also gives rise to some limitations of our research. For instance, the PINT87aa expression was not evaluated in the molecular sub-types of GBM[56]; some of the experiments were performed not by using the patient-derived BTICs; and some molecular features such as chromosome 7 hypermethylation was not studied regarding to PINT87aa. Also, some critical genes variation such as *c-Myc*, *sox-2* may also be induced by intra-tumoral heterogeneity instead of PINT87aa expression[59]. Nevertheless, our data supported that PINT87aa is a potential tumor-suppressive peptide and is lowly expressed in human cancers other than glioma. Substantial further works, including PINT87aa with GBM sub-types, PINT87aa upstream regulation or even single-cell level investigation, are needed for clarifying PINT87aa's comprehensive tumor-suppressive roles.

Although no evidence showed that PINT87aa directly binds to DNA and act as a transcription factor, its interaction with PAF1 complex suggested its role during transcriptional elongation. To our knowledge, the binding partner of PAF1 complex decided its biological functions and cellular localization[60–62]. As a newly identified interacting protein, PINT87aa may serve its role in deciding PAF1 complex proper localization.

Our discovery of peptides encoded by circRNAs and their regulatory effects on human malignancies show that circRNAs may exert more biological functions than previously predicted. Additionally, our findings raise many questions. For example, how many translatable circRNAs are present in mammalian cells, and what are their general functions? With respect to this study, how does circPINTexon2 translocate into the cytoplasm, while *LINC-PINT* remains in the nucleus? Do *circPINTexon2* and PINT87aa serve as specific tumor biomarkers for detection, intervention, and/or prognostication? Further investigations are

clearly warranted to address these questions. Nevertheless, the present study shows that circRNA-encoded small peptides regulate human cancer behavior and may have important clinical implications and applications. Additionally, our results provide a fresh perspective regarding circRNAs and long ncRNAs, whose biological functions are largely yet to be revealed.

## Methods

**Human cancer and normal tissues**. All human cancerous and adjacent normal tissues were collected from the 1st Affiliated Hospital of Sun Yat-sen University. The human materials were obtained with informed consent, and the study was approved by the Clinical Research Ethics Committee.

**Cell culture and RNase-R treatments**. All cells used in this study were tested for mycoplasma contamination. 293T, U251, A172, Hs683, and SW1783 cells were authenticated December 2015 by STR sequencing. 293T cell was purchased from ATCC (ATCC number: CRL-11268), and A172 cell was also from ATCC (ATCC number: CRL-1620). Specifically, A172 cells were not contained by U251MG as listed in ICLAC (http://iclac.org/databases/cross-contaminations), authenticated by STR sequencing. U251, Hs683, and SW1783 cells were kindly provided by Dr. Suyun Huang (MD Anderson). These cells were cultured in Dulbecco's modified Eagle's medium (Gibco, Carlsbad, CA) supplemented with 10% fetal bovine serum (Gibco, Carlsbad, CA) according to standard protocols. NHA were purchased from Lonza and cultured using an AGM Bullet Kit™ (Lonza, Walkersville, MD) as recommended by the manufacturer. BTICs including 456, 4121, 387, and H2S were kindly provided by Dr. Jeremy N. Rich (UCSD) and were cultured in Neurobasal medium (Gibco, Carlsbad, CA) with B27 (without vitamin A), basic fibroblast growth factor (20 ng ml$^{-1}$, R&D, Indianapolis, IN) and epidermal growth factor (20 ng ml$^{-1}$, R&D, Indianapolis, IN). hNSC, purchased from Gibco (Gibco, Carlsbad, CA, A15654), were cultured as the recommendation of the manufacturer (StemPro® NSC SFM (A10509-01)) supplemented with 2 mM GlutaMAX™-I Supplement (35050), 6 U/ml heparin (Sigma, St. Louis, MO, H3149), and 200 μM ascorbic acid (Sigma, St. Louis, MO, A8960). Only the early passages of hNSC were used (less than 3 passages). RNase-R (Epicentre Biotechnologies, Madison, WI) treatment (20 U/μl) was performed on total RNA (20 μg) at 37 °C for 15 min.

**Ribosome-nascent chain complex (RNC) extraction**. Cells were pre-treated with 100 μg/ml cycloheximide for 15 min, followed by pre-chilled phosphate buffered saline washes and the addition of 2 ml of cell lysis buffer [1% Triton X-100 in ribosome buffer (RB buffer): 20 mM HEPES-KOH (pH 7.4), 15 mM MgCl₂, 200 mM KCl, 100 μg/ml cycloheximide, and 2 mM dithiothreitol]. After a 30-min ice bath, cell lysates were scraped and transferred to pre-chilled 1.5-ml tubes. Cell debris was removed by centrifuging at 16,200$g$ for 10 min at 4 °C. Supernatants were transferred to the surface of 20 ml of sucrose buffer (30% sucrose in RB buffer). RNCs were pelleted after ultra-centrifugation at 185,000$g$ for 5 h at 4 °C. RNC-RNA and total RNA were reverse-transcribed with a RevertAid H Minus First Strand cDNA Synthesis Kit (Thermo Fisher, Waltham, MA) using random hexamer primers (oligo-dTs primers for poly-A RNAs were used in Fig. 3c), following the manufacturer's instructions. The cDNA was then subjected to PCR with specific primers using DreamTaq PCR Master Mix (Thermo Fisher, Waltham, MA) for 40 cycles (95 °C for 30 s and 68 °C for 60 s for each cycle).

**Strand-specific RNA-seq library construction and sequencing**. Total RNA was isolated using TRIzol (Life Technologies, Carlsbad, CA). After total RNA was extracted, rRNA was removed by using VAHTS Total RNA-seq (H/M/R) Library Prep Kit for Illumina (Vazyme Biotech Co., Ltd, Nanjing, China) to retain other types of RNA, including mRNAs and ncRNAs. The enriched mRNAs and ncRNAs were fragmented into short fragments in fragmentation buffer (MgCl₂) integrated into VAHTS Total RNA-seq (H/M/R) Library Prep Kit for Illumina (Vazyme Biotech Co., Ltd, Nanjing, China), then the RNA fragments were reverse-transcribed into cDNA with random primers. Second-strand cDNA was synthesized with DNA polymerase I, RNase H, dNTPs (dUTP instead of dTTP), and buffer. Next, the cDNA fragments were purified with a QiaQuick PCR extraction kit, end-repaired, underwent poly(A) addition, and ligated to Illumina sequencing adapters. Then, UNG (uracil-N-glycosylase) was used to digest the second-strand cDNA. The digested products were size-selected via agarose gel electrophoresis, PCR-amplified, and sequenced using an Illumina HiSeq™ 4000 by Gene Denovo Biotechnology Co. (Guangzhou, China).

**Bioinformatics analysis and identification of target circRNAs**. The short reads alignment tool Bowtie2 was used to map reads to an rRNA database. The mapped rRNA reads were then removed. The remaining reads were further subjected to alignment and analysis. The rRNA removed reads were then mapped to a reference genome (Human genome hg38) using TopHat2 (version 2.0.3.12). Reads that mapped to genomes were discarded according to the mapping information recorded in bam files using in-house perl scripts, and unmapped reads were then collected for circRNAs identification. Next, 20mers from both ends of the

unmapped reads were extracted using in-house perl scripts and aligned to the reference genome (bowtie2, version 2.3.0) to locate unique anchor positions within splice sites. Anchor reads that aligned in the reverse orientation (head-to-tail) indicated circRNA splicing and were then subjected to find_circ (version 1.2, https://github.com/marvin-jens/find_circ/) to identify circRNAs. The anchor alignments were then extended such that the complete aligned reads and the breakpoints were flanked by GU/AG splice sites. A candidate circRNA was called if it was supported by at least two unique back-spliced reads from at least one sample. Host genes of identified circRNAs were determined using in housed perls scripts according to gtf files from Ensembl database (http://www.ensembl.org/). CircRNAs were also blasted against circBase to determine if they have been reported. circRNAs that were not annotated were defined as newly discovered circRNAs. To quantify circRNAs, back-spliced junction reads were scaled to RPM (reads per million mapped reads) using the following formula:

$$RPM = \frac{10^6 C}{N}$$

In this formula, $C$ is the number of back-spliced junction reads that uniquely align to a circRNA. $N$ is the total number of back-spliced junction reads. Differential expression analysis was performed using edgeR in OmicShare, an online platform for data analysis (www.omicshare.com/tools). The default parameters of edgeR were used, and differentially expressed genes (DEGs) were selected based on log2 fold-changes ≥1 and q-values > 0.05. Because there were no replicates in this study, the biological coefficient of variation (BCV), which is the square-root of dispersions, was set to 0.01 following the suggestion of the edgeR official manual. GO enrichment analysis was performed for the host genes of differentially expressed circRNAs.

**Northern blotting**. Approximately 10–20 μg of total RNA and circular RNA was run in a 1.2% agarose gel containing formaldehyde. The RNA was then transferred to Amersham hybond-N1 membranes (GE Healthcare, Pittsburgh, PA). The membranes were hybridized with digoxin-labeled DNA oligonucleotides specific to LINC-PINT exon 2 in Church buffer (0.5 M NaPO$_4$, 7% SDS, 1 mM EDTA, 1% BSA, pH 7.5) at 37 °C and washed in 2× SSC (300 mM NaCl, 30 mM Na-citrate, pH 7.0) with 0.1% SDS at room temperature. The membranes were finally exposed on phosphorimager screens and analyzed using Quantity One or Image Lab software (Bio-Rad, Foster City, CA).

**RNA fluorescence in situ hybridization (FISH)**. Oligonucleotide probes complementary to circular exon 2 of LINC-PINT and linear exon 2 of LINC-PINT were designed using the Clone Manager suite of analysis tools (Sci Ed Central, listed in Supplementary Table 5). Cells were seeded on glass coverslips in 12-well plates. The cells were washed in PBS, fixed in 4% paraformaldehyde for 15 min, and then permeabilized overnight in 70% ethanol. Next, the cells were washed twice in PBS containing 5 mM MgCl$_2$ (PBSM) and rehydrated for 10 min in 50% formamide and 2× SSC. For FISH, cells were incubated at 37 °C in a solution containing 50% formamide, 2× SSC, 0.25 mg/ml Escherichia coli transfer RNA, 0.25 mg/ml salmon sperm DNA (Invitrogen, Carlsbad, CA), 2.5 mg/ml BSA (Roche, Indianapolis, IN), and fluorescently-labeled circular probes and linear probes at 125 nM (obtained from Generay Biotech Co, Ltd, Shanghai, China). After 12 h, the cells were washed twice for 20 min at 37 °C in 50% formamide and 2× SSC followed by 5-min washes for 4 times in PBS [with the penultimate wash containing 4,6-diamidino-2-phenylindole (DAPI)] and an additional brief wash in nuclease-free water. The cells were mounted in ProLong Gold (Invitrogen, Carlsbad, CA) and left overnight at room temperature.

**Plasmids and transfection**. GFP and mCherry sequences were amplified from the pEGFP-C1 and pLVX-mCherry-N1 vectors (Takara, Mountain View, CA), respectively. The wild-type and mutant IRES sequences of circular LINC-PINT were obtained through chemical gene synthesis. mCherry-IRES-GFP frames were obtained via overlap PCR and were then cloned into the psin-EF2 vector (Daen Gene Co, Ltd, Guangzhou, China) at the EcoRI and BamHI sites. GFP sequences without the ATG initiation codon and TAA stop codon were inserted into the C-terminal putative coding sequences of PINT87aa and circPINT69aa. Next, exon 2 of LINC-PINT, together with PINT87aa-GFP or circPINT69aa-GFP, was cloned into the psin-EF2 vector using the EcoRI and BamHI sites. For the artificial circRNA overexpression vectors, the junction of natural circPINTexon2 was moved inside the 87-aa ORF, and the side-flanking acceptor and donor sequences were added. The IRES sequence was deleted in the negative control plasmid, and the CMV-87aa-ORF linear overexpression vector was cloned as a positive control. The plasmids were transfected using Lipofectamine 3000 (Invitrogen, Carlsbad, CA) according to the manufacturer's instructions.

**Stable cell line generation**. For stable PINT87aa overexpression, the artificial circPINTexon2 overexpression plasmid or the linear 87-aa-GFP ORF was cloned into the psin-EF2 vector using EcoRI and BamHI for lentiviral production. The cell lines were then infected, followed by selection with 2 μg/ml puromycin for 72 h. To generate PINT87aa stable knockdown cell lines, lentivirus-induced shRNA (GenePharma, Shanghai, China) was used according to the manufacturer's instructions (shRNA sequences are listed in Supplementary Table 5).

**Target DNA deletion using CRISPR/Cas9 technology**. We designed CRISPR gRNAs to target the 87-aa ORF sequences of PINT87aa using the online software at http://crispr.mit.edu/. To clone the target sequences into the lentiCRISPRv2 (AddGene 52961) backbone, we synthesized oligos containing the same overhangs after BsmBI. The transfer plasmid lentiCRISPRv2-gRNA (1.2 μg) was co-transfected into 293T cells with the lentiviral packaging plasmids pVSVg (AddGene 8454, 0.5 μg) and psPAX2 (AddGene 12260, 1 μg). Sixty hours post-transfection, lentiviral infection of the SW1783 and HS683 cell lines was performed. After 48 h, cells were selected with 2 mg/ml puromycin for 1 week. Then, the cells were trypsinized and plated in 96-well plates at a seeding density of 1 cell/well. The cells were monitored for single colony formation and expanded upon confluency. A DNA extraction kit (QIAGEN, Germantown, MD) was used for DNA extraction. PCR was carried out to amplify the targeting region, and Sanger sequencing was performed to detect gene mutations.

**Antibody generation and western blotting**. A polyclonal antibody against the 87-aa peptide produced by circPINTexon2 was obtained by inoculating rabbits. The antibody was purified using affinity chromatography columns. After extraction with RIPA buffer and quantified with a BCA kit (Thermo, Waltham, MA, 23228), equal loading proteins of cell lysate or tissue lysate were added to each well of SDS PAGE. Followed by electrophoresising, transfer-membraning, and blocking with 5% non-fat milk in PBST for 1 h, then diluted primary antibodies were incubated at 4 °C overnight. After washing with PBST every 10 min for 3 times, diluted horseradish peroxidase (HRP)-conjugated secondary antibodies were incubated for 1 h at room temperature, then the signals were visualized. The 87-aa antibody was used at a 1:2000 dilution. The other primary antibodies used were anti-PAF1 (abcam; ab137519; 1:1000), anti-β-actin (Sigma; A5441; 1:5000), anti-GFP (abcam; ab1218; 1:3000), anti-flag (Sigma; F3165; 10 μg/ml; 1:1000), anti-HA (abcam; ab18181; 1:1000), anti-γ-H2AX (abcam; ab2893; 1:1000), anti-CPEB1 (abcam; ab155204; 1:5000), anti-Cyclin D1 (abcam; ab134175; 1:1000), anti-C-myc (abcam; ab51156; 1:1000), anti-SOX2 (abcam; ab97959; 1:1000), anti-H3 (abcam; ab1791; 1:1000), anti-EGFR (abcam; ab52894; 1:1000), anti-MET (abcam; ab51067; 1:1000), anti-PDGFR (abcam; ab32570; 1:5000), anti-P53 (abcam; ab26; 1:1000).

**PCR**. Reverse transcription for mRNA and circRNAs was performed using an MMLV-RT kit (Takara, Mountain View, CA) with random hexamers according to the manufacturer's instructions. PCR was subsequently performed with a 1:10 dilution of reverse-transcribed cDNA. The PCR products were run in a 2% agarose gel. Real-time q-PCR (RT-q-PCR) was performed using an Applied Biosystems PCR System (ABI 7500). RT-qPCR SYBR Green Mix (Takara, Mountain View, CA) was employed, with forward and reverse primers at 50 nM in a 20-μl reaction system. Relative expression levels were calculated using the 2-ΔΔCT method. To determine the absolute quantity of RNA, the purified PCR product amplified from cDNA corresponding to the circPINTexon2 sequence was serially diluted to generate a standard curve. (Oligo sequences are listed in Supplementary Table 5.)

**Immunoprecipitation (IP)**. Cells were lysed in co-IP buffer [10 mM HEPES (pH 8.0), 300 mM NaCl, 0.1 mM EDTA, 20% glycerol, 0.2% NP-40, protease and phosphatase inhibitors]. The lysates were then centrifuged and cleared via incubation with 25 μl of protein A/G agarose for 1.5 h at 4 °C. The pre-cleared supernatant was subjected to IP using the indicated primary antibodies at 4 °C overnight. Then, the protein complexes were collected by incubating with 30 μl of protein A/G gel for 2 h at 4 °C. The collected protein complexes were separated via SDS-PAGE and analyzed by performing MS or blotting.

**Neurosphere formation assay**. Extreme limiting dilution analysis (ELDA) was performed to evaluate self-renewal capacity. BTICs were dissociated in TrypLE™ Select (Gibco, Carlsbad, CA) for 5 min at 37 °C. Cells were triturated into a single-cell solution. The solution was incubated with Hoechst (Thermo, Waltham, MA, 33342) for 30 min at 37 °C. Live cells were identified using a LIVE/DEAD staining kit (Thermo, Waltham, MA, L10119). Live cells were sorted into 96-well plates. Spheres were counted at 14 days. Cell density per well ranged from 1, 10, 25, 50, 100, 250, 500 to 1000. Each condition was tested in 10 independent wells. Volume of medium per well was 200 μl medium with growth factors spike-ins every 3–4 days. Neurosphere-forming capability was determined using the ELDA web-based tool (http://bioinf.wehi.edu.au/software/elda/).

**Cell proliferation assays**. Cells were seeded into 96-well plates. At the indicated time points, the cells were incubated with 100 μl of sterile MTT for 4 h at 37 °C, after which the medium was removed and replaced with 150 μl of DMSO. The absorbance was measured at 570 nm. All experiments were performed in triplicate.

**EdU incorporation assay**. 8-Well chamber slides were coated with poly-L-lysine. Cells were then seeded at 40,000 cells per well. 10 μM EdU was added to each well.

Cells were fixed after 24 h using 4% paraformaldehyde in PBS and stained using the Click-iT EdU kit as protocol (Invitrogen, Carlsbad, CA). Proliferation index was then determined by quantifying percentage of EdU labeled cells using confocal microscopy at 200× magnification. All experiments were performed in three biological replicates.

**LC-MS analysis**. Proteins were separated via SDS-PAGE, and gel bands were manually excised and digested with sequencing-grade trypsin (Promega, Madison, WI). The digested peptides were analyzed using a QExactive mass spectrometer (Thermo Fisher, Carlsbad, CA). Fragment spectra were analyzed using the National Center for Biotechnology Information nonredundant protein database with Mascot (Matrix Science).

**Colony formation assays**. Cells were plated in 6-well plates (1000 cells per plate), cultured for 10 days, fixed with 10% formaldehyde for 5 min, stained with 1.0% crystal violet for 30 s, and counted. All experiments were performed in three biological replicates.

**ChIP and re-ChIP assays**. For the ChIP assay, $2 \times 10^6$ cells were prepared with the ChIP assay kit (Cell Signaling Technology, Danvers, MA, 56383) according to the manufacturer's instructions. The resulting precipitated DNA samples were analyzed by PCR. In the re-ChIP assay, $10^7$ cells were used for the first-step ChIP. The DNA complexes were first immunoprecipitated using the indicated antibodies and then eluted by incubation for 30 min at 37 °C in 100 µl of 10 mM DTT. After centrifugation, the supernatant was diluted 50× with re-ChIP buffer and immunoprecipitated again using the indicated antibodies, as with the ChIP procedure. (Primer sequences are listed in Supplementary Table 5.) All experiments were performed in three biological replicates.

**Cell cycle analyses**. Cells were harvested via trypsinization, washed in ice-cold PBS, fixed in ice-cold 75% ethanol in PBS, centrifuged at 4 °C, and suspended in PBS. RNase A was then added at a final concentration of 4 mg/ml, followed by incubation at 37 °C for 30 min, after which 20 mg/ml propidium iodide (Beyotime, Shanghai, China) was added, and the samples were incubated for 20 min at room temperature. The cells were analyzed via flow cytometry. All experiments were performed in three biological replicates.

**IF staining and confocal microscopy**. Cells were grown on chamber slides precoated with poly-L-ornithine and fibronectin. The cells were fixed with 4% paraformaldehyde, permeabilized for 5 min with PBS containing 0.1% Triton X-100 (PBS-T), quenched with 50 mM NH$_4$Cl in PBS-T, and blocked with 1% BSA in PBS-T. Immunostaining was performed with appropriate primary and secondary antibodies, and images were acquired using an Olympus FluoView FV1000 confocal microscope.

**Animal care and ethics statement**. Four-week-old female BALB/c-nu mice were purchased from the Laboratory Animal Center of Sun Yat-sen University. Mice were housed in a temperature-controlled (22 °C) and light-controlled pathogen-free animal facility with free access to food and water. All experimental protocols concerning the handling of mice were approved by the institutional animal care and use committee of Sun Yat-sen University.

**Intracranial injection**. We intracranially injected 2000 cells for each of the indicated BTIC types into nude mice. Eight mice were injected for each group. The mice were sacrificed after 60 days or when they showed clinical symptoms such as weight loss and orientation dysfunction. The brain of each mouse was harvested, fixed in 4% formaldehyde and embedded in paraffin. Tumor formation and phenotypes were determined through histologic analysis and assessed in hematoxylin and eosin-stained sections. Total survival curves were calculated.

**Subcutaneous xenograft assay**. Xenograft experiments were performed through the subcutaneous injection of $5 \times 10^6$ cancer cells into nude mice. Five mice were injected for each group. Day 0 indicates the first day of injection. Tumor progression was monitored by imaging with a Xenogen Spectrum small animal imaging system (Caliper, Hopkinton, MA) or via caliper measurements, as indicated.

**Statistical analysis**. Statistical tests were conducted using Prism (GraphPad) software unless otherwise indicated. Data are presented as mean ± s.e.m. from three independent experiments. For parametric data, unpaired, two-tailed Student's $t$-tests were used. For non-parametric data, two-sided Mann–Whitney test was used. Data distribution was assumed to be normal, but this was not formally tested. The limiting dilution assay to test for neurosphere-forming capacity was analyzed with a Chi-squared test using the ELDA web-based tool (http://bioinf.wehi.edu.au/software/elda/). A level of $P < 0.05$ was used to designate significant differences. For all experiments, analyses were done in biological triplicate. No animals or data points were excluded from the analyses for any reason. Blinding and randomization were performed in all experiments. Statistical analyses for RNA-seq data are described above in the respective sections.

## Data availability
The RNA-seq data obtained in this study have been deposited in the SRA database, with the accession code PRJNA401369. Other data that support the findings of this study are available from the corresponding author on reasonable request.

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

## Acknowledgements

We are sincerely grateful to Dr. Jeremy N. Rich (UCSD) for kindly providing BTICs. This work was supported in part by the National Key Research and Development Program of China (2016YFA0503000 to N.Z.), the National Natural Science Outstanding Youth Foundation of China (81822033 to N.Z.), the National Natural Science Foundation of China (81370072, 81572477, 81772683 to N.Z. and 81802484 to K.H.), the Science and Technology Project of Guangdong Province (S2013050014535 and 2016A050502017 to N.Z.), the 863 Research Program (2014AA020504 to G.Z.)

## Author contributions

N.Z. and S.H. conceived and designed the project. M.Z., K.Z., and X.X. planned and performed most of the experiments. N.Z., K.X., and S.H. wrote the manuscript. S.Y., P.W., N.H., and X.Y. helped with the experiments. H.L. designed and performed some of the experiments shown in Fig. 4. J.X. performed the gastric cancer experiments. F.X. performed the statistical analysis. J.L. provided experimental advice. G.Z. performed the RNC-RNA-related experiments. H.Z. performed the bioinformatic analysis. K.H. helped with the data organization and figure preparation. Y.Y. helped with the revision. S.H. and N.Z. supervised the study.
