## [Peer Review File · Nature Communications]

Reviewers' Comments:

Reviewer #1:

Remarks to the Author:

The manuscript describes a comprehensive and interesting study that provides novel insights into the largely unexplored world of circular RNAs (circRNA). Mainly, they characterised a circular RNA originating from a long non-coding RNA, and they showed that this circRNA encodes an 87 amino acids peptide. Furthermore, they found that the peptide interacts with PAF1 and thereby inhibits transcription of oncogenes. As a result, it suppresses the proliferation of cancer cells.

Overall the authors provided an excellent piece of work that will catch the attention from a wide scientific community to the unexplored realm of circRNAs and the implication of their expression on cell functions. There are, however, many minor issues in the manuscript -especially in the methods and in the description of the approaches used in bioinformatics analysis - that need to be addressed by the authors. Basically, it needs the careful revisiting and checking all aspects for publication.

Specific comments to each part of the manuscript:

Methods:

1. Page 6, line 126 - 127 The authors stated the following statement "RevertAid H Minus First Strand cDNA Synthesis Kit (Thermo Fisher, Waltham, MA) using oligo-dTs or random hexamer primers, following the manufacturer's instructions." I am wondering if the authors mean oligo-dTs AND random hexamers? The authors should clarify and add clear description of the procedure.

2. Page 7, line 134 "rRNA was removed to retain other types of RNA, including mRNAs and ncRNAs." Please provide some details on how the ribosomal RNAs were removed. If a kit is used then please provide the name of the kit and company.

3. Page 7, line 135: The authors stated 'The enriched mRNAs and ncRNAs were fragmented into short fragments in fragmentation buffer and reverse-transcribed into cDNA with random primers' Please provide more details about the RNA fragmentation protocol. Was the RNA fragmented using MgCl₂ and heat, enzymatically or by sonication?

Bioinformatics:

4. Page 7, line 147: The authors stated the following "The rRNA reads removed from each sample were then mapped to a reference genome using TopHat2 (version 2.0.3.12), respectively". If the rRNA reads were removed in the first step (lines 145 – 146), why do they have to be removed again? Did the authors mean that the mRNA reads were mapped and removed? Please also indicate the reference genome used and the version. It is not clear if the authors mean 'reference genomes' or 'reference genome'? Also the word 'respectively' seems to be out of place. Finally, the authors want to indicate how the mapped reads were removed and what tools were used.

5. Page 7, line 150: I am wondering how the authors extracted the 20mer reads and what program did they use to align those reads to a reference genome? Was TopHat again or another aligner?

6. Page 8, line 155 -156: The authors should provide the name of the program used to annotate the identified circRNAs.

7. Page 8, line 166: Please correct the abbreviation of correct p-value to q-value instead of Q value.

8. Page 8, line 169: Please indicate the tool used for Gene ontology enrichment and use the full name followed by the abbreviation, gene ontology (GO).

Results section:

9. Page 16, line 346 How is it possible that the circRNAs identified from total RNA-seq are less than those identified from the RNC-RNA-seq? Should it not be the opposite?

10. Page 16, line 346. The authors indicated that 4597 circRNAs were already known ones without giving any details about how many were identified from total RNA-seq and how many from RNC-RNA-seq. Please show a breakdown of the number and more details would benefit this part.

11. Page 17, line 353. Minor inconsistency: Please write the term followed by abbreviation between brackets, False discovery rate (FDR).

Figures:

12. Overall figure 1 needs to be reviewed and edited by the authors to match the writing in the results section, correct inconsistencies in the numbering and the names of the figure panels. Examples: Fig. 1F, the figure legend does not match the information provided in the results section (line 354). The authors referred to the heatmap in the figure legend whereas the figure itself shows a scatter plot which matched the description in the results section (line 354). There is no mention of the scatter plot in the figure legend and there is no mention of the heatmap in the results section. Furthermore, the authors should show the units the used on both axes of the scatter plots and on the key of the heatmap. It would also be more informative if more details are provided, for example if the pseudo-count was used then please indicate that in the figure legend. Figure 1G right, and there is no left and right in figure 1G, there is only a Venn diagram unless the authors are referring to the left and right sides of the Venn diagram.

13. Generally for figures. All figures should be checked for proper labelling and size. Furthermore, the content of the figures should be clearly explained in the legends.

14. The sequence of the 87 amino acids peptide should be shown in one of the figures. It is shown in part but I could not find the entire sequence somewhere in the manuscript. Furthermore, an access. No. for the peptide should be obtained and added to the paper.

Reviewer #2:

Remarks to the Author:

The manuscript by Zhang et al. showed that an endogenous circRNA generated from a long noncoding RNA might encode a regulatory peptide in human cells. The authors identified an 87-amino-acid peptide encoded by the circular form of the long intergenic non-protein-coding RNA p53-induced transcript (LINC-PINT). This peptide could suppress cancer cell proliferation in vitro and in vivo. Mechanistic studies revealed that it directly interacts with PAF1 and sequentially inhibits the transcriptional elongation of some oncogenes. Furthermore, the expression of this peptide and its corresponding circRNA were downregulated in human cancers compared with the levels in normal tissues. These results may provide novel evidence that circRNA can be translated and functions in cancer. This study is potentially interesting and at times comprehensive, however, it lacks in many aspects and is therefore not convincing.

1. The result of RNC-RNA-seq data is confusing. The authors performed RNA-seq analyses of ribosomal RNA-depleted total RNAs and ribosome-nascent chain complex (RNC)-RNAs from two cell lines. The total RNAs should include RNC-RNAs and ribosome-free RNAs. However, a total of 12,863 circRNAs were identified in RNC-RNA-seq, but only 7,017 in total RNA-seq. How about the sequencing depths? All the 12,863 circRNAs identified in RNC-RNA-seq are binding to ribosome and translational? The approach to extract RNC-RNAs is quite rough. The authors should calculate the enrichment of circRNAs in RNC-RNA-seq compared to total RNA-seq (or input).

2. Figure 1 should show the comparison of RNC-RNA-seq and total RNA-seq data, but not just the

differently expressed circRNAs between two cells. Fig.1E is no different and too small, and may be deleted.

3. The existence and expression of circPINTexon2 from LINC-PINT should be carefully checked. Although this circRNA was annotated in circBase, there is no score for it. The authors claimed that this circRNA is highly expressed in 293T cells, how many copies per cell or delta CT (compared to Actin) for LINC-PINT and circPINTexon2? Which one is with higher expression level?
4. Figure2A should provide the sequencing reads information from IGV, which could show the real circRNA and host RNA expression level. The name of host RNA in Figure 2A and 2B is not consistent. The primers for host RNA are not correct (do not locate in Exon1 and Exon2).
5. The coding peptide could also be generated from LINC-PINT. The coding region for this peptide is located in both LINC-PINT and circPINTexon2. Although LINC-PINT is shown in nuclear, it is still present in cytoplasm from ENCODE RNA-seq data. Does introduction of LINC-PINT expressing vector produce the peptide?
6. The authors used GFP fusion protein labeling and found this peptide was concentrated in nucleus. Does this peptide contain nuclear location signal?
7. Why Figure7A have two clear band in PINT87aa WB result in first panel but not others?
8. Some obvious errors: A duplicated construct was shown in Figure3A. has circ-0082389 should be hsa_circ_0082389.

Reviewer #3:

Remarks to the Author:

The MS entitled "A Novel Peptide Encoded by Circular Form of Long Intergenic Non-protein-coding RNA p53-induced Transcript Suppresses Oncogenic Transcriptional Elongation in Human Malignancy" investigates the importance of PINT87 peptide encoded by CircRNA of LINC-PINT ncRNA in human malignancy. The work done in the MS is interesting and is of importance in context of understanding the role of circular RNA's in governing cellular functions, however, there needs to be detailed explanation of some data and to perform more experiments.

Major Comments:

1. FISH experiments shown in Figure 2 describes the differential localisation of linear and circular RNA, with the later being cytoplasmic and former nuclear. Then the experiments performed in Figure 3 shows only CircRNA is translated to give PINT87aa and not the linear form. Further, when CircRNA was overexpressed as a GFP fusion, its localisation was only nuclear. So, considering all the data is it right to assume that in vivo also only Circular version is translatable, due to its presence in cytoplasm and not the linear form? If so, then what could be the factors distinguishing these two and enabling a differential location of these RNA's.
2. GFP-PINT87aa showed nuclear localisation, is this because this contains nuclear localisation signal or it interacts with other proteins like PAF1 in this case to get localised inside nucleus?
3. GFP-PINT87aa seems to also show a punctate pattern of localisation inside nucleus, as seen clearly in Figure 6E. What are these puncta, could they be aggregates, especially considering in the mass spec data a lot of proteins seem to be either splicing factors or ribonucleoproteins, which are known to be aggregates it becomes important to understand this subnuclear localisation? Also, GFP-PINT87aa can also be seen in cytoplasm in U87 cells and not in U251. Why this discrepancy?
4. For the interaction of PAF1 and PINT87aa, it is clear 150-300 aa of PAF1 is important, but what are the regions in PINT87aa which are important? May be revisiting the molecular docking might give clues.
5. Authors describe that PINT87aa interaction with PAF1 is the reason behind the phenotypic effects observed. However, ChIP experiments need to be performed to show that PINT87aa interaction with PAF1 is responsible to low transcription of the later genes. Further, mechanistic insights are needed to understand what PAF1-PINT87aa interaction does.

Minor comments:

Figure 1 legend, in description of panels B and C it should be top and bottom instead of left and right.

1. The readability of heat maps in Figure 1 F should be increased.
2. The quality of gel pics in Figure 3 D and 6B is of questionable quality.
3. What are the other bands in in IP lane, abundance of which are more than the bait.
4. Figure 4 F needs to be referred alongside Figure 5A in the text.
5. Does the PAF1 IP pull down PINT87aa?
6. What are the levels of linear form in the glioma tissues.

Reviewer #4:

Remarks to the Author:

MAJOR CRITICISM:

One of the key hurdles for the development of successful anti-tumor therapeutics is remarkably high level of heterogeneity that is apparent in solid tumors. Glioblastoma is no exception to that rule - multiple subtypes have been described, and they can co-exist within individual tumors. Some subpopulations are invasive, some are not; some are more differentiated, some retain stem cell characteristics. High heterogeneity is reflected both at molecular and phenotypic levels. Thus, to faithfully model pathobiology of glioblastoma it is imperative to use multiple, patient-derived cell populations. Unfortunately, authors of presented manuscript failed to meet this crucial requirement. They used barely one or two established cell lines, cultured in high serum, and non-malignant astrocytes as a control (please see specific criticisms below). This in my opinion disqualifies the manuscript for being accepted in Nature Communications. It is really unfortunate, as the subject of the study is really timely and informative. Authors developed and used appropriate methodology. I am positive that studying novel class of ncRNA, namely circRNAs in the context of heterogeneous disease is promising and interesting area of research.

SPECIFIC CRITICISMS

1) The major flaw of the manuscript however is use inadequate, outdated model. It is simply unacceptable these days to model such heterogeneous disease as glioblastoma using one established cell line for initial screen (and merely two in phenotypic/molecular studies). U87 and U251 glioma cell lines used in most experiments have huge number of indels, copy number variation due to long term cultures serum (PMID – 28074274). They do not recapitulate GBM subtype heterogeneity, that do not recapitulate critical phenotypes (e.g. are not invasive in vivo), they were cultured in 10% FBS for countless generations. Besides, U87 are not what they were thought to be (<http://www.nature.com/news/venerable-brain-cancer-cell-line-faces-identity-crisis-1.20515>).

2) Use of astrocytes as non-malignant control is very controversial. Optimal would be comparing non-malignant neural cells vs. GBM stem cells, both in serum-free culture.

3) In vivo study: apart from using wrong cells to model GBM, the results on Fig. 7C open several eyebrows raising questions: 1) authors claim to sacrifice mice after 70 days and experimental group is seemingly tumor-free. Yet quite shortly after substantial portion of them died. 2) control tumors kill mice in quite surprisingly wide time window (between day 40-70 or 60-90); this never should be the case for control cells. 3) Authors use really high number of cells – half a million; yet controls lived until 70-90 days. Such high number of cells would kill controls within 20 days. At least U87 I used to work with.

4) Fig.2F and Supp Fig 2A – Authors showed the expression of circPINTexon2 and LINC-PINT are mostly cytoplasmic and nuclear respectively using junction-specific probes specific probe and w linear specific probes specific to LINC-PINT. But the expression of translated protein are mostly nuclear (Fig 4A and Fig. 6E) shown only by IF. It is not well addressed, why the proteins are translocated to nucleus even though host gene are mostly cytoplasmic. The author should check the protein level using cell fractionating assay.

5) Line 362 - It is not clear that if the authors have found only 10 non-coding host genes or if they have only selected 10 genes selected out of many? Out of these 10 nc host genes, 5 of them have unique peptide coding potential. It is not well addressed why authors exclude the other 4 (TTN-AS1, CDKN2B-AS1, FIRRE, and XIST from Supplemental Table 3) and chose LINC-PINT for further evaluation? "As mentioned previously, most circRNAs shared the same CDS with their host protein-coding genes. To exclude false-positive data from further investigation, we focused only on non-coding host genes among these 274 candidates. A total of ten non-coding host genes were selected (Supplementary Table 2), and among them was LINC-PINT (ENSG00000231721)." The authors intended to choose to focus on non-coding host genes to rule out the possibility of translating proteins from circRNAs which are backspliced from CDS of host gene. But there 4 more long non-coding RNA listed in Supplemental Table 3. Since these 4 lncRNAs also have coding potential, the authors should address in detail why they chose to exclude these 4 lncRNAs.

6) Figure 2B and Supplementary Figure 1 - According to circBASE (see below) there are three more annotated circular RNA can be backspliced from the host non-coding gene LINC-PINT. Author should check the expression of those circRNAs too and analyze whether they have coding potential or not.

7) Fig.2E (Left) – RNAse R treatment is widely used to increase the circular isoform by depleting most of linear RNAs. It is quite surprising that there was no enrichment of circular isoform after RNAse R treatment.

8) Figure 2F and Supplementary Figure 2 – The authors did not show any data by FISH or qPCR/circRNA localization in GBM cells, only in normal brain and HEK293 cells. Side by side comparison of the expression/localization of circPINT RNA in normal brain vs. tumor tissue and malignant non-malignant cells would be required.

9) Authors did not consider checking the status of p53 before using these cell lines. The p53 status is different among these cell lines (U87- wild type and U251 – mutant). Though author showed the expression of p53 upon overexpression of PINT87aa (Supplementary Figure 12a), it will be interesting to show the expression of the circPINT and its translated protein upon the inducing or inhibiting p53 activity in these cell lines as it has been shown that LINC-PINT expression is induced in a p53-dependent manner (PUBMED ID – 24070194)

10) The peptide sequence spanning circRNA back-splice junction can be explicitly recognized as circRNA-encoded products. Since the peptide from circPINT originated from the region which is on the exon2 of linear host gene, it is reasonable to suppose that the peptides identified might also translated from the linear part. The should clone the full length of LINC-PINT (3144 nt) in their expression vector and check if overexpressing full length host gene can also increase the expression of circPINT87aa protein (See the sequence of LINC-PINT and associated circular) to rule out the possibility that identified peptide is only from the circularPINT not from the linear isoform.

11) Figure 5 – Showing merely cell proliferation and colony formation assay is rather disappointing. What about limiting dilution assay, apoptosis, migration/invasion drug/radiation resistance. The author should test whether the expression of PINT87aa or circPINT would lead to the activation of common or distinct proto-oncogenic drivers (EGFR, PDGFR, MET).

12) Statistical analysis is of rather poor quality.

13) Few samples of non-malignant brain were analyzed

Point by point Response to Reviewers

Manuscript NCOMMS-17-27528

Reviewer #1, Expertise: Transcriptome/translatome sequencing (Remarks to the Author):

The manuscript describes a comprehensive and interesting study that provides novel insights into the largely unexplored world of circular RNAs (circRNA). Mainly, they characterised a circular RNA originating from a long non-coding RNA, and they showed that this circRNA encodes an 87 amino acids peptide. Furthermore, they found that the peptide interacts with PAF1 and thereby inhibits transcription of oncogenes. As a result, it suppresses the proliferation of cancer cells.

Overall the authors provided an excellent piece of work that will catch the attention from a wide scientific community to the unexplored realm of circRNAs and the implication of their expression on cell functions. There are, however, many minor issues in the manuscript –especially in the methods and in the description of the approaches used in bioinformatics analysis – that need to be addressed by the authors. Basically, it needs the careful revisiting and checking all aspects for publication.

Specific comments to each part of the manuscript:

Methods:

1. Page 6, line 126 – 127The authors stated the following statement
“RevertAid H Minus First Strand cDNA Synthesis Kit (Thermo Fisher, Waltham, MA) using oligo-dTs or random hexamer primers, following the

manufacturer's instructions.” I am wondering if the authors mean oligo-dTs AND random hexamers? The authors should clarify and add clear description of the procedure.

Response:

We sincerely apologize for this misunderstanding. We used random hexamers for RNC library construction. Oligo-dTs and random hexamers were separately used for the reverse-transcription to distinguish linear and circular form of *LINC-PINT* RNA products (Figure 3C, oligo-dTs primers for poly-A RNAs and random hexamers for all RNAs). We corrected it in the revised manuscript.

2. Page 7, line 134 “rRNA was removed to retain other types of RNA, including mRNAs and ncRNAs.” Please provide some details on how the ribosomal RNAs were removed. If a kit is used then please provide the name of the kit and company.

Response:

We sincerely apologize for not providing this information. Followed our protocol, this part has been revised as “rRNA was removed by using VAHTS Total RNA-seq (H/M/R) Library Prep Kit for Illumina (Vazyme Biotech Co.,Ltd, Nanjing, China)” in the revised version.

3. Page 7, line 135: The authors stated ‘The enriched mRNAs and ncRNAs were fragmented into short fragments in fragmentation buffer and reverse-transcribed into cDNA with random primers’ Please provide more details about the RNA fragmentation protocol. Was the RNA fragmented using MgCl₂ and heat, enzymatically or by sonication?

Response:

We sincerely apologize for not providing this information. As our protocol, this part has been revised as “The enriched mRNAs and ncRNAs were fragmented into short fragments in fragmentation buffer(MgCl₂) integrated in VAHTS Total RNA-seq (H/M/R) Library Prep Kit for Illumina (Vazyme Biotech Co.,Ltd, Nanjing, China),

then the RNA fragments were reverse-transcribed into cDNA with random primers.”

Bioinformatics:

4. Page 7, line 147: The authors stated the following “The rRNA reads removed from each sample were then mapped to a reference genome using TopHat2 (version 2.0.3.12), respectively” . If the rRNA reads were removed in the first step (lines 145 - 146), why do they have to be removed again? Did the authors mean that the mRNA reads were mapped and removed? Please also indicate the reference genome used and the version. It is not clear if the authors mean ‘reference genomes’ or ‘reference genome’ ? Also the word ‘respectively’ seems to be out of place. Finally, the authors want to indicate how the mapped reads were removed and what tools were used.

Response :

We sincerely apologize for the misunderstanding in this part. The original sentence was “The rRNA reads removed from each sample were then mapped to a reference genome using TopHat2 (version 2.0.3.12), respectively”. It has been revised to “The rRNA removed reads were then mapped to a reference genome (Human genome hg38) using TopHat2 (version 2.0.3.12)”. Reads that mapped to genomes were discarded according to the mapping information recorded in bam files using in-house perl scripts. We also added this information in the revised version. We have corrected and re-written this part to make the procedure clearer in the revised manuscript.

5. Page 7, line 150: I am wondering how the authors extracted the 20mer reads and what program did they use to align those reads to a reference genome? Was TopHat again or another aligner?

Response:

We thank the reviewer for this good question. We added the detail description in the revised manuscript. “20mers from both ends of the unmapped reads were extracted

using in-housed perl scripts and aligned to the reference genome using bowtie2 (version 2.3.0) to locate unique anchor positions within splice sites. Anchor reads that aligned in the reverse orientation (head-to tail) indicated circRNA splicing and were then subjected to find_circ (version 1.2, https://github.com/marvin-jens/find_circ/) to identify circRNAs.”

6. Page 8, line 155 –156: The authors should provide the name of the program used to annotate the identified circRNAs.

Response:

We thank the reviewer for this good suggestion. We used in-housed perl scripts and Blast. We have added this detail in the revised manuscript. “host genes of identified circRNAs were determined using in housed perl scripts according to gtf files from Ensembl database (<http://www.ensembl.org/>). CircRNAs were also blasted against circBase to determine if they have been reported”

7. Page 8, line 166: Please correct the abbreviation of correct p-value to q-value instead of Q value.

Response:

We thank the reviewer for this good suggestion. We corrected it in the revised manuscript.

8. Page 8, line 169: Please indicate the tool used for Gene ontology enrichment and use the full name followed by the abbreviation, gene ontology (GO).

Response:

We thank the reviewer for this good suggestion. We corrected it in the revised manuscript.

Results section:

9. Page 16, line 346 How is it possible that the circRNAs identified from total RNA-seq are less than those identified from the RNC-RNA-seq? Should it not be the opposite?

Response:

We thank the reviewer for this very good question. Before sequencing, we expected that circRNAs should have extremely low expression level in RNC samples.

Therefore, we designed to sequence more data in RNC samples in original experimental plan. We collected 4 times more data from RNC-seq samples compared with RNA-seq samples. This data was shown in lower table, and this table has been added to supplementary data.

Data size of samples sequencing in our study

sample	Clean Data(bp)	Q20(%)	Q30(%)
NHA-RNC	71,365,714,950	69,469,221,718 (97.34%)	66,051,111,845 (92.55%)
NHA-RNA	17,639,669,924	17,227,387,816 (97.66%)	16,488,445,529 (93.47%)
U251-RNC	86,980,594,910	84,672,062,021 (97.35%)	80,464,669,535 (92.51%)
U251-RNA	16,185,135,272	15,836,185,796 (97.84%)	15,200,074,412 (93.91%)

The sequencing results indicated that the number of middle to high expression of detected circRNAs (RPM>100) were similar or slight higher in RNA-seq samples compared with that in RNC-seq samples (shown in the figures below). However, as RNC-seq samples obtained more than 4 times sequencing data, the numbers of low expression circRNAs (RPM<100) detected in RNC-seq samples were significantly higher than the number in RNA-seq samples. This was the reason that total number of circRNAs identified in RNC-seq samples was more than that in RNA-seq samples.

The distribution of circRNAs number within different expression ranges

10. Page 16, line 346. The authors indicated that 4597 circRNAs were already known ones without giving any details about how many were identified from total RNA-seq and how many from RNC-RNA-seq. Please show a breakdown of the number and more details would benefit this part.

Response:

We thank the reviewer for this very good suggestion. As shown in the table below, a total of 15,189 circRNAs (7,016 from RNA-seq and 12,863 from RNC-RNA seq) were identified in our seq-data and 4,597 of which were matched in circBase. Among the 4,597 circRNAs, 3303 (47.08% of 7,016) and 3975 (30.90% of 12, 863) were detected from RNA-seq and RNC-seq, respectively. Comparing the results from RNA-seq and RNC-seq, the most obvious difference was low expressing (average RPM<100) circRNAs. The number of low expression of circRNAs detected from two methods was 4,568 versus 11,277, and the annotation rate from circBase was 38.55% versus 26.53%. It meant that RNC-seq have detected more novel circRNAs, probably due to higher sequencing depth than RNA-seq.

Number of high and low expression circRNAs recorded in circBase

	circRNA number	circRNA recorded in circBase	annotation rate from circBase
RNA-seq	7,016	3,303	47.08%
RNC-seq	12,863	3,975	30.90%
RNA-seq (average RPM>100)	2,448	1,542	62.99%
RNC-seq (average RPM>100)	1,586	983	61.98%
RNA-seq (average RPM<=100)	4,568	1,761	38.55%
RNC-seq (average RPM<=100)	11,277	2,992	26.53%

11. Page 17, line 353. Minor inconsistency: Please write the term followed by abbreviation between brackets, False discovery rate (FDR).

Response:

We thank the reviewer for this good suggestion. We have corrected it in the revised manuscript.

Figures:

12. Overall figure 1 needs to be reviewed and edited by the authors to

match the writing in the results section, correct inconsistencies in the numbering and the names of the figure panels. Examples: Fig. 1F, the figure legend does not match the information provided in the results section (line 354). The authors referred to the heatmap in the figure legend whereas the figure itself shows a scatter plot which matched the description in the results section (line 354). There is no mention of the scatter plot in the figure legend and there is no mention of the heatmap in the results section. Furthermore, the authors should show the units the used on both axes of the scatter plots and on the key of the heatmap. It would also be more informative if more details are provided, for example if the pseudo-count was used then please indicate that in the figure legend. Figure 1G right, and there is no left and right in figure 1G, there is only a Venn diagram unless the authors are referring to the left and right sides of the Venn diagram.

Response:

We thank the reviewer for these very good questions and sincerely apologize for previous mistakes. We corrected the description of Figure 1F in the Figure legend part (changed to scatter map). We labeled the scatter plots with the units we used in the revised Figure 1F. For the heatmap, RPKM normalized by Z score was used as the axes unit (we added it in the Figure Legend). In Figure 1F, the numbers on axes were "1e-01, 1e+01, 1e+3". They were the value per se but no pseudo-value. We added the description in the revised Figure Legend. In Figure 1G, we changed the left and right to upper and lower.

13. Generally for figures. All figures should be checked for proper labelling and size. Furthermore, the content of the figures should be clearly explained in the legends.

Response:

We thank the reviewer for these good questions. We carefully checked the labelling and the Figure Legends. Necessary further descriptions were added in the revised manuscript.

14. The sequence of the 87 amino acids peptide should be shown in one of the figures. It is shown in part but I could not find the entire sequence somewhere in the manuscript. Furthermore, an access. No. for the peptide should be obtained and added to the paper.

Response:

We thank the reviewer for this good suggestion. We showed the full-length PINT87aa sequences in the revised Figures 3D. We also applied a GenBank TPA number for PINT87aa. Nucleotide sequence data reported are available in the Third Party Annotation Section of the DDBJ/ENA/GenBank databases under the accession number TPA: BK010446. This sequence record will be held confidential until the data or accession numbers appear in print. Detail working GenBank submission is listed as follow:

LOCUS BK010446 1084 bp RNA circular PRI
15-FEB-2018
DEFINITION TPA_exp: Homo sapiens PINT87aa (circPINTexon2) gene,
complete cds.
ACCESSION BK010446
VERSION BK010446
KEYWORDS Third Party Data; TPA; TPA:experimental.
SOURCE Homo sapiens (human)
ORGANISM Homo sapiens
Eukaryota; Metazoa; Chordata; Craniata; Vertebrata;
Euteleostomi;

Mammalia; Eutheria; Euarchontoglires; Primates;

Haplorrhini;

Catarrhini; Hominidae; Homo.

REFERENCE 1 (bases 1 to 1084)

AUTHORS Zhang, M.

TITLE A Novel Peptide Encoded by Circular Form of Long Intergenic
Non-protein-coding RNA p53-induced Transcript Suppresses
Oncogenic Transcriptional Elongation in Human Malignancy

JOURNAL Unpublished

REFERENCE 2 (bases 1 to 1084)

AUTHORS Zhang, M.

TITLE Direct Submission

JOURNAL Submitted (23-JAN-2018) Neurosurgery, The 1st Affiliated
Hospital of Sun Yat-sen University, Zhongshan two Road 58,
Guangzhou,

Guangdong 510080, China

COMMENT LocalID: circ-PINT-1084[Homo

FileID:

this_circRNA_encoded_peptide_PINT87aa./circ-PINT-1084[Hom
o

Bankit Comment: TPA Submission:true.

Bankit Comment: TPA EVIDENCE:We studied the long intergenic
non-protein coding RNA, p53 induced transcript (LINC-PINT),
which was previously reported as a tumor suppressor and
connected p53 activation with polycomb repressive complex 2
(PRC2). We selected this long noncoding RNA (lncRNA) for
further analysis because LINC-PINT has a long exon 2 which
in accordance with the bioinformatical analyzed circular RNA
standard. The following immunoblotting showed 87aa peptide

level also decreased, indicating that this peptide is encoded by circPINTexon2. We name this circRNA encoded peptide PINT87aa.

Bankit Comment: BankIt2081657.

Bankit Comment: TPA Submission:true.

Bankit Comment: TPA EVIDENCE: We studied the long intergenic non-protein coding RNA, p53 induced transcript (LINC-PINT), which was previously reported as a tumor suppressor and connected p53 activation with polycomb repressive complex 2 (PRC2). We selected this long noncoding RNA (lncRNA) for further analysis because LINC-PINT has a long exon 2 which in accordance with the bioinformatical analyzed circular RNA standard. We found circular RNA circPINTexon2 translates a small protein contains 87 amino acids. We name this circRNA encoded peptide PINT87aa.

##Assembly-Data-START##

Sequencing Technology :: Sanger dideoxy sequencing

##Assembly-Data-END##

PRIMARY	TPA_SPAN	PRIMARY_IDENTIFIER	PRIMARY_SPAN
COMP	1-1084	AC013434	77297-78380
	1-1084	AK125651	159-1242
	1004-1084	AL713664	1-81
	153-1084	BC130416	1-932
	1-306	DB472406	181-486

FEATURES

Location/Qualifiers

source 1..1084

/organism="Homo sapiens"

/mol_type="genomic RNA"

gene /db_xref="taxon:9606"
479..742
/gene="circPINTexon2"
/note="formed by the circularization of exon 2 of
the long
intergenic non-protein-coding RNA p53-induced
transcript
(LINC-PINT) gene"
CDS 479..742
/gene="circPINTexon2"
/function="controls the cell cycle and cell
proliferation"
/codon_start=1
/product="PINT87aa"

/translation="MLWLPDRGSCSARSPSGMLRGAPGGWRYGRRCGRRRQSCCCCC

CSHVGAPLSFHREASLVSHDGHDIMKQHCGEESIRGAHGYKNK"

BASE COUNT 254 a 294 c 298 g 238 t

ORIGIN

1 gtcatatctc cgtacctcac ttctgacac aaacaagttt tcactgttgt
cagcaacaaa

61 gcctaatat agctgaggaa gagaaaaact gcattgcatt ttgcctcctg
caagcatcat

121 caacagttac tggaggaacg taattccaga aagcttgaaa gccgtggtga
tgtaattat

181 gatacaaatg cctggttcta tttctgttat tattgttttg tcatttctgt
tttcccagcg

241 atctgactga actgcagag ggacaaatcc agtttttctt tttgactttt
gtcaaactaa

301 atcaggcctg atagaaaact cattgctctc cgggaaaca aagtaggagc
cacgaaatgt

361 cattttaaca gagcgtgggt ttggtgactg taggaaagga tttgaggacg
ctccttctgt

421 tcggttcct atgtcatgag cacaggctcc acgcacgcac agacaccacg
gctcccggat

481 gctgtggctc cccgatcggg gctcctgcag cgccagaagc ccctccggga
tgcttcgagg

541 ggctcccggg ggggtggaggt acggacgccg ctgcggccgc cgccgccagt
cctgctgctg

601 ttgttgctgc tgcagtcacg tgggagcccc ttaagtctc catagagagg
cctctctggt

661 gtcacatgat ggacatgata taatgaaaca acattgtgga gaggaaagca
ttaggggagc

721 ccacggctac aaaaacaagt gtagtgaag aggtgggagg aagagaaact
acgccacctc

781 ccctgcagcc gtagtcacgc agcagcctgg cgtgacaagt gggcgacgcc
ggggggcagg

841 gagccggggt ccttggcct ggccggggac cccaccgcc accgcgagga
ggacaacttt

901 tagccggcag cccagaccag cgcggcacct gtctccggag tctccaccgc
tcctccgat

961 tcatcccagg gaaattctca agaatacgt ctacaaatct acgtgcgcat
cattttcacc

1021 tcgctcgcg cccgggagga aggaacgagg caaggagcta aagcagcgtg
cgttcagccc

1081 tggg

Reviewer #2, Expertise: Circular RNA (Remarks to the Author):

The manuscript by Zhang et al. showed that an endogenous circRNA generated from a long noncoding RNA might encode a regulatory peptide in human cells. The authors identified an 87-amino-acid peptide encoded by the circular form of the long intergenic non-protein-coding RNA p53-induced transcript (LINC-PINT). This peptide could suppress cancer cell proliferation in vitro and in vivo. Mechanistic studies revealed that it directly interacts with PAF1 and sequentially inhibits the transcriptional elongation of some oncogenes. Furthermore, the expression of this peptide and its corresponding circRNA were downregulated in human cancers compared with the levels in normal tissues. These results may provide novel evidence that circRNA can be translated and functions in cancer. This study is potentially interesting and at times comprehensive, however, it lacks in many aspects and is therefore not convincing.

1. The result of RNC-RNA-seq data is confusing. The authors performed RNA-seq analyses of ribosomal RNA-depleted total RNAs and ribosome-nascent chain complex (RNC)-RNAs from two cell lines. The total RNAs should include RNC-RNAs and ribosome-free RNAs. However, a total of 12,863 circRNAs were identified in RNA-RNA-seq, but only 7,017 in total RNA-seq. How about the sequencing depths? All the 12,863 circRNAs identified in RNC-RNA-seq are binding to ribosome and translational? The approach to extract RNC-RNAs is quite rough. The authors should calculate the enrichment of circRNAs in RNC-RNA-seq compared to total RNA-seq (or input).

Response:

We thank the reviewer for these very good questions. Before sequencing, we expected that circRNAs should have extremely low expression level in RNC samples.

Therefore, we designed to sequence more data in RNC samples in original experimental plan. We collected 4 times more data from RNC-seq samples compared with RNA-seq samples. Data was shown in lower table.

data size of samples sequencing in our study

sample	Clean Data(bp)	Q20(%)	Q30(%)
NHA-RNC	71,365,714,950	69,469,221,718 (97.34%)	66,051,111,845 (92.55%)
NHA-RNA	17,639,669,924	17,227,387,816 (97.66%)	16,488,445,529 (93.47%)
U251-RNC	86,980,594,910	84,672,062,021 (97.35%)	80,464,669,535 (92.51%)
U251-RNA	16,185,135,272	15,836,185,796 (97.84%)	15,200,074,412 (93.91%)

The sequencing results showed that the number of middle to high expression of circRNAs (RPM>100) detected were similar or slight higher in RNA-seq samples compared with the number in RNC-samples. However, as RNC-seq samples obtained more than 4 times sequencing data, the numbers of low expression circRNAs (RPM<100) detected in RNC-seq samples were significantly higher than the number in RNA-seq samples. This was the reason that total number of circRNAs identified in RNC-seq samples were more than that in RNA-seq samples (see below).

Supplementary Fig1A. The distribution of circRNAs number within different expression ranges

We can't get the conclusion that all 12,863 RNC-seq identified circRNAs are all translatable. Although RNC-seq provided the initial clue of coding circRNAs, further validations are strictly required as we showed in our study. Also, as the reviewer pointed out, RNC-seq is a screen procedure and may provide false positive data and this is the reason we cross-matched RNC-seq results with RNA-seq results and verified the initial results with further experiments.

The RNC purification procedure we used was published several times (Nucleic Acids Res. 2018 Jan 4;46(D1): D206-D212; J Proteome Res. 2017 Dec 1;16(12):4446-4454; Nucleic Acids Res. 2013 May;41(9):4743-54). Precisely capturing the translating RNA is still a technical problem in now days. Pamuduriti et al. used ribosome footprinting(RFP) and identified the translational potential of circMbl in fly heads (Mol Cell. 2017 Apr 6;66(1):9-21.e7. doi: 10.1016/j.molcel.2017.02.021). They found that the RFP pattern around the stop codon of circMbl, which strongly suggested the coding ability. In our RNC-seq, the reads were also enriched around the sORF stop codon in LINC-PINT exon 2 (see next question, the IGV plot).

We also searched normal brain/brain tumor ribosome-seq data published by other groups to validate our results. Gonzalez C et al. performed ribosome profile in 3 normal brain tissues and 2 brain tumor tissues (Journal of Neuroscience, 2014, 34(33): 10924-10936. **GSE51424**). We downloaded their data and analyzed the RPM of exon 2 of LINC-PINT, as shown below.

Except sample 1, normal brain tissues had a similar RPM on exon2 of LINC-PINT compare with NHA in our RNC-seq (RPM of above samples: 0.44, 1.24, 2.61, 0.92, 0.71). The RFP reads were around the sORF stop codon of exon 2 LINC-PINT.

As above data had a small data-size (only 5 samples), we further searched some larger data-set. Loayza-Puch F et al. reported RFP in 9 normal kidney and 17 renal

carcinomas (Nature, 2016, 530(7591): 490 **GSE59821**). The PRM on exon 2 of LINC-PINT was shown below:

The medium RPM of normal vs. tumor was 4.11 vs. 1.55 (mean: 3.483 ± 2.193 vs. 2.626 ± 2.80 , $p=0.43$). Although there's no statistical significance (may due to the sample size or variation), the RPM strongly suggested the coding potential. Also, The RFP reads were around the sORF stop codon of exon 2 LINC-PINT.

We also analyzed the breast cancer cell line data in the same paper. As shown below, the MCF10A, MCF7 and T47D had differential RPM on exon 2 of LINC-PINT.

Median RPM of MCF10A, MCF7 and T47D: 0.222, 0.112, 0.062

Mean RPM of MCF10A, MCF7 and T47D: 0.2697 ± 0.143 , 0.1006 ± 0.099 ,

0.06203 ± 0.0109 , $p < 0.05$). The breast cancer cells' RFP indicated that MCF10A has higher RPM on exon 2 of LINC-PINT than that of MCF7 and T47D. This result was similar with our RNC-seq (note that breast tissue has few LINC-PINT expression, Figure 4B) and the RFP reads were still around the sORF stop codon of exon 2 LINC-PINT.

Followed the reviewer's suggestion, we provided the enrichment efficiency of circRNAs in RNC-seq to RNA-seq and the data was shown as below (revised Figure 1E). As shown in the Venn plot, 62.53% (3051 of 4879 circRNAs detected in NHA RNA-seq data) and 50.12% (2038 of 4066 circRNA detected in U251 RNA-seq data) of RNA-seq identified circRNAs were detected in RNC-seq data. 32.22% (3051 of 9451 circRNAs detected in NHA-RNC-seq) and 34.01% (2038 of 5992 circRNAs detected in U51-RNC-seq) of RNC-seq identified circRNAs were detected in RNA-seq data. Lots of circRNAs were only observed in RNC-seq data due to larger data size acquired in RNC-seq, especially lower expressed circRNAs (see above response).

Revised Fig1E. The Venn plot of number of circRNAs detected in RNA-seq or RNC-seq

2. Figure 1 should show the comparison of RNC-RNA-seq and total RNA-seq

data, but not just the differently expressed circRNAs between two cells. Fig. 1E is no different and too small, and may be deleted.

Response:

We thank the reviewer for this very good suggestion. As we mentioned in previous question, we provided the Veen plot to show the comparison of RNC-seq and RNA-seq results and this data was added to the revised Figure 1E. Also, we deleted the originate Figure 1E in the revised Figure 1.

3. The existence and expression of circPINTexon2 from LINC-PINT should be carefully checked. Although this circRNA was annotated in circBase, there is no score for it. The authors claimed that this circRNA is highly expressed in 293T cells, how many copies per cell or delta CT (compared to Actin) for LINC-PINT and circPINTexon2? Which one is with higher expression level?

Response:

We thank the reviewer for this very good question. We used several experiments to confirm the existence of circPINTexon2. Firstly, circPINTexon2 junction reads were identified in RNA-seq and RNC-seq. The junction reads only targeted the head-to-tail splicing of designated circRNA (circPINTexon2), so these reads numbers were specific for circPINTexon2. Besides, other reads from exon 2 of LINC-PINT (may also from circPINTexon2) were shown in IGV (see next question). Next, Northern blot and FISH showed that circPINTexon2 was easily detectable in cells, although the expression level of circPINTexon2 was lower than the expression of linear LINC-PINT. In Figure 4B, we used absolute q-PCR to show that circPINTexon2 was ubiquitously expressed in several human tissues. The expression of circPINTexon2 is higher in brain, liver, kidney and stomach but lower in breast, intestine, thyroid and pancreas. All tissues showed higher expression level of LINC-PINT than the expression of circPINTexon2. Specifically, there were approximate 200 circPINTexon2 copies and more than 300 LINC-PINT copies in 200pg brain tissue total RNAs. We used absolute q-PCR to detect circPINTexon2 and LINC-PINT in

293T cells. As shown in revised Supplementary Figure 2D, there were around 150 copies of circPINTexon2 in 200pg total 293T RNAs, compared with around 250 copies of LINC-PINT. From all our results, LINC-PINT has a higher expression level than that of circPINTexon2. Taking above evidences together, we concluded that CircPINTexon2 is not an extremely low expressed circRNA. The reason circPINTexon2 has no score in circBase may be due to the different sequencing depth and high GC-rich junction region, which we have mentioned in the discussion part.

4. Figure2A should provide the sequencing reads information from IGV, which could show the real circRNA and host RNA expression level. The name of host RNA in Figure 2A and 2B is not consistent. The primers for host RNA are not correct (do not locate in Exon1 and Exon2).

Response:

We thank the reviewer for this good suggestion. we have provided IGV plot for the expression level of the second exon of host gene (*LINC-PINT*) below.

Revised Figure 2A. IGV plot of LINC-PINT exon 2 reads in RNA and RNC sequencing.

IGV plot of normal mapping reads within LINC-PINT

IGV program was invented much earlier before deep-sequencing was used in discovering circRNAs. The head-to-tail reads numbers that represent circRNAs could not be drawn by IGV program. Thus, we only showed the junction reads numbers of circPINTexon2 in previously showed Figure 2A. We added the IGV plot of exon 2 in the revised Figure 2A, lower panel.

Compared with LINC-PINT exon 2 reads, circPINTexon2 junction reads were much lower (as shown in Figure 2A, upper panel). However, those non-junction reads could not distinguish between circPINTexon2 and linear LINC-PINT, as junction reads represent only circPINTexon2 but non-junction reads may be from both circRNA and linear RNA.

We think that if the junction reads were captured (not extremely low), the existence of potential circRNA was highly possible. But absolute real-time PCR using specific primers should be used to identify the circRNA and its related linear RNA's level.

We have changed the host gene names in Figure 2A from Ensembl numbers to transcripts names. LINC-PINT has several transcripts and only LINC-PINT-208 which contained the longest exon2 formed circPINTexon2.

We checked our q-PCR primers in previous Supplementary Table 1, the reviewer may see following primers:

PINT (Divergent primer)	AGGAACGAGGCAAGGAGCTA	TGCAGGAGGCAAAATGCAAT
PINT (convergent primer)	CAGTCCTGCTGCTGTTGTTG	GGCTCCCCTAATGCTTTCCT

The convergent primer above was not the cross exon 1 and 2 primer we used in the manuscript. These primers were located inside exon 1 and we only used them in the beginning of this research. We forgot to delete them before the submitting.

There's another pair of primers named "convergent primer new" in the previous Supplementary Table 1 (below the PINT (convergent primer)):

convergent primer new	GGGCTTGGCAGCAGAAGGCA	AGGTACGGACATATGACCTCTC
----------------------	------------------------

These primers were what we really used. The downstream primer located just in the exon 1 and exon 2 junction, which is specific for linear LINC-PINT. We sincerely apologizing for the misunderstanding caused by our unclear labeling. We replaced the "PINT (convergent primer)" with "convergent primer new" in the revised Supplementary Table 1.

5. The coding peptide could also be generated from LINC-PINT. The coding region for this peptide is located in both LINC-PINT and circPINTexon2. Although LINC-PINT is shown in nuclear, it is still present in cytoplasm from ENCODE RNA-seq data. Does introduction of LINC-PINT expressing vector produce the peptide?

Response:

We thank the reviewer for this very good question. To find the cytoplasm data of LINC-PINT, we searched the ENCODE database and found 7 human samples included the cytoplasmic and nucleus sequencing data

(<https://www.encodeproject.org/matrix/?type=Experiment&searchTerm=cytoplasm>).

Experiment Matrix

Click or enter search terms to filter the experiments included in the matrix.

cytoplasm

Organism
Homo sapiens 4
Drosophila melanogaster 3

Biosample type
 immortalized cell line 7

Organ
 blood 2
 liver 2
 gonad 1

Project
 ENCODE 4
 modENCODE 3

Genome assembly (visualization)
 GRCh38 4
 hg19 4

Lab
 Eric Lécuyer, IRCM 6
 Susan Celniker, LBNL 1

Audit category: missing documents 4

Audit category: missing external identifiers 3
 low read depth 2
 missing spikeins 2
 missing genotype 1

Assay
 polyA RNA-seq 4
 total RNA-seq 3

Assay category
 Transcription 7

Date released
 April, 2016 4
 March, 2014 3

Available data
 fastq 7
 bam 4
 bigWig 4
 tsv 4

7 results

immortalized cell line

Cell Line	polyA RNA-seq	total RNA-seq
HepG2	1	1
K562	1	1
ML-DmD17-c3	1	1
OSS	1	

Download Visualize

We further analyzed these samples which were from human HepG2 and K562 cells:

No	Cell line	Sub-cellular location	strategy	ID
1	HepG2	cytoplasm	Total RNA	ENCSR813BDU
2	K562	cytoplasm	Total RNA	ENCSR696YIB
3	HepG2	cytoplasm	ployA	ENCSR019MXZ
4	K562	cytoplasm	ployA	ENCSR594NJP
5	HepG2	nucleus	Total RNA	ENCSR061SFU
6	K562	nucleus	Total RNA	ENCSR040YBR

7	HepG2	nucleus	ployA	ENCSR058OSL
8	K562	nucleus	ployA	ENCSR530NHO

We blasted the LINC-PINT sequences in above samples, and the results showed as below:

We found that LINC-PINT expression was very low in the upper four sequencing data (all from cytoplasm), no matter using PolyA and total RNA strategy. In contrast, LINC-PINT signal was obvious in the sample 5, 6 and 8 which were from nucleus. In sample 7, LINC-PINT also showed low expression. But considering sample 7 used PolyA sequencing, the enrichment efficiency maybe lower than that of total RNAs (sample 5). Taking above information together, LINC-PINT generally showed low expression in the cytoplasm but enriched in the nucleus. Those low-reads from cytoplasm may come from the contamination during the cytoplasm and nucleus isolation.

Besides, the RNC-seq reads number of the exon 2 LINC-PINT were much higher than the number of exon 1 and exon 3 (see above IGV plot). If LINC-PINT could be translated, the exon 1 and exon 3 should have similar reads to exon 2.

To further respond to the second question, we used the LINC-PINT overexpression plasmid as shown in previous Figure 2F, lower panel. As verified by FISH, introduction of the plasmid increased the LINC-PINT expression in 293T cells. We used q-PCR to detect LINC-PINT level quantitatively, as shown in revised Figure 3H. Introduction of LINC-PINT expression vector in hNSC and 293T cells successfully up-regulated LINC-PINT RNA expression, respectively. However, the expression level of PINT87aa in LINC-PINT overexpressed hNSC and 293T cells was not changed (Revised Figure 3H, upper panel). Together with our previous data that using LINC-PINT specific ASO did not change PINT87aa level, we conclude that PINT87aa was not encoded by LINC-PINT but by circPINTexon2.

According to our experience, we found that most coding circRNAs were generated from coding exons (unpublished data). Few of them has a cross-junction ORF, which could translate some unique novel peptides, as we previously published FBXW7-185aa and SHPRH-254aa (J Natl Cancer Inst. 2018 Mar 1;110(3). doi: 10.1093/jnci/djx166; Oncogene. 2018 Mar;37(13):1805-1814. doi: 10.1038/s41388-017-0019-9.). However, most of circRNAs translated peptides have the same amino acids as part of their host gene's products. Cross-junction ORF encoded peptides are much easier to distinguish from their host gene's products compare with non-cross-junction ORF encoded peptides, which was the reason we chose non-coding host RNA in this study.

6. The authors used GFP fusion protein labeling and found this peptide was concentrated in nucleus. Does this peptide contain nuclear location signal?

Response:

We thank the reviewer for this good question. We have searched the PINT87aa sequences before and found no nuclear localization signal. We assumed that

PINT87aa may interact with some nuclear proteins (such as PAF1) and co-transported into the nucleus. The mechanisms how PINT87aa enter the nucleus are still under investigation in our lab.

7. Why Figure 7A have two clear band in PINT87aa WB result in first panel but not others?

Response:

We sincerely apologize for this misunderstanding. We showed the uncropped images of Figure 7A as below. PINT87aa in the first panel also showed a single band at 10kD. As the bands shape was curved, we cannot exclude all the non-specific bands larger than 10Kd in order to keep the enough spaces besides the target band. This antibody worked well in our hands, as all other cell lysates WB showed a single band. However, clinical samples (tissues) or long-exposure sometimes showed some nonspecific bands around 17kD (also seen in Figure 4C, tissue wb; Supplementary Figure 6).

8. Some obvious errors: A duplicated construct was shown in Figure 3A. has circ_0082389 should be hsa_circ_0082389.

Response:

We sincerely apologize for these mistakes. We had them corrected in the revised Figure 3A and revised manuscript.

Reviewer #3, Expertise: Transcriptional regulation of gene expression (Remarks to the Author):

The MS entitled “A Novel Peptide Encoded by Circular Form of Long Intergenic Non-protein-coding RNA p53-induced Transcript Suppresses Oncogenic Transcriptional Elongation in Human Malignancy” investigates the importance of PINT87 peptide encoded by CircRNA of LINC-PINT ncRNA in human malignancy. The work done in the MS is interesting and is of importance in context of understanding the role of circular RNA’ s in governing cellular functions, however, there needs to be detailed explanation of some data and to perform more experiments.

Major Comments:

1. FISH experiments shown in Figure 2 describes the differential localisation of linear and circular RNA, with the later being cytoplasmic and former nuclear. Then the experiments performed in Figure 3 shows only CircRNA is translated to give PINT87aa and not the linear form. Further, when CircRNA was overexpressed as a GFP fusion, its localisation was only nuclear. So, considering all the data is it right to assume that in vivo also only Circular version is translatable, due to its presence in cytoplasm and not the linear form? If so, then what could be the factors distinguishing these two and enabling a differential location of these RNA’ s.

Response:

We thank the reviewer for this good question. We provided the IGV plot of the LINC-PINT exon 2 from NHA and U251 cells (see below). Exon 2 of LINC-PINT

has a higher reads number compared with exon 1 and 3. If LINC-PINT was translatable, exon 1 and 3 should have similar reads numbers as exon 2 (Non-junction reads from circPINTexon2 were default considered as linear reads, which may cause the high reads number in RNC-seq).

Revised Figure 2A. IGV plot of LINC-PINT exon 2 reads in RNA and RNC sequencing.

IGV plot of normal mapping reads within LINC-PINT

To further address that PINT87aa was encoded by circPINTexon2 rather than LINC-PINT, we transfected 293T cells with LINC-PINT expression vector (we used in original Figure 2F). Introduction of the overexpression plasmid only increased the

LINC-PINT, as showed in revised Figure 3H. Next, total cell lysates were subjected to western blot. The PINT87aa expression level did not increase (Revised Figure 3H). Together with our previous data that ASO targeting LINC-PINT could not reduce PINT87aa expression, we concluded that PINT87aa was not encoded by LINC-PINT.

To our knowledge, most of the circRNAs from exon localized in the cytoplasm where they may function as miRNA, protein decoys, transporter or scaffolds (Oncogene. 2018 Feb 1;37(5):555-565. doi: 10.1038/onc.2017.361; RNA. 2013 Feb;19(2):141-57. doi: 10.1261/rna.035667.112; PLoS One. 2012;7(2):e30733. doi: 10.1371/journal.pone.0030733). However, some circRNAs, formed by intron and exon and called ElciRNAs, are reported to be only located in nucleus (Nat Struct Mol Biol. 2015 Mar;22(3):256-64. doi: 10.1038/nsmb.2959). We also noticed that one circRNA generated from LncRNA ANRIL (circANRIL) was reported to localize in cytoplasm while its host linear form ANRIL primarily localized in nucleus (Int J Mol Sci. 2017 Jun 27;18(7). pii: E1378), quite similar to LINC-PINT and circPINTexon2. However, another group reported that circANRIL localized in nucleolar (Nat Commun. 2016 Aug 19;7:12429). During the revision, we noticed that a recent report showed circRNA nucleus localization may decide by its length. Huang et al. showed that depletion of evolutionary conserved UAP56 or URH49 protein caused long and short circRNAs, respectively, to become enriched in the nucleus (Genes Dev. 2018 May 17. doi: 10.1101/gad.314856.118. [Epub ahead of print]). As CircPINTexon2 is 1084nt (middle length), its localization may be decided by both factors mentioned above. This is the only report that we found to underlie the circRNAs cellular localization mechanism. So far, FISH and ISH assay are still the best ways to find the cellular localization for each circRNA.

2. GFP-PINT87aa showed nuclear localisation, is this because this contains nuclear localisation signal or it interacts with other proteins like PAF1 in this case to get localised inside nucleus?

Response:

We thank the reviewer for this good question. We did not find nuclear localization signal sequences in PINT87aa. Based on the nuclear localization of PINT87aa and its several many nuclear binding proteins, we assumed that PINT87aa is transported into nuclear through interacting with them. Further investigations are already launched in our lab.

3. GFP-PINT87aa seems to also show a punctate pattern of localisation inside nucleus, as seen clearly in Figure 6E. What are these puncta, could they be aggregates, especially considering in the mass spec data a lot of proteins seem to be either splicing factors or ribonucleoproteins, which are known to be aggregates it becomes important to understand this subnuclear localisation?

Also, GFP-PINT87aa can also be seen in cytoplasm in U87 cells and not in U251. Why this discrepancy?

Response:

We thank the reviewer for this good question. As we showed in our Mass spectra and immunoprecipitation, PINT87aa interacted with PAF1. In the revised manuscript, ChIP assay also confirmed that PINT87aa and PAF1 were colocalized in downstream genes transcription region. PAF1 is a well characterized nucleoplasm protein that formed a complex with RNA Pol II and regulates transcription elongation as well as histone modifications including ubiquitylation and methylation (Cell. 2015 Aug 27;162(5):1003-15. doi: 10.1016/j.cell.2015.07.042; Science. 2017 Sep 22;357(6357):1294-1298. doi: 10.1126/science.aan3269). Based above information, we assumed that PINT87aa may also be involved in the RNA Pol II complex and this could explain the punctate pattern of the PINT87aa signal.

We also noticed that PINT87aa may interact with splicing factors or nuclear ribonucleoproteins, as the mass spectra data showed. Reports also showed that PAF1 complex could interact with some nuclear ribonucleoproteins or splicing factors (Nucleic Acids Res. 2017 Jul 7;45(12):7180-7190. doi: 10.1093/nar/gkx321.; Mol Cell. 2005 Oct 28;20(2):225-36), which also can explain the PINT87aa punctate

pattern. Further investigations are still needed to confirm the relationship of PINT87aa with those nuclear ribonucleoproteins or splicing factors.

About the different cellular localization pattern of PINT87aa in U87 and U251 cells, we believed that it was caused by transfection efficiency. We used the same confocal settings in these different cells, which may have different PINT87aa overexpression level due to the transfection efficiency. Overdose of PINT87aa expression may cause PINT87aa signal outside the nucleus. In the revised manuscript, we changed these cell lines to BTIC (brain tumor initiating cells) and re-performed the experiments. 456 and 4121 BTIC cells have very low levels of endogenous PINT87aa expression. Overexpressing PINT87aa-GFP in above cells by using lenti-virus, we also observed the nucleoplasm localization of PINT87aa (Revised Figure 4A).

4. For the interaction of PAF1 and PINT87aa, it is clear 150–300 aa of PAF1 is important, but what are the regions in PINT87aa which are important? May be revisiting the molecular docking might give clues.

Response:

We thank the reviewer for this good question. Followed the suggestions of the reviewer, we revisited the molecular docking data. In PINT87aa, R20, G21, P23, C32, R36 and S53 are critical for PAF1 interaction. As PINT87aa is a small peptide, we chose to do mutation rather than truncation to test the binding affinity. As shown in revised Supplementary Figure 9, when above mentioned amino acids were mutated, PINT87aa could not pull down PAF1. Reversely, PAF1 also could pull down wild type PINT87aa (revised Figure 6C). These data suggested that amino acid R20, G21, P23, C32, R36 and S53 decided PINT87aa-PAF1 interaction.

5. Authors describe that PINT87aa interaction with PAF1 is the reason behind the phenotypic effects observed. However, ChIP experiments need to be performed to show that PINT87aa interaction with PAF1 is responsible to low transcription of the later genes. Further, mechanistic insights

are needed to understand what PAF1-PINT87aa interaction does.

Response:

We thank the reviewer for this very good suggestion. To further address this concern, we did more experiments to investigate the molecular mechanism. In revised Figure 6F, the secondary ChIP assay confirmed that PINT87aa is co-localized in the same region of the CPEB1 promoter as PAF1-Pol II complex. PAF1 complex has critical roles in gene transactivation. After PAF1 depletion, the CTD kinase SEC is recruited and subsequently phosphorylates Pol II on serine 2, which facilitates the release of paused Pol II into productive elongation (Cell. 2015 Aug 27;162(5):1003-15. doi: 10.1016/j.cell.2015.07.042). Other reports showed that PAF1 is overexpressed in a subset of human malignancies and achieves multiple roles in carcinogenesis other than promoter proximal pause, such as cell cycle regulation, histone covalent modification and MAPK signaling (Cancer Res. 2018 Jan 15;78(2):313-319. doi: 10.1158/0008-5472.CAN-17-2175). Based on the above evidences, we checked PAF1 expression in circPINTexon2 overexpressed 456 and 4121 cells. Results showed that PINT87aa overexpression did not change PAF1 level. Instead, ChIP assay showed that PINT87aa overexpression enhanced PAF1 and its target genes promoter interaction (Revised Figure 6G). Further investigation showed that overexpression PAF1 in 293T cells, or knocking down PINT87aa in hNSC, could enhance or attenuate PAF1/DNA interaction, respectively. From these information, we assumed that PINT87aa may work as an “anchor” and keep PAF1 complex on target genes promoter, which sequentially pauses Pol II induced mRNA elongation. Loss of PINT87aa or overexpression of PAF1, which has been seen in many cancers, results in PAF1 losing its DNA localization. “Freed” PAF1 sequentially involved in many other biological processes such as cell cycle regulation, histone modification, MAPK signaling transduction as well as cancer-stem cell self-renewal. Our data suggested that PINT87aa is critical for keeping PAF1 in the “right place” and loss of PINT87aa could enhance PAF1 induced malignant transformation.

Minor comments:

Figure 1 legend, in description of panels B and C it should be top and bottom instead of left and right.

Response:

We sincerely apologize for previous wrong labeling. We had them corrected in the revised Figure 1.

1. The readability of heat maps in Figure 1 F should be increased.

Response:

We sincerely apologize for these low-resolution images. We have changed this heat map to a higher resolution version in the revised Figure 1F.

2. The quality of gel pics in Figure 3 D and 6B is of questionable quality.

Response:

We sincerely apologize for these low-resolution images. We increased the gel pictures' resolution in Figure 3D and Figure 6B. Specifically, we re-took the pictures of the gel in Figure 6B, as shown in the revised Figure 6. The gel in Figure 3D was discarded, since we used it in the beginning of this project (antibody preparation) which is about 2 years ago. We repurified GFP-PINT87aa as previously done and replaced the gel picture, as shown in revised Figure 3D.

3. What are the other bands in in IP lane, abundance of which are more than the bait.

Response:

We sincerely apologize for this misunderstanding caused by our unclear labeling. In the revised Figure 6C, we correct the labeling.

4. Figure 4 F needs to be referred alongside Figure 5A in the text.

Response:

We thank the reviewer for this good suggestion. We re-arranged the Figures according to the text in the revised Figure 4 and Figure 5. Figure Legends were also corrected.

5. Does the PAF1 IP pull down PINT87aa?

Response:

We thank the reviewer for this good question. To address this concern, we performed the reverse IP shown in revised Figure 6. The result that PAF1-HA full length and PAF1 140-300-HA could pull down PINT87aa, further confirmed the interaction of PINT87aa and PAF1. Furthermore, R20, G21, P23, C32, R36, S53 mutated PINT87aa (determined by molecular docking) could not be pulled down by PAF1, suggesting that these amino acids decided the PAF1 binding.

6. What are the levels of linear form in the glioma tissues.

Response:

We thank the reviewer for this good question. In supplementary Figure 13 we detected the expression level of LINC-PINT in several glioma tissues. LINC-PINT showed lower expression in glioma tissues, than that in normal brain tissues or their paired adjacent normal tissues.

Reviewer #4, Expertise: GBM, mouse models (Remarks to the Author):

MAJOR CRITICISM:

One of the key hurdles for the development of successful anti-tumor therapeutics is remarkably high level of heterogeneity that is apparent in solid tumors. Glioblastoma is no exception to that rule - multiple subtypes have been described, and they can co-exist within individual

tumors. Some subpopulations are invasive, some are not; some are more differentiated, some retain stem cell characteristics. High heterogeneity is reflected both at molecular and phenotypic levels. Thus, to faithfully model pathobiology of glioblastoma it is imperative to use multiple, patient-derived cell populations. Unfortunately, authors of presented manuscript failed to meet this crucial requirement. They used barely one or two established cell lines, cultured in high serum, and non-malignant astrocytes as a control (please see specific criticisms below). This in my opinion disqualifies the manuscript for being accepted in Nature Communications. It is really unfortunate, as the subject of the study

is really timely and informative. Authors developed and used appropriate methodology. I am positive that studying novel class of ncRNA, namely circRNAs in the context of heterogeneous disease is promising and interesting area of research.

Response:

We thank the reviewer for this very good question. In the revised manuscript, we used patient derived brain tumor initiating cells (BTIC) to repeat all experiments previously done by using U87 and U251. Human neural stem cell (hNSC) were used as non-malignant control. Similar results were obtained by using these cells. New data were provided in the revised manuscript and figures (please see following point by point response).

SPECIFIC CRITICISMS

1) The major flaw of the manuscript however is use inadequate, outdated model. It is simply unacceptable these days to model such heterogeneous disease as glioblastoma using one established cell line for initial screen (and merely two in phenotypic/molecular studies). U87 and U251 glioma cell lines used in most experiments have huge number of indels, copy number

variation due to long term cultures serum (PMID - 28074274). They do not recapitulate GBM subtype heterogeneity, that do not recapitulate critical phenotypes (e. g. are not invasive *in vivo*), they were cultured in 10% FBS for countless generations. Besides, U87 are not what they were thought to be

(<http://www.nature.com/news/venerable-brain-cancer-cell-line-faces-identity-crisis-1.20515>).

Response:

We thank the reviewer for this very good question and we fully understand the reviewer's concern. To address this concern, we used brain tumor initiating cells (BTIC) 456, 4121, 387, H2S (Nature. 2017 Jul 20;547(7663):355-359; Nat Neurosci. 2017 May;20(5):661-673; Nat Med. 2017 Nov;23(11):1352-1361) which were kindly provided by Dr. Jeremy N. Rich to replace glioma cell lines in this study. Human neural stem cells (hNSC) purchased from Gibco (A15654) were used as normal control. All *in vitro* and *in vivo* experiments related to U87 and U251 or other GBM cell lines were re-done by using above cells in the revised manuscript. Generally, BTIC have low or non-detectable PINT87aa expression while hNSC have abundant PINT87aa expression. Overexpressed PINT87aa in BTIC could reduce the malignant phenotype both *in vitro* and *in vivo*, determined by several functional or mechanistic experiments. Detailed experiments and results were shown in each of the revised manuscript and figures. In figure 2E and 2F, NHA and GBM cell lines were replaced by hNSC and BTICs. In figure 3G and H, NHA were replaced by hNSC. In figure 4A, C, D, NHA and GBM cell lines were replaced by hNSC and BTICs. In addition, in figure 5, all functional experiments were repeated by BTICs. In figure 6A, E, F, NHA and GBM cell lines were replaced by hNSC and BTICs. In figure 7C, all animal experiments were repeated by BTICs.

We also searched normal brain/brain tumor ribosome-seq data published by other groups to validate our RNC-seq results. Gonzalez C et al. performed ribosome footprinting profile (RFP) in 3 normal brain tissues and 2 brain tumor tissues (Journal

of Neuroscience, 2014, 34(33): 10924-10936. **GSE51424**). We downloaded their data and analyzed the RPM of exon 2 of LINC-PINT, as shown below.

Except sample 1, normal brain tissues had a similar RPM on exon2 of LINC-PINT compare with NHA in our RNC-seq (RPM of above samples: 0.44, 1.24, 2.61, 0.92, 0.71). The RFP reads were around the sORF stop codon of exon 2 LINC-PINT, as reported by Pamuduriti et al. who used ribosome footprinting(RFP) and identified the translational potential of circMbl in fly heads (Mol Cell. 2017 Apr 6;66(1):9-21.e7. doi: 10.1016/j.molcel.2017.02.021).

As above data had a small data-size (only 5 samples), we further searched some larger data-set. Loayza-Puch F et al. reported RFP in 9 normal kidney and 17 renal carcinomas (Nature, 2016, 530(7591): 490 **GSE59821**). The PRM on exon 2 of LINC-PINT was shown below:

The medium RPM of normal vs. tumor was 4.11 vs. 1.55 (mean: 3.483 ± 2.193 vs. 2.626 ± 2.80 , $p=0.43$). Although there's no statistical significance (may due to the sample size or variation), the RPM strongly suggested the coding potential. Also, The RFP reads were around the sORF stop codon of exon 2 LINC-PINT.

We also analyzed the breast cancer cell line data in the same paper. As shown below, the MCF10A, MCF7 and T47D had differential RPM on exon 2 of LINC-PINT.

Median RPM of MCF10A, MCF7 and T47D: 0.222, 0.112, 0.062

Mean RPM of MCF10A, MCF7 and T47D: 0.2697 ± 0.143 , 0.1006 ± 0.099 , 0.06203 ± 0.0109 , $p < 0.05$). The breast cancer cells' RFP indicated that MCF10A has higher RPM on exon 2 of LINC-PINT than that of MCF7 and T47D. This result was similar with our RNC-seq (note that breast tissue has low LINC-PINT expression, Figure 4B) and the RFP reads were also around the sORF stop codon of exon 2 LINC-PINT.

Above information was like the result we acquired from RNC-seq (see the IGV plot in response to question 10), which implied the coding potential of circPINTexon2 and the differential expression pattern in normal and cancer cells.

2) Use of astrocytes as non-malignant control is very controversial. Optimal would be comparing non-malignant neural cells vs. GBM stem cells, both in serum-free culture.

Response:

We understand the reviewer's concern. To address this point, human neural stem cells (hNSC) purchased from Gibco (A15654) were used as non-malignant control in the revised manuscript. Only the early passages of hNSC were used in our experiments and new data were provided in the revised manuscript and figures accordingly.

3) In vivo study: apart from using wrong cells to model GBM, the results on Fig. 7C open several eyebrows raising questions: 1) authors claim to sacrifice mice after 70 days and experimental group is seemingly tumor-free. Yet quite shortly after substantial portion of them died. 2) control tumors kill mice in quite surprisingly wide time window (between day 40-70 or 60-90); this never should be the case for control cells. 3) Authors use really high number of cells - half a million; yet controls lived until 70-90 days. Such high number of cells would kill controls within 20 days. At least U87 I used to work with.

Response:

We understand the reviewer's concern. In the revised manuscript, we used the 456 and 4121 BTIC to repeat the in vivo model. 2000 cells were used for intracranial injection and the survival proportion was calculated in both PINT87aa overexpressed cells and the control cells. New results were provided in the revised manuscript.

4) Fig. 2F and Supp Fig 2A - Authors showed the expression of circPINTexon2 and LINC-PINT are mostly cytoplasmic and nuclear respectively using junction-specific probes specific probe and w linear specific probes specific to LINC-PINT. But the expression of translated protein are mostly nuclear (Fig 4A and Fig. 6E) shown only by IF. It is not well addressed, why the proteins are translocated to nucleus even though host gene are mostly cytoplasmic. The author should check the protein level using cell fractionating assay.

Response:

We thank the reviewer for this good question. To address this point, we isolated cell protein fraction from 293T cells and hNSC. Followed western blot indicated that PINT87aa was mainly localized in nuclear instead of cytoplasm, which is consistent with the IF results (Revised Supplementary Figure 2C). Based on the evidence we have, we think PINT87aa was translated in the cytoplasm and then transported into the nucleoplasm by interacting with some nuclear localized proteins such as PAF1.

5) Line 362 - It is not clear that if the authors have found only 10 non-coding host genes or if they have only selected 10 genes selected out of many? Out of these 10 nc host genes, 5 of them have unique peptide coding potential. It is not well addressed why authors exclude the other 4 (TTN-AS1, CDKN2B-AS1, FIRRE, and XIST from Supplemental Table 3) and chose LINC-PINT for further evaluation? "As mentioned previously, most circRNAs shared the same CDS with their host protein-coding genes. To exclude false-positive data from further investigation, we focused only

on non-coding host genes among these 274 candidates. A total of ten non-coding host genes were selected (Supplementary Table 2), and among them was LINC-PINT (ENSG00000231721).” The authors intended to choose to focus on non-coding host genes to rule out the possibility of translating proteins from circRNAs which are backspliced from CDS of host gene. But there 4 more long non-coding RNA listed in Supplemental Table3. Since these 4 lncRNAs also have coding potential, the authors should address in detail why they chose to exclude these 4 lncRNAs.

Response:

We thank the reviewer for this good question. As the reviewer mentioned, we chose our candidates that were reported as non-coding host genes to rule out false positive results (only 10 ncRNAs were found out of the 274 host genes in the cross-matched data). 5 of 10 selected non-coding genes have coding potential. We previously also analyzed the potential ORFs in the other 4 genes as below:

TTN-AS1-ORF

ATGATATATGTGCCAGTGTCACTTCTCTTACTATCAACAATAGTCACTGTG
GATTTCTTAGGGACATTTTCAATGGTAATTCTTTTGTCTGCTTCAGAATCA
TATCAGCCTTTGTCCAAGTTATTTTAGGTTTCAGGTTTTCCGGTTACGGTGG
CAGGAAGTTCAATCTTGGTCCCAGCTTTTACAGTGAGACCAGCAAGGAGC
TTCACATCGAGGAAGATTTCTGGGGCCTCTGAATTGGAAAAGATTATTTAT
GATGTTATCAAGTTCAAGCAGATTTAA

TTN-AS1-93aa

MIYVPVSLLLLSTIVTVDFLGTFSMVILLSCFRIISAFVQVILGSGFPVTVAGSSI
LVPAFTVRPARSFTSRKISGASELEKIIYDVIKFKQI

CDKN2B-AS1-ORF

ATGACTTTCTTTGTGGTAGTTAGGGTGTGGTATGTGCCACTGAGGCCACACA
CCTATTGCTGCAATTTATAGCACTGATCTGTCATCAATACCACTTGCTGTC
TTGGATGTGAAGATGATTTTTCTGCAGGGATTCCCTCTACAAAATTA
ACACTGGGCATGTGGAAATAA

CDKN2B-AS1-57aa

MTFFVVRVWYVPLRPTPIAAIYSTDLSSIPLAVLDVKMIFPAGIPSTKLKTLG
MWK

FIRRE-ORF

ATGTACACCATCATCAACGGGCCAGCAAGTTGGTCGCGCAGCGCCGCAC
AGGTCTCACGCAGCAGCAGGTGAAGGGCCAGCTCCAGGAGCTCCTGAAA
AGCCGGCAGCCCGCGCCGCGACCTTGCAGCCCCAGCGGGCGCAGCCCTT
CGCGCAGCCGCTGGGACCCTGGCCCCTGTCGAGTGCAGGGCCAAGGCTTG
TGTTCAATCGTGTGAATCGCCGGCGGGACCCCTCCAAGTCC
CCATCCCTCCAGGGGACCCAGGAGACCTACACACTGGCCCACAAGGAGA
ATGTCCGCTTTGTGTCCGAAGCCTGGCAGCAGAGACTAAGGTGTCAGTAT
GTTCTTCAAGCTGCTCTGCTCCTGGGCCCAAGCTATTCTCCTGCCTCAGCC
TTCCAAGAAGCTGAAACTACAAGAACACAAGACTGTACCTGGCTTGCAAA
CACCATTGCTAATAAGAAGATATTACATGGATCCCTGAGGTTCGGTCCCCA
ATACGACAAGACAATTTGATATCATAATAGAACACTGCAGAAACAATGCT
GAGTGA

FIRRE-181aa

MYTIINGPSKLVAQRRRTGLTQQQVKGQLQELLKSRQPAPPTLQPQRAQPFAQP
LGPWPLSSAGPRLVFNRVNRRRDPSKSPSLQGTQETYTLAHKENVRFVSEAW
QQRLRCQYVLQAALLGPSYSPASAFQEAETTRTQDCTWLANTIANKKILHG
SLRSVPNTTRQFDIIIHCRNNAE

XIST-ORF

ATGTGTGTAAGTGCACATGGCCCATCCCATCTGAATAAGGTCCTACTCTCA
GACCCCTTTTGCAGTACAGTAGGTGTGCTGATAACCAAGGCCCTCTTCCT
GGCCTGTAAACGTATGTGATTATATTTGTCTGGGTTCAGTGTATAAGACA
TGGAAGCCTCCCCTGCCCCACCCACCCCTCAATCTTCCTTTCCCTTCTGGC
AGGGAGTGCCAGCTCCATAAGAACCTTACATTTGGACAGTCAAGGTGCAC
AATTCTAAGTGACCGCAGCCATGCACCTTGGTCAATAATGTGTGTAAGT
CACACGGCCTATCTCATCTGAATAAGGCCTTACTCTCAGACCCCTTTTGCA
GTACAGCAGGGGTGCTGATAACCAAGGCCCATTTTCCTGGCCTGTTATGT
GTGTGA

XIST-136aa

MCVTAHGPSHLNKVLLSDPFCSTVGV LITKAPLPGLLTYVIIFVWVPVYK TW
KPPLPHPTLNLPFPSGRECQLHKNLTFGQSRCTILSDRSHAPWSIMCVTAHGLS
HLNKALLSDPFCSTAGVLITKAHFPGLLCV

FIRRE did not pass the initial screen (procedure described in Supplementary Figure 4, cloning an in-frame GFP after the ORF). We generated antibodies against rest 4 potential peptides (including PINT87aa), but other 2 antibodies worked not so well except LINC-PINT and TTN-AS1. TTN-AS1-93aa is still under investigation in our lab. TTN-AS-1 was reported to play some parts in carcinogenesis. For example, TTN-AS-1 promotes esophageal squamous cell carcinoma progression and metastasis (Clin Cancer Res. 2018 Jan 15;24(2):486-498). Further investigation needs to be done considering whether TTN-AS1-93aa has distinct function or not.

6) Figure 2B and Supplementary Figure 1 – According to circBASE (see below) there are three more annotated circular RNA can be backspliced from the host non-coding gene LINC-PINT. Author should check the expression of those circRNAs too and analyze whether they have coding potential or not.

Response:

We thank the reviewer for this good question. CircBASE reported 6 circular RNA variants of LINC-PINT (<http://www.circbase.org/FLJ43663>), including hsa_circ_0082387, hsa_circ_0007239, hsa_circ_0082388, hsa_circ_0001744, hsa_circ_0001745 and hsa_circ_0082389. Hsa_circ_0082389 is what we focused on in this study. Through homology comparison, only hsa_circ_0082388 and hsa_circ_0082389 have conserve cross-species ORFs while other four circRNAs have no conserve ORFs longer than 50 amino acid (Also, these four circRNAs were mostly generated from intron). The non-conserve and short ORFs implied the low coding potential. We used q-PCR to detect the hsa_circ_0082388 expression previously (see below). Compared to hsa_circ_0082389, hsa_circ_0082388 expression was extremely low in cells and tissues, indicating that it is not the dominant variant of LINC-PINT.

Based above information, we finally chose hsa_circ_0082389 as our primary candidate.

7) Fig. 2E (Left) - RNase R treatment is widely used to increase the circular isoform by depleting most of linear RNAs. It is quite surprising that there was no enrichment of circular isoform after RNase R treatment.

Response:

We thank the reviewer for this good question. RNase R treatment usually is used to validate the resistance of circRNAs to digestion. We checked some similar references listed below. Ivanov A et al. tested some circRNAs' resistance to RNase R and their results showed no enrichment of these candidates (Cell Rep. 2015 Jan 13;10(2):170-7. doi: 10.1016/j.celrep.2014.12.019., Figure 3A). Boeckel JN et al. also validated that 7 of 8 circRNAs were not enriched after RNase R treatment by q-PCR (Circ Res. 2015 Oct 23;117(10):884-90. doi: 10.1161/CIRCRESAHA.115.306319. Figure 2D). To our knowledge, linear RNAs are sensitive to RNase R treatment, but circRNAs are partially resistant to this treatment. Lower concentration RNase R digestion usually maintained circRNAs level unchanged or slightly lower, but circRNAs often showed 10-fold more resistant than linear RNAs.

3A

(A) CircRNA candidates from high-, medium-, and low-expression sets were assayed by qPCR with divergent primers and RNase R treatment. Linear control: VCL, GAPDH, TFRC; positive control: a known circRNA (Memczak et al., 2013).

Circ Res. 2015 Oct 23;117(10):884-90.

doi:10.1161/CIRCRESAHA.115.306319. Figure 2D

D, Quantification of circRNAs after RNase R digestion by RT-PCR (n=3).

8) Figure 2F and Supplementary Figure 2 - The authors did not show any data by FISH or qPCR/circRNA localization in GBM cells, only in normal

brain and HEK293 cells. Side by side comparison of the expression/localization of circPINT RNA in normal brain vs. tumor tissue and malignant non-malignant cells would be required.

Response:

We thank the reviewer for this good question. In the revised Figure 2, we provided the FISH images of circPINT_{Exon2} and LINC-PINT in BTICs and tumor tissues and hNSC.

9) Authors did not consider checking the status of p53 before using these cell lines. The p53 status is different among these cell lines (U87- wild type and U251 - mutant). Though author showed the expression of p53 upon overexpression of PINT87aa (Supplementary Figure 12a), it will be interesting to show the expression of the circPINT and its translated protein upon the inducing or inhibiting p53 activity in these cell lines as it has been shown that LINC-PINT expression is induced in a p53-dependent manner (PUBMED ID - 24070194)

Response:

We thank the reviewer for this good question. We considered the p53 status in these different cells (U87 wild type and U251 mutant). We used these cells which have different p53 status to show that PINT87aa overexpression did not alter p53 level and p53 downstream targets (no matter wild type (U87) or mutant p53(U251)), and to indicate that PINT87aa worked as an independent factor of p53. A172 and 293T are p53 wild-type cells. To address this point that whether p53 regulates PINT87aa, we overexpressed p53 in A172 and 293T cells. Results showed that circPINT_{Exon2} and PINT87aa were increased accordingly (revised Supplementary Figure 12). We believed that p53 could enhance *LINC-PINT* gene transcription. Thus, the pre-mRNA of *LINC-PINT* increased after p53 overexpression. LINC-PINT linear RNA and circPINT_{Exon2} formed by alternative splicing from the pre-mRNA, both sequentially upregulated in these cells.

10) The peptide sequence spanning circRNA back-splice junction can be explicitly recognized as circRNA-encoded products. Since the peptide from circPINT originated from the region which is on the exon2 of linear host gene, it is reasonable to suppose that the peptides identified might also translated from the linear part. The should clone the full length of LINC-PINT (3144 nt) in their expression vector and check if overexpressing full length host gene can also increase the expression of circPINT87aa protein (See the sequence of LINC-PINT and associated circular) to rule out the possibility that identified peptide is only from the circularPINT not from the linear isoform.

Response:

We thank the reviewer for this good question. We provided the IGV plot of the LINC-PINT exon 2 from NHA and U251 cells (see below). Exon 2 of LINC-PINT has a higher reads number compared with exon 1 and 3. If LINC-PINT was translatable, exon 1 and 3 should have similar reads numbers as exon 2 (Non-junction reads from circPINTexon2 were default considered as linear reads, which may cause the high reads number in RNC-seq). Pamuduriti et al. used ribosome footprinting and identified the translation potential of circMbl in fly heads (Mol Cell. 2017 Apr 6;66(1):9-21.e7. doi: 10.1016/j.molcel.2017.02.021). They found that the RFP pattern around the stop codon of circMbl, which strongly suggested the coding ability. In our RNC-seq, the reads were also enriched around the sORF stop codon in LINC-PINT exon 2.

Revised Figure 2A. IGV plot of LINC-PINT exon 2 reads in RNA and RNC sequencing.

IGV plot of normal mapping reads within LINC-PINT

To further address this concern, we used the LINC-PINT overexpression plasmid as shown in previous Figure 2F, lower panel. As shown by FISH, introduction of the

plasmid increased the LINC-PINT expression in 293T cells. We used q-PCR to detect LINC-PINT level quantificationally, as shown in revised Figure 3H. Overexpression of LINC-PINT expression vector in hNSC and 293T cells both successfully up-regulated LINC-PINT RNA expression (Revised Figure 3H, lower panel). However, PINT87aa expression in LINC-PINT overexpressed hNSC or 293T cells was not changed (Revised Figure 3H, upper panel). Together with our previous data that using LINC-PINT specific ASO did not change PINT87aa level, we conclude that PINT87aa was not encoded by LINC-PINT but by circPINTexon2.

We found that most coding circRNAs were generated from coding exons (unpublished data). As the reviewer mentioned, peptide sequence spanning circRNA back-splice junction can be explicitly recognized as circRNA-encoded products, as we previously published FBXW7-185aa and SHPRH-254aa (J Natl Cancer Inst. 2018 Mar 1;110(3). doi: 10.1093/jnci/djx166; Oncogene. 2018 Mar;37(13):1805-1814. doi: 10.1038/s41388-017-0019-9.). However, most of circRNAs translated peptides have the same amino acids as part of their host gene's products. Cross-junction ORF encoded peptides are much easier to distinguish from their host gene's products compare with non-cross-junction ORF encoded peptides, which is why we chose non-coding host RNA in this study.

11) Figure 5 - Showing merely cell proliferation and colony formation assay is rather disappointing. What about limiting dilution assay, apoptosis, migration/invasion drug/radiation resistance. The author should test whether the expression of PINT87aa or circPINT would lead to the activation of common or distinct proto-oncogenic drivers (EGFR, PDGFR, MET).

Response:

We thank the reviewer for these suggestions. In the revised manuscript, we tested above functional experiments in BTICs and the results were provided in the revised Figures. Generally, PINT87aa decreased the BTIC self-renewal (reduced limited dilution efficiency), proliferation and radiation resistance. However, invasion

difference after PINT87aa overexpression may be caused by reducing cell proliferation rate. Furthermore, EGFR, PDGFR and MET expression were not affected by PINT87aa overexpression.

12) Statistical analysis is of rather poor quality.

Response:

We sincerely apologize for the unsatisfied statistical analysis. A statistician was invited and revisited our statistical analysis in this study.

13) Few samples of non-malignant brain were analyzed

Response:

We thank the reviewer for these good suggestions. We selected more normal brain tissues randomly from our tissue bank and tested the PINT87aa expression, which was shown in revised Figure 4C.

Reviewers' Comments:

Reviewer #1:

Remarks to the Author:

The manuscript describes a comprehensive and interesting study that provides novel insights into the largely unexplored world of circular RNAs (circRNA). Mainly, they characterised a circular RNA originating from a long non-coding RNA, and they showed that this circRNA encodes an 87 amino acids peptide. Furthermore, they found that the peptide interacts with PAF1 and thereby inhibits transcription of oncogenes. As a result, it suppresses the proliferation of cancer cells.

Overall the authors provided an excellent piece of work that will catch the attention from a wide scientific community to the unexplored realm of circRNAs and the implication of their expression on cell functions. The authors have provided a refined version of the manuscript and have provided satisfactory responses and corrections to the comments raised in the previous submission.

Reviewer #2:

Remarks to the Author:

The authors have addressed most of the questions. However, I am very concerned about the expression of this circRNA. Indeed, we have used the primers (QPCR-ciR-PINT, GCGTTCAGCCCTGGGGTCATAT and CAGTTTTTCTCTCCGAGCTA, Supplementary Table1) to explore the expression of this circRNA in different cell lines. Unexpectedly, we could not detect any expression of this circRNA in different cells including HEK-293T cells. How the authors detect this circRNA with high expression in HEK-293T cells by qRT-PCR (the same primers), northern blot and RNA FISH??? The expression of this circRNA should be validated by at least another independent lab using the provided primers. The authors should also provide the raw results of the qRT-PCR (eg. Ct value).

Reviewer #3:

Remarks to the Author:

In my opinion the manuscript is greatly improved and authors have answered all the concerns raised with good explanations.

However I have some remaining concerns:

1. Figure 4A, shows localization of PINT87aa-GFP and GFP (vector control) in two cell lines- 456 and 4121 BTIC. In both these the GFP expression seems to be predominantly nuclear, which should not be the ideal case as GFP should be pan-cellular with uniform distribution across the cell. This observation raises a serious concern that may be the vector used for the study has some predisposition to localize proteins in nucleus.
2. Figure 6F,G and H shows a clear correlation between PAF1, PolII and PINT87aa based on which authors conclude that PINT87aa is important for enabling anchorage of PAF1 to its target genes, which could be one of the interpretation. However ChIP-qPCR analysis is important for this and also authors need to show that PINT87aa binds to DNA by EMSA and it further also increases the binding of PAF1 complex to its target site. This especially is very important considering that, it is now already published that PAF-1 heterodimerises with Leo1 to bind directly to histone H3 and not necessarily to DNA.
3. Authors when explain the vector construction in Supplementary figure 4 write in its legend of western blot with GFP antibody, the data of which is not there in the Figure.

The authors should provide some explanations if not perform experiments for this before the article should be accepted.

Reviewer #5:

Remarks to the Author:

As protein-coding sequences accounts for only less than 2% of the genome, it has become increasingly apparent that aberrations of the non-protein-coding genes, including lncRNAs, and relatively recently discovered circular RNAs act as a functional transcript in the human transcriptome; driving important developmental and pathological programs. To-date research on ncRNA showed their multifunctional role and tissue/organ/cell-specific expression and circRNAs expression and function are increasingly in the scope of interest in this dynamic and important area of research, especially in the brain where circRNAs are highly expressed. Thus, basic research addressing structural and signal transduction mechanistic studies are needed in carefully planned models. Unfortunately, the model used in in vitro and in vivo studies by authors does not address such obstacles. This criticism is not only for the benefit of readers but also to prevent false positive targets/tools. Authors included large number of improvements of this manuscript, but unfortunately the recapitulation of some assays using different set of cells without transcriptomic/phenotypic characteristic (especially in case of p53 status/chromosome 7 amplification/ hypermethylation status in vivo) did not fully addressed the concerns of the model. In general, these novel findings should be more focused and precise.

The glioblastoma is highly heterogeneous tumor – how these linc/circ RNAs are expressed in specific subclasses? E.g. SOX2 was shown by authors as downstream target of PINT87aa-PAF1, but its expression is subtype specific; similarly, c-myc expression is different in different anatomic niches of the tumor (prevalent in perinecrotic part). Authors showed that PINT87aa is downregulated in GBM - how this can explain different expression of its downstream targets?

Although authors provided some experiments (in vitro cell assays in vivo assays) on patient derived cells and found similar phenotype, the mechanistic study depends on assays performed in 293 cells or SW1783 etc. ncRNA are multifunctional and may have different functions depending on the cell origin, and it is not addressed in presented manuscript.

The expression/overexpression of p53 is not necessarily functional as it needs to be activated by stress. Experiment provided by authors did not fully address concern about p53 status.

Authors used genome editing by CRISPR–Cas9 to knock out circ PINT. Recently (Nature Medicine PMID: 29892067) it was shown that this strategy induces p53-mediated DNA damage response and cell cycle arrest leading to a selection against cells with a functional p53 pathway – so p53 should be closely monitored when developing cell-based strategy utilizing CRISPR–Cas9.

Hypermethylation is associated with a gain of chromosome 7, a hallmark of glioblastoma, and may compensate for tumor-driven enhanced gene dosage as a rescue mechanism by preventing undue gene expression. What is the mechanism of circ PINT (located on ch7) loss ?

Additional bands on Western blots may suggest posttranslational modification of peptide PINT87aa/covalent modification etc. Mass spectrometry with these bands should be performed to validate these new antibodies and other potential splicing forms of peptide.

Fig6E in text should be Fig6D – besides co-localization in irrelevant 293cells does not mean that reintroduction of circ PINT into glioma cells would be functional in the same way, other indispensable partners (e.g. chaperons necessary for nuclear localization) can be absent in glioma cells.

Authors initiated this project from global screen analysis to discover circRNA translational peptide – but with 2 samples only this is not representative approach.

What is the ratio of translation circPINT to peptide, the analysis of number of molecules /per peptide was not performed.

Authors' concluded that alternative splicing results in the emergence of peptide from ncRNA – (line 99) - is this the fact? if peptide is produced it is no longer ncRNA and its annotation should be revised.

Authors stated that circ RNA showed no difference in chromosome distribution – with obvious gain in ch7 and loss of ch10 in glioblastoma this should be discussed or perhaps the model used can't demonstrate it and sampling for initial screen was not carefully justified for this study.

All figures should have the same font size as in Fig 7, otherwise is hardy legible.

Point by point Response to Reviewers

Manuscript NCOMMS-17-27528A

Reviewer #1, Expertise: Transcriptome/Translatome sequencing (Remarks to the Author):

The manuscript describes a comprehensive and interesting study that provides novel insights into the largely unexplored world of circular RNAs (circRNA). Mainly, they characterised a circular RNA originating from a long non-coding RNA, and they showed that this circRNA encodes an 87 amino acids peptide. Furthermore, they found that the peptide interacts with PAF1 and thereby inhibits transcription of oncogenes. As a result, it suppresses the proliferation of cancer cells.

Overall the authors provided an excellent piece of work that will catch the attention from a wide scientific community to the unexplored realm of circRNAs and the implication of their expression on cell functions. The authors have provided a refined version of the manuscript and have provided satisfactory responses and corrections to the comments raised in the previous submission.

Response:

We sincerely thank the reviewer for the constructive suggestions, which had improved the manuscript significantly.

Reviewer #2, Expertise: circular RNA (Remarks to the Author):

The authors have addressed most of the questions. However, I am very concerned about the expression of this circRNA. Indeed, we have used the primers (QPCR-ciR-PINT, GCGTTCAGCCCTGGGGTCATAT and CAGTTTTTCTCTTCCGCAGCTA, Supplementary Table1) to explore the expression of this circRNA in different cell lines. Unexpectedly, we could not detect any expression of this circRNA in different cells including HEK-293T cells. How the authors detect this circRNA with high expression in HEK-293T cells by qRT-PCR (the same primers), northern blot and RNA FISH??? The expression of this circRNA should be validated by at least another independent lab using the provided primers. The authors should also provide the raw results of the qRT-PCR (eg. Ct value).

Response:

We sincerely thank the reviewer for his careful and comprehensive comments. The most possible reason that the reviewer did not detect circPINTexon2 in 293T may due to the PCR condition.

We chose the divergent primers according to the circPINTexon2 junction, where do not have many designing-options. These divergent primers need higher annealing temperature. Also, PCR for circRNAs need the denature step to open the secondary structure.

The following are our PCR condition:

Reverse transcription (20 μ L)

HiScript II 1st Strand cDNA Synthesis Kit (Vazyme R212)

A

Random hexamers (50 ng/μl) 1 μL

Total RNA 1 μg

RNase free ddH₂O to 10 μL

65 °C 5min, then keep on ice

B

10 ×RT Mix 2μL

HiScript® II Enzyme Mix 2μL

RNase free ddH₂O to 10 μL

A and **B** mixed: 25 °C 5min, 37 °C 10min, 42 °C 20min, 85 °C 5min

Polymerase Chain Reaction (PCR) (20 μL)

2 × AceTaq Master Mix (Vazyme P412)

2 × PCR Mix 10μL

CDNA 0.5 μL

F-primer (10 μ M) 0.5 μL

R-primer (10 μ M) 0.5 μL

RNase free ddH₂O to 20μL

PCR condition : 95 °C 5min, (95 °C 30s, 60-68 °C 1min) 40 PCR cycles, 72 °C 3min

The PCR results are showed as below:

To confirm the 63 °C amplified products are our target sequences, we send the indicated PCR products for sequencing, shown as below:

Sequencing results clearly showed that the PCR products are exact the predicted

cross-junction sequences on circPINTexon2.

Base on above results, we performed the q-PCR by using the same condition.

Quantitative Real-time PCR(Q-RT-PCR) (20 μ L)

AceQ Universal SYBR qPCR Master Mix (Vazyme Q511)

2 \times QPCR Mix 10 μ L

CDNA 0.5 μ L

F-primer (10 μ M) 0.5 μ L

R-primer (10 μ M) 0.5 μ L

RNase free ddH₂O to 20 μ L

q-PCR condition: 95 \square 5min, (95 \square 30s, 63 \square 1min) 40 PCR cycles, Collecting fluorescence signal at 63 \square

The q-PCR results are showed as below:

Circ-PINT Amplification Plot

β -actin Amplification Plot

Circ-PINT Melting Curve

β -actin Melting Curve

Professor Gong Zhang, worked at Jinan University and run his own lab, successfully amplified the target products from RNC-RNAs, which expression should be much lower than total RNAs. He used 68 °C for the RNC-RNA annealing (different enzyme system), two steps PCR. His results were showed below.

1: RNC 65°C退火 30s, 72°C延伸 30s

2: RNC 68°C退火兼延伸1min, 两步法PCR

1. RNC 65 °C 30s, 72 °C 30s
2. RNC 68 °C 60s, two steps PCR

We also showed the raw Ct value previously reported in the manuscript (see the attached Excel files).

Compare to linear LINC-PINT, circPINTexon2 expression was obviously lower (Ct value at around 29, internal control actin around 16) and was coincident with other experimental results such as Northern blot and FISH. We have corrected the description “highly expressed” which was used to describe circPINTexon2 in the previous manuscript. Clearly linear LINC-PINT is the major form of *LINC-PINT* gene transcripts. However, circPINTexon2 encoded PINT87aa has independent functions,

implying that gene's functions may be diversified-executed through different splicing variants. Besides, although circPINTexon2 is not a very highly expressed circular RNA, PINT87aa is not a very low-expressed protein. We inferred this may be due to the stability and the translation efficiency of circRNA may be higher than linear mRNA. A most recently published paper indicated that translatable circular RNA could produce more protein than linear RNA, probably due to the stability and translation efficiency of circRNA, which in part supported our conclusion (Nat Commun. 2018 Jul 6;9(1):2629). Specifically, in human cancers, both linear and circular LINC-PINT was down-regulated. Restore the PINT87aa expression partially rescued the malignant phenotype, which suggested *LINC-PINT*'s tumor-suppressive role was carried out by both nuclear localized LINC-PINT and circRNA translated PINT87aa.

Reviewer #3, Expertise: Transcriptional regulation of gene expression (elongation step) (Remarks to the Author):

In my opinion the manuscript is greatly improved and authors have answered all the concerns raised with good explanations.

However I have some remaining concerns:

1. Figure 4A, shows localization of PINT87aa-GFP and GFP (vector control) in two cell lines- 456 and 4121 BTIC. In both these the GFP expression seems to be predominantly nuclear, which should not be the ideal case as GFP should be pan-cellular with uniform distribution across the cell. This observation raises a serious concern that may be the vector used for the study has some predisposition to localize proteins in nucleus.

Response:

We sincerely thank the reviewer for his careful and comprehensive comments. As we described previously, overdose of GFP transfection could induce the aggregated localization. Although we reduced the GFP-vector transfection dose, it looked still “aggregated” in the nucleus. To further address this point, we used other PINT87aa expression vectors: PINT87aa-RFP. We performed the IF in 4121 and 456 BTICs which were transfected with this plasmid. RFP based IF images were shown. RFP signaling was expressed all over the cells and PIN87aa-RFP was only in the nucleus. These results excluded the predisposition of the GFP vector.

2. Figure 6F,G and H shows a clear correlation between PAF1, PolIII and PINT87aa based on which authors conclude that PINT87aa is important for enabling anchorage of PAF1 to its target genes, which could be one of the interpretation. However ChIP-qPCR analysis is important for this and also authors need to show that PINT87aa binds to DNA by EMSA and it further also increases the binding of PAF1 complex to its target site. This especially is very important considering that, it is now already published that PAF-1 heterodimerises with Leo1 to bind directly to histone H3 and not necessarily to DNA.

Response:

We sincerely thank the reviewer for these good suggestions. We provided the ChIP-qPCR results according to the Figure 6F, G, H experimental design. The results were consistent with the RT-PCR assay, as shown in the revised Figure 6.

We also understand the reviewer’s concern, that EMSA could provide clear evidences that PINT87aa directly interact DNA and increasing the binding of PAF1 complex to its target site. However, we still don’t know whether PINT87aa is a transcription factor or not, or its direct interact with DNA or not. To our knowledge, PAF1’s

function is dependent on its interacting partner. There are five known proteins in PAF1 complex: PD2/Paf1, parafibromin, Leo1, Ctr9 and Ski8. Parafibromin interaction will inhibited Cyclin D1 expression; Ski8 interaction will change PAF1 complex localization; PD2/PAF1 exhibited an oncogene's function (Oncogene. 2007 Nov 29;26(54):7499-507). We also noticed that PAF1/Leo1 complex is required to prevent the spreading of heterochromatin into euchromatin by mapping the heterochromatin mark H3K9me2 (EMBO Rep. 2015 Dec;16(12):1673-87). Another report showed that cells lacking Leo1 have reduced PAF1 complex recruitment as well as decreased levels of trimethylated H3K4 within transcribed chromatin (J Biol Chem. 2010 Oct 29;285(44):33671-9). Above information indicated that the interaction partners are critical for PAF1 complex' function and localization. We showed by IP and CHIP that PINT87aa is a newly identified PAF1 interaction partner and involved in Pol II transcription elongation. Although no direct evidence confirming that PINT87aa is binding to DNA, gain and loss function tests showed that PINT87aa is critical for PAF1 target gene's mRNA expression. In another way, whether PINT87aa directly interacted with DNA or not is not the crucial condition for its biological function so far we identified.

We added these discussions in the revised manuscript, and EMSA is one of our future-plans to decide whether PINT87aa is a transcription factor or not. Clearly, as a novel peptide, CHIP sequencing is needed to determine the potential conserve DNA binding domains for PINT87aa, which needs substantial more work and time.

3. Authors when explain the vector construction in Supplementary figure 4 write in its legend of western blot with GFP antibody, the data of which is not there in the Figure.

Response:

We sincerely thank the reviewer for this good question. We missed the GFP western blot is S Figure 4. Now we had it added to the revised S Figure 4.

The authors should provide some explanations if not perform experiments for this before the article should be accepted.

Response:

We had added the discussion part to the revised manuscript, especially the PAF1 related questions. Although we did not perform EMSA in this paper, we believe we had reported a novel functional peptide which clearly involved in the PAF1 complex regulation, thus sequentially inhibited target gene's elongation. Future work is already planned in our lab, including the PINT87aa ChIP sequencing.

Reviewer #5, replacement referee for Reviewer #4, Expertise: GBM, mouse models (Remarks to the Author):

As protein-coding sequences accounts for only less than 2% of the genome, it has become increasingly apparent that aberrations of the non-protein-coding genes, including lncRNAs, and relatively recently discovered circular RNAs act as a functional transcript in the human transcriptome; driving important developmental and pathological programs. To-date research on ncRNA showed their multifunctional role and tissue/organ/cell-specific expression and circRNAs expression and function are increasingly in the scope of interest in this dynamic and important area of research, especially in the brain where circRNAs are highly expressed. Thus, basic research addressing structural and signal transduction mechanistic studies are needed in carefully planned models. Unfortunately, the model used in in vitro and in vivo studies by authors

does not address such obstacles. This criticism is not only for the benefit of readers but also to prevent false positive targets/tools. Authors included large number of improvements of this manuscript, but unfortunately the recapitulation of some assays using different set of cells without transcriptomic/phenotypic characteristic (especially in case of p53 status/chromosome 7 amplification/ hypermethylation status in vivo) did not fully address the concerns of the model. In general, these novel findings should be more focused and precise.

The glioblastoma is highly heterogeneous tumor – how these linc/circ RNAs are expressed in specific subclasses? E.g. SOX2 was shown by authors as downstream target of PINT87aa-PAF1, but its expression is subtype specific; similarly, c-myc expression is different in different anatomic niches of the tumor (prevalent in perinecrotic part). Authors showed that PINT87aa is downregulated in GBM - how this can explain different expression of its downstream targets?

Response:

We sincerely thank the reviewer for this good question. We agree with that glioblastoma is a highly heterogeneous tumor. To date, the four-subtype category based on molecular classification was well accepted (Cancer Cell. 2010 Jan 19;17(1):98-110). However, there was no patient-derived glioblastoma cells that could precisely reflect these subtypes (e.g. 4121, 456 BTICs cannot be classified as any subtype). Clearly the subtype cell models are needed if we hope to investigate PINT87aa expression on protein level.

On RNA level, in randomly selected several glioma samples including WHO I-IV, both LINC-PINT and circPINTexon2 were downregulated. Although the classification was not based on the sub-type category, the results still suggested that linear and circular PINT are both tumor-suppressor. We actually are doing the molecular classification of our GBM tissue-bank (more than 600 GBM samples from 2008-2018), but the transcriptome sequencing need a lot of time. Although TCGA may provide similar results, it is not easy to download such huge amount data from China mainland. Nevertheless, after established the molecular classification of our own database, we will investigate the circPINTexon2 expression in each sub-type. MYC and SOX-2 are well addressed tumor markers for GBM. We checked several high-throughput assays, SOX-2 and MYCN are frequent gains of genes in GBM (Cell. 2013 Oct 10;155(2):462-77; Cell. 2016 Jan 28;164(3):550-63). We agree with the reviewer that SOX-2 and MYC expression may diversified in individual tumors, but their generally expression in GBM were high. Besides, SOX2 and MYC regulation may decide by many factors, PINT87aa possibly are only one of them. The heterogenicity of GBM may explain that some GBM have low PINT87aa but still have higher level of SOX2 and MYC, et al.

We acknowledged the above limitations of our finding in the discussion part.

Although authors provided some experiments (in vitro cell assays in vivo assays) on patient derived cells and found similar phenotype, the mechanistic study depends on assays performed in 293 cells or SW1783 etc. ncRNA are multifunctional and may have different functions depending on the cell origin, and it is not addressed in presented manuscript.

Response:

We sincerely thank the reviewer for this good question. When determining the PAF1 and PINT87aa interaction domains, we used 293T cells because it is a common cell model for mechanistic research, especially truncated-plasmid overexpression. Also,

the co-localization of PAF1 and PINT87aa was determined in NSC (Figure 6D). Besides, 456 and 4121 BTICs were used to determine the PAF1 target genes after PINT87aa overexpression (Figure 6E); ChIP assay was also performed in NSC, 4121, 456 cells to show the PINT87aa-PAF1 co-operation (Figure 6G, H).

SW1783 and Hs683 was only used for PINT87aa K.O., as these cells has little PINT87aa expression and after PINT87aa K.O. they showed enhance malignant phenotype (Figure 5).

Based on above information, we showed that PINT87aa played a tumor-suppressive role in GBM at our best attempt. Specifically, there were more than just 293T and SW1783 et al in the mechanistic research.

The expression/overexpression of p53 is not necessarily functional as it needs to be activated by stress. Experiment provided by authors did not fully address concern about p53 status.

Response:

We sincerely thank the reviewer for this good question. We noticed in a previous report showed that p53 pathway was found to be dysregulated in 85.3% GBM (214/251), through mutation/deletion of p53 (27.9%), amplification of MDM1/2/4 (15.1%) and/or deletion or CDKN2A (57.8%) (Cell. 2013 Oct 10;155(2):462-77). All above dysregulation may cause wild type p53 loss function in GBM.

Another previously report showed that LINC-PINT is regulated by p53 at the transcriptional level (Genome Biol. 2013;14(9):R104). We inferred that circPINTexon2 may be also regulated by p53, and the loss-function of p53 in GBM maybe the reason that circPINTexon2 downregulated in GBM. We showed forced overexpressed wild-type p53 enhanced PINT87aa expression. Also, above mentioned report shown that stress (DNA damage) induced p53 activation also upregulated linear LINC-PINT. As LINC-PINT and circPINTexon2 are both from the same *LINC-PINT* pre-mRNA, we infer that stress induced p53 activation also could up-regulated PINT87aa expression accordingly.

Authors used genome editing by CRISPR–Cas9 to knock out circ PINT. Recently (Nature Medicine PMID: 29892067) it was shown that this strategy induces p53-mediated DNA damage response and cell cycle arrest leading to a selection against cells with a functional p53 pathway – so p53 should be closely monitored when developing cell-based strategy utilizing CRISPR–Cas9.

Response:

We sincerely thank the reviewer for this very good question. We indeed noticed these two reports published recently (Nat Med. 2018 Jun 11; Nat Med. 2018 Jul;24(7):939-946). These reports are focused on two major concerns: 1. Functional p53 could reduce the Cas9 editing efficiency and 2. Successfully edited cells may have p53 mutation and cell replacement therapies using CRISPR/Cas9-engineered hPSCs should proceed with caution (be monitored for p53 function). In our study, we used single colony-pick to generate SW1783 and Hs683 K.O. cells. Although the efficiency is low, the K.O. cell lines were verified by western blot. Also, SW1783 and Hs683 are tumor cells, our study was not related to “benign cells malignant transformation” problem. Nevertheless, the reviewer raised an important issue that should be noticed during future Cas9 application.

Hypermethylation is associated with a gain of chromosome 7, a hallmark of glioblastoma, and may compensate for tumor-driven enhanced gene dosage as a rescue mechanism by preventing undue gene expression. What is the mechanism of circ PINT (located on ch7) loss ?

Response:

We sincerely thank the reviewer for this very good question. To address this concern, we checked several methylation databases. Genomic sequencing data from multiple

cell lines and brain tissues were extracted from the Roadmap Epigenomics Project(<http://www.epigenomebrowser.org/>) database, LINC-PINT genomic region was analyzed (TCGA methylation data was not provided from the internet). Shown as below, there is a CpG island (25 CpGs) in front of the LINC-PINT transcript. In multiple cancer cell lines and normal brain tissue, this CpGs is **lower methylated** (Image 1 and 2). As the reviewer mentioned, we also noticed that most of other CpG island on Chromosome 7 were hypermethylated (Image 3). Although direct evidences from GBM samples was not available, we implied that the DNA methylation may not be the major reason that LINC-PINT down-regulated in GBM. However, the p53 loss-function, seen in 85% GBM, maybe caused LINC-PINT lower expression. Clearly, further methylated ChIP seq from large cohort GBM samples are still needed.

Data source:
Roadmap Epigenomics Project(<http://www.epigenomebrowser.org/>)

Reference genome version: H19

Data source:
Roadmap Epigenomics Project(<http://www.epigenomebrowser.org/>)

Reference genome
 version: H19

Enlarged

RRBS data from
 multiple cell lines
 Methylation
 microarray from
 multiple cell
 lines
 Brain tissue
 MeDIP-seq
 And MRE-seq

BHM: Brain Hippocampus Middle
BGM: Brain Germinal Matrix
NGED: Neurosphere Cultured Cells Ganglionic Eminence Derived
NCD: Neurosphere Cultured Cells Cortex Derived

BHM BS
 BGM BS
 NGED
 BSMedIP
 NCD
 BSMedIP

Additional bands on Western blots may suggest posttranslational modification of peptide PINT87aa/covalent modification etc. Mass spectrometry with these bands should be performed to validate these new antibodies and other potential splicing forms of peptide.

Response:

We understand the reviewer's concern. As we addressed to the previous reviewer 4, the antibody against PINT87aa worked fine in our hand. In many experiments we only detected a single band for PINT87aa, except for several tissue samples. We showed the original western blot film and those extra bands were non-specific bands (much higher molecular weight). Although no evidence so far supported that PINT87aa modification is required for PAF1 binding (purified mutual interaction was done between PINT87aa and PAF1), we still plan to do the Mass spectrometry in future to clarify this novel peptide comprehensively.

Fig6E in text should be Fig6D – besides co-localization in irrelevant 293cells does not mean that reintroduction of circ PINT into glioma cells would be functional in the same way, other indispensable partners (e.g. chaperons necessary for nuclear localization) can be absent in glioma cells.

Response:

We understand the reviewer's concern. We also performed the colocalization in NSC showed in Figure 6D. Besides, as shown in Figure 6G, PAF1 was expressed in both 456 and 4121 BTICs. In these cells, CHIP using PAF1 antibody could capture the down-stream promoter, indicating that PAF1 also localized in the nuclear.

Authors initiated this project from global screen analysis to discover circRNA translational peptide – but with 2 samples only this is not representative approach. What is the ratio of translation circPINT to peptide, the analysis of number of molecules /per peptide was not performed.

Response:

We thank the reviewer for this very good question and we fully understand the reviewer's concern. To address this question, we previously verified our results by using many other published high-through put sequencing data (Journal of Neuroscience, 2014, 34(33): 10924-10936. **GSE51424**), Nature, 2016, 530(7591): 490 **GSE59821**) in the last version of response. Data acquired from above paper was like the result we acquired from RNC-seq. We admitted that there's maybe some false positive or negative results when using only two cell lines for the initial screen. However, the expression pattern and the translation of circPINTexon2 was undeniable in GBM. We now are working on the ribosome seq by using paired clinical GBM samples, which could provide more precise information for future work. But the ribosome seq from tissue samples are highly skilled experiments and needs more time.

Also, we implied that the RNC-seq results could provide the clue of circPINTexon2/PINT87aa ratio. The junction reads from RNC-seq could clearly showed the circPINTexon2 translation level. We showed that the reads number was 15 in NHA-RNC and 7 in NHA-RNA. To estimate how many circPINTexon2 was translating, the junction reads from ribo-seq/junction reads from RNA seq could provide the theoretical translation efficiency. Besides, we acquired 4 more times data from NHA-RNC, the rough ratio should be 0.535. Nevertheless, this is only an estimate results to show the circRNA translation efficiency. A most recently published paper indicated that translatable circular RNA could produce more protein than linear RNA, probably due to the stability and translation efficiency of circRNA, which in

part supported our conclusion (Nat Commun. 2018 Jul 6;9(1):2629). Their conclusion implied that circRNA/peptide ratio should be higher than linear RNA/peptide ratio.

Authors' concluded that alternative splicing results in the emergence of peptide from ncRNA – (line 99) - is this the fact? if peptide is produced it is no longer ncRNA and its annotation should be revised.

Response:

We thank the reviewer for this very good question. Our meaning was that alternative splicing divided the *LINC-PINT* pre-mRNA to linear LINC-PINT and circPINTexon2, which exerted their independent functions. Indeed, our unpublished results from other circRNAs showed that circular spliced variants may have similar, or independent function to their host genes.

We had applied a new access number for PINT87aa, TPA: BK010446. This sequence record will be held confidential until the data or accession numbers appear in print. Further annotation for LINC-PINT maybe also changed after this manuscript is online.

Authors stated that circ RNA showed no difference in chromosome distribution – with obvious gain in ch7 and loss of ch10 in glioblastoma this should be discussed or perhaps the model used can't demonstrate it and sampling for initial screen was not carefully justified for this study.

Response:

We thank the reviewer for this very good question and we fully understand the reviewer's concern. We noticed a previously published circRNA sequencing database by using GBM samples (Nucleic Acids Res. 2016 May 19;44(9):e87).

Figure 6. *CircRNA distribution in normal tissue and gliomas. The upper panel shows the circRNA distribution in different chromosomes in normal tissue and gliomas. The*

circRNA number in GBM was significantly lower compared to normal tissues or oligodendroblastoma (Wilcoxon rank-sum test, P-value = 1.944e-09, 0.0002117 separately); no significant difference in the circRNA number between normal tissue and oligodendroblastoma (Wilcoxon ank-sum test, P-value = 0.1516). The lower panel shows the circRNA number in each chromosomes in normal tissue and gliomas.

In comparison, our sequencing data of the circRNAs distribution was showed below:

Clearly both results showed that the chromosome distribution was not unified in GBM vs Normal brain. However, these differences were not significant, details need to be explored, especially chromosome 7 and chromosome 10. Nevertheless, our sequencing data were similar with previously published one, suggesting that using U251 and NHA could at least partially reflect the circRNAs distribution in GBM. We corrected the sentence in the manuscript and added the reference, to avoid any misunderstanding.

All figures should have the same font size as in Fig 7, otherwise is hardly legible.

Response:

We thank the reviewer for this very good question. We had other figures adjusted to the size of Figure 7.

Reviewers' Comments:

Reviewer #3:

Remarks to the Author:

Authors have performed most of the experiments asked for and also have provided good explanation of their data, which had led to significant improvement in the article. Due to which I believe it should be accepted.

Concern:

Figure 4A: Labelling has to be changed, fluorescent filters are labelled incorrectly.

Reviewer #5:

Remarks to the Author:

Authors did provide some explanation to the previous criticism; however, the main obstacle for the approval of this work is the model.

Authors argued that there are no patient-derived glioblastoma cells that precisely reflect known subtypes – however such data was published, e.g., PMID: 23650391

Authors also argued (not providing any data) that downstream effectors (SOX-2 and MYCN) “are frequent gains of genes in GBM” – but this can also be due to the co-existence of cells from different subtypes within one tumor (see PMID: 24925914).

The status of p53 was not resolved.

Point by Point response of manuscript NCOMMS-17-27528B

Reviewer #3 (Remarks to the Author):

Authors have performed most of the experiments asked for and also have provided good explanation of their data, which had led to significant improvement in the article. Due to which I believe it should be accepted.

Concern:

Figure 4A: Labelling has to be changed, fluorescent filters are labelled incorrectly.

Response:

We thank the reviewer for his comprehensive and constructive suggestions, which have improved our manuscript significantly.

Also, we had corrected the labelling in Figure 4A.

Reviewer #5 (Remarks to the Author):

Authors did provide some explanation to the previous criticism; however, the main obstacle for the approval of this work is the model.

Response:

We thank the reviewer for his valuable concerns. In this manuscript, we used the most popular and well-addressed BTICs to establish the *in vitro* and *in vivo* models^{1,2,3}. We think that although more cells such as glioma cell lines or patients derived BITCs could be further tested for PINT87aa expression, or validate PINT87aa's biological function, similar results are very possible. Although PINT87aa expression was not extensively tested in sub-types of GBM (see next response), we showed that PINT87aa did not affect EGFR, PDGFR or MET et al, which are driver-genes in GBM and are critical for sub-types. We are now establishing molecular classification

based BTIC cell lines, and PINT87aa expression and functions will be tested in these models in our future work.

Authors argued that there are no patient-derived glioblastoma cells that precisely reflect known subtypes – however such data was published, e.g., PMID: 23650391.

Response:

We thank the reviewer for this good question and we carefully searched related information. In the BTICs we used, there was no sub-type information was available about 456 BTICs^{1, 2, 3, 4, 5}. 4121 is reported to be the classical type or mesenchymal type^{6, 7}. 387 is mesenchymal type and 3691 is proneural type (387 and 3691 also had non-detectable PINT87aa expression, although these cells were not used in this manuscript). From current data, we implied that PINT87aa low-expression is commonly seen in GBM. We are working on the generation of sub-type-based patient derived BTICs and will test the PINT87aa expression in more BTICs further. Also, pathologically diagnosed GBM are classified into sub-types in our hospital recently. We will also test the PINT87aa in these newly classified clinical samples. We think these larger numbers of BTICs and clinical samples could address the conclusion that whether PINT87aa expression is generally low regardless of the sub-types.

Authors also argued (not providing any data) that downstream effectors (SOX-2 and MYCN) “are frequent gains of genes in GBM” – but this can also be due to the co-existence of cells from different subtypes within one tumor (see PMID: 24925914).

Response:

We understand the reviewer’s concern. We also noticed this paper the reviewer mentioned. As we explained previously, heterogeneity indeed could cause the effectors variation. This possibility could only be resolved by a large scale of

single-cell level investigation, which is not practical in our lab currently. Thus, we admitted this situation, acknowledged our limitation and added the discussion part in our revised manuscript.

The status of p53 was not resolved.

Response:

We thank the reviewer for this question. The reason we mentioned p53 is that LINC-PINT was reported to be regulated by p53. However, PINT87aa does not involved in p53 related functions according to our data. From current results we cannot get a conclusion that p53 directly regulates PINT87aa. Thus, in the revised manuscript we had toned down the conclusion that correlated to p53 and added further discussion. Substantial further works are required to study how p53 could regulate PINT87aa expression or functions.

1. Lathia JD, *et al.* Integrin alpha 6 regulates glioblastoma stem cells. *Cell Stem Cell* **6**, 421-432 (2010).
2. Bao S, *et al.* Stem cell-like glioma cells promote tumor angiogenesis through vascular endothelial growth factor. *Cancer Res* **66**, 7843-7848 (2006).
3. Jin X, *et al.* Targeting glioma stem cells through combined BMI1 and EZH2 inhibition. *Nat Med* **23**, 1352-1361 (2017).
4. Li Z, *et al.* Hypoxia-inducible factors regulate tumorigenic capacity of glioma stem cells. *Cancer Cell* **15**, 501-513 (2009).

5. Bao S, *et al.* Glioma stem cells promote radioresistance by preferential activation of the DNA damage response. *Nature* **444**, 756-760 (2006).

6. Jung J, *et al.* Nicotinamide metabolism regulates glioblastoma stem cell maintenance. *JCI Insight* **2**, (2017).

7. Alvarado AG, *et al.* Coordination of self-renewal in glioblastoma by integration of adhesion and microRNA signaling. *Neuro Oncol* **18**, 656-666 (2016).